# A statistical framework for analysis of trial-level temporal dynamics in fiber photometry experiments

**Gabriel Loewinger[1]\*, Erjia Cui[2], David Lovinger[3], Francisco Pereira[1]**

[1]Machine Learning Core, National Institute of Mental Health, Bethesda, United States; [2]Division of Biostatistics and Health Data Science, University of Minnesota, Minneapolis, United States; [3]Laboratory for Integrative Neuroscience, National Institute on Alcohol Abuse and Alcoholism, Rockville, United States

## eLife Assessment

This **important** study presents a statistical framework for the analysis of photometry signals and provides an open-source implementation. The evidence supporting the benefits of the presented functional mixed-effect modeling analysis as opposed to (1) summary statistics and (2) other pointwise regression models is **convincing** with a thorough comparison with other methods and datasets. This work will be of great interest to researchers using not only fiber photometry, but other time-series data such as calcium imaging or electrophysiology data, and wanting to implement trial-by-trial temporal analysis, taking also into account variability within the dataset.

**Abstract** Fiber photometry has become a popular technique to measure neural activity in vivo, but common analysis strategies can reduce the detection of effects because they condense *within-trial* signals into summary measures, and discard trial-level information by averaging *across-trials*. We propose a novel photometry statistical framework based on functional linear mixed modeling, which enables hypothesis testing of variable effects at *every trial time-point*, and uses trial-level signals without averaging. This makes it possible to compare the timing and magnitude of signals across conditions while accounting for between-animal differences. Our framework produces a series of plots that illustrate covariate effect estimates and statistical significance at each trial time-point. By exploiting signal autocorrelation, our methodology yields *joint* 95% confidence intervals that account for inspecting effects across the entire trial and improve the detection of event-related signal changes over common multiple comparisons correction strategies. We reanalyze data from a recent study proposing a theory for the role of mesolimbic dopamine in reward learning, and show the capability of our framework to reveal significant effects obscured by standard analysis approaches. For example, our method identifies two dopamine components with distinct temporal dynamics in response to reward delivery. In simulation experiments, our methodology yields improved statistical power over common analysis approaches. Finally, we provide an open-source package and analysis guide for applying our framework.

## Introduction

Fiber photometry is a photonic technique used to measure neural activity in vivo. The assay quantifies bulk fluorescence emitted from fluorescent biosensors that detect neurotransmitters or physiological processes (e.g. calcium influx) with high neurochemical and cell-type specificity (*Cui et al., 2013*; *Gunaydin et al., 2014*; *Simpson et al., 2024*). The popularity of photometry has increased nearly

\*For correspondence:
gloewinger@gmail.com

**Competing interest:** The authors declare that no competing interests exist.

exponentially since its development (*Cui et al., 2013*; *Gunaydin et al., 2014*), with roughly 1500 references to it in the last year alone (see *Appendix 1—figure 1* for an analysis of the number of references to photometry). Although photometry is an invaluable tool, there is little consensus on analysis strategies for the data produced. Many common analysis procedures were not designed for photometry, specifically, but rather grew organically out of adapting approaches historically applied in the cyclic voltammetry (*Phillips et al., 2003*; *Heien et al., 2005*), EEG (*Adrian and Matthews, 1934*), and electrophysiology communities (*Fatt and Katz, 1952*). Arguably the most common photometry analysis strategy proceeds by: (1) averaging event-aligned signals across trials ('trial-averaging') and animals for comparison of different conditions (e.g. treatment/control), (2) graphing each condition's average signal ('trace'), (3) calculating a signal summary measure (e.g. Area Under the Curve [AUC]), and (4) conducting hypothesis tests (e.g. ANOVA) on that summary statistic.

Although these analysis conventions are parsimonious, they may dilute important patterns in the data related to, for example, individual animal differences in the timing and magnitude of signals, and the evolution of signals across trials. Part of the appeal of photometry is that probes can be implanted chronically, thereby enabling its application in sophisticated multi-session ('longitudinal') experiments. Such designs yield, however, complex datasets in which associations between the signal and experimental variables can vary across trials (e.g. due to learning) and animals. To illustrate this, we present a typical analysis of photometry data (*Coddington et al., 2023*) in *Figure 1*. These measurements were collected on mesolimbic dopamine neurons in well-trained, head-fixed animals performing a Pavlovian reward learning task. *Figure 1A* shows that the signals exhibit considerable heterogeneity across animals, suggesting that it can be difficult to identify one summary measure that captures the target effect in all subjects. Even within-animal, traces are highly variable across conditions (*Figure 1B*), trials within-session (*Figure 1C*), and sessions (*Figure 1D*). These figures illustrate how averaging the signal *across trials* can obscure behavior–signal associations, and how summarizing *within trial* signals (e.g. with AUC) can reduce one's ability to distinguish between trial-level signals that differ in dynamics, but yield similar summary values.

Despite the complexity of the data, there are few analysis methods developed specifically for photometry. *Encoding models* of point-by-point signal-behavior relationships have been used for predicting the signal values from behavioral variables (*Markowitz et al., 2023*; *Willmore et al., 2022*; *Choi et al., 2020*). When used for inference, however, these approaches only test whether or not there is an overall behavioral effect on the signal in the analyzed time-window. By not testing the association at each time-point, it makes it difficult to determine when the temporal dynamics of associations are meaningful. Moreover, one model is fit per animal and thus data is not pooled across subjects, which can substantially reduce statistical power. In *Jean-Richard-Dit-Bressel et al., 2020*, the authors propose to compare photometry signals through the combination of *permutation testing* and non-parametric (cluster/subject-level) bootstraps to construct confidence intervals (CIs). This is restricted to comparisons between two conditions, however, which precludes testing for continuous variables, multi-level factors, or multivariate analyses. Investigators have analyzed data from techniques like photometry and calcium imaging by fitting *pointwise generalized linear models (GLMs)* (*Pinto and Dan, 2015*), or Pearson correlations (*Markowitz et al., 2023*) to assess associations between variables and the signal at each trial time-point. However, standard GLMs and Pearson correlations do not yield valid inference when applied to multi-animal repeated measures datasets. This, therefore, requires one to analyze a single trial-averaged signal per animal, discarding trial-level information. Moreover, the methods do not adjust for multiple comparisons of testing at different time-points, which can inflate Type I errors. *Lee et al., 2019* fit *pointwise linear mixed models* and then apply Benjamini-Hochberg correction. However, this method does not yield *joint* CIs and thus does not exploit the correlation between signal values across time-points. The method, therefore, requires one to adjust for a different test at each sample in a trial's signal. Thus, two analyses of the same data, down-sampled with different sampling rates, could yield significant results in one analysis and not the other, simply because higher sampling rates require one to correct for more comparisons. More generally, this method can be very conservative and dramatically hinder the detection of effects.

In sum, we argue that existing photometry analysis approaches reduce the ability to detect effects. Summary measure analyses coarsen information by (1) condensing the photometry signal into a single statistic (e.g. AUC) that summarizes across time-points *within-trial*, and/or (2) averaging *across-trials* for each animal before conducting hypothesis tests. For methods that estimate associations at each

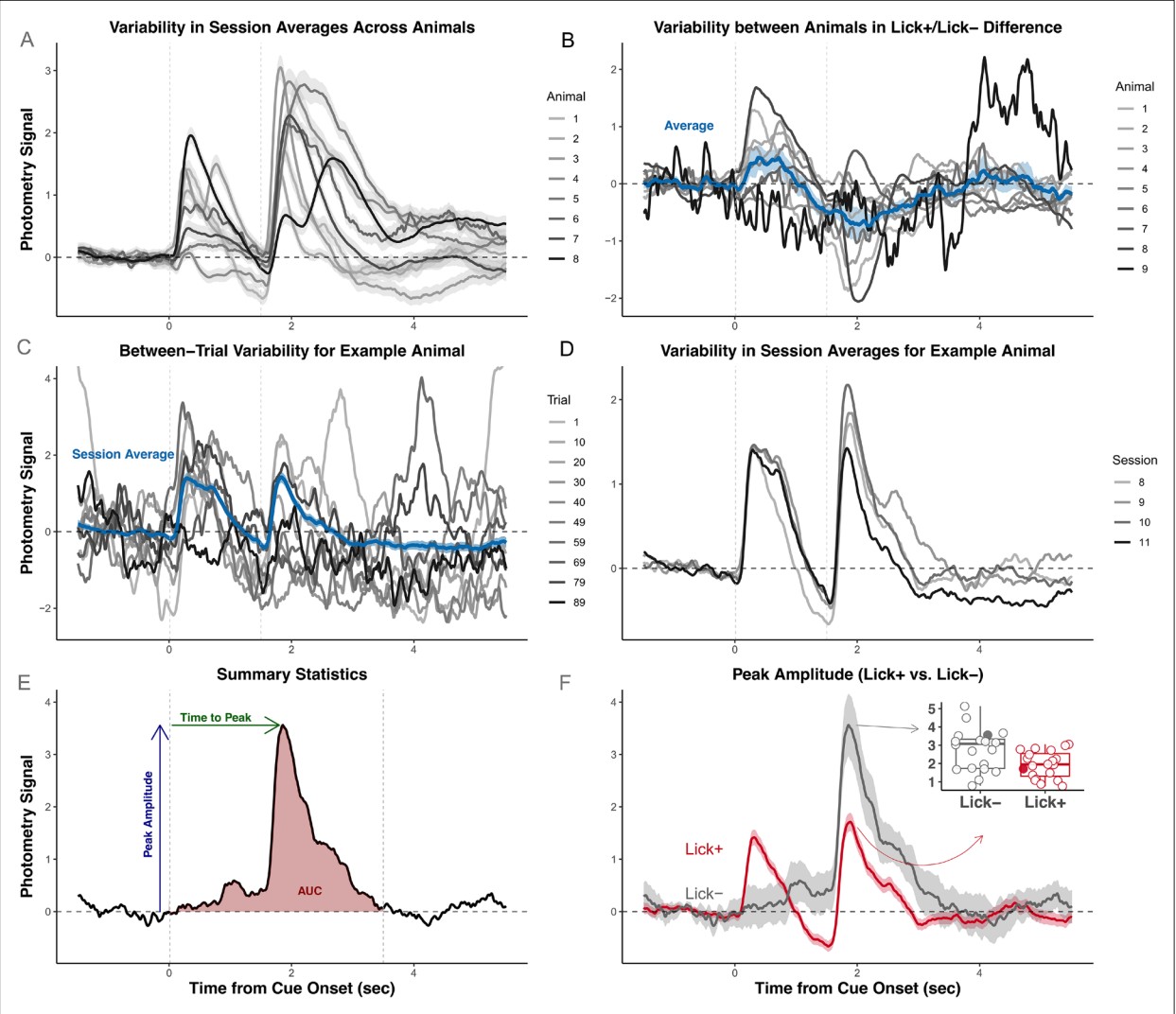

**Figure 1.** Variability in photometry signals highlights the need for trial-level analyses. Signals were recorded from a Pavlovian task in which reward-delivery (sweetened water) followed a stimulus-presentation (0.5 sec auditory cue) after a 1 sec delay. Signals are aligned to cue-onset. (**A**) Signals exhibit heterogeneity across animals. Each trace is a trial-averaged signal on one session for one animal. (**B**) Signals exhibit heterogeneity across animals in the effect of condition. Each trace is from one animal on the same session as in (**A**). Signals were separately averaged across trials in which animals did (Lick+) or did not (Lick-) engage in anticipatory licking. Each trace represents the pointwise *difference* between average Lick+ and Lick- signals. (**C**) Signals exhibit heterogeneity across trials within animal. Each trace is a randomly selected trial from the same animal in the same session. (**D**) Signals exhibit heterogeneity across sessions. Each trace plotted is the trial-averaged signal for one session for one subject. (**E**) Illustration of common summary measures. Depending on the authors, Area-Under-the-Curve (AUC) can be the area of the shaded region or the average signal amplitude. (**F**) Example hypothesis test of Lick+/Lick- differences using peak amplitude as the summary measure. All signals are measurements of calcium dynamics from axons of mesolimbic dopamine neurons recorded from fibers in the nucleus accumbens (***Coddington et al., 2023***).

trial time-point, effects can be obscured because current approaches do not exploit the correlation across time-points, do not provide *joint* CIs, and thus yield very conservative inference.

We present an analysis framework that fills this gap and extracts more nuanced information from photometry data by (1) enabling hypothesis testing of the effects of experimental variables on signals at *every trial time-point*, and (2) using the signal from every trial and animal in the analysis. Our proposed approach is based on functional linear mixed models and allows one to compare the temporal evolution ('temporal dynamics') of the signal between conditions – in timing and magnitude – while accounting for between-animal differences. The statistical procedure uses (1) mixed effects modeling to enable the analysis of sophisticated nested experiments (e.g. designs that include multiple conditions, sessions, and trials), and (2) functional regression to exploit autocorrelation in

the signal to calculate *joint* 95% CIs. These *joint* CIs account for examining effects throughout the entire trial, but are not overly conservative CIs. Our framework outputs a plot for each covariate in the model (e.g. behavior, cue-type), which shows whether that covariate is significantly associated with the photometry signal at each time-point. The framework, therefore, unifies the stages of plotting signals and then conducting hypothesis tests into a joint analysis and visualization procedure.

## Results

In this section, we introduce our photometry analysis framework based on functional linear mixed models. We focus on explaining the implementation steps and analysis outputs. We then demonstrate how the approach can be used to formulate the scientific questions posed in a recent paper, by re-analyzing their datasets and expanding their results. Finally, we conduct realistic data-driven simulations to show that our approach has desirable statistical properties for photometry analyses.

### Functional Linear Mixed Models (*FLMM*)

Linear mixed models (LMM) are a class of methods for testing the association between covariates ('independent' variables) and outcomes ('dependent' variables) for repeated measures data (see Methods section **Linear mixed models** for a brief introduction to LMM). They can be used to analyze trial-level summary measures (e.g. AUCs) pooled across all trials and animals, preventing the loss of information from trial-averaging. However, this still requires condensing signals into scalar summary measures, which coarsens within-trial information across time-points. In contrast, functional regression methods can be used to model a photometry time series as a 'function,' which makes it possible to test the association between covariates of interest and the signal value *at each time-point in the trial* (see Methods section **Functional linear regression** for a brief introduction to functional regression). However, most functional regression methods require trial averaging of the signals prior to analysis, which discards trial-level information.

The photometry analysis framework we are introducing is based on Functional Linear Mixed Models (*FLMM*), which combines the benefits of LMM and functional regression to extract the information in the signal both *across-* and *within-* trials (*Cui et al., 2022*; *Scheipl et al., 2015*; *Davidson, 2009*; *Morris and Carroll, 2006*). By modeling covariates that can vary between (1) trial (e.g. cue-type, latency-to-press), (2) session (e.g. training stage), and (3) animal (e.g. treatment group), *FLMM* estimates the effects (termed *functional fixed-effects*) and statistical significance of those covariates for longitudinal designs. By further including *functional random-effects*, the framework can also estimate and adjust for between-animal differences in (1) photometry signal dynamics, and (2) effects of a covariate on the signal, within- and across- trials. In essence, this enables one to model between-animal and between-trial variability in both the 'shape' of the signal, and the evolution of covariate effects across trial time-points. The result is a *single plot of the coefficient estimates for each covariate*, which visualizes when (and to what extent) the covariate has a statistically significant (fixed-effect) association with the photometry signal during a trial.

In *Figure 2A*, we illustrate the steps in *FLMM* parameter estimation, which are implemented by our software (see Methods section **Functional mixed models** for details). The first input is a matrix $\mathbb{Y}$, where column $s$, $\mathbf{Y}(s)$, contains the photometry signal value at trial time-point $s$ from every trial in every session and animal, as shown in *Figure 2A(1)*. The other inputs are covariate matrices, $\mathbb{X}$ and $\mathbb{Z}$, associated with the fixed-effects and random-effects regression coefficients, respectively. Then, for each time-point, $s$, we fit a LMM model to $\mathbf{Y}(s)$ as the outcome variable, with the conditional mean of animal $i$ modeled as:

$$\mathbb{E}\left[\mathbf{Y}_i(s) \,\middle|\, \mathbb{X}_i, \mathbb{Z}_i, \gamma_i(s)\right] = \mathbb{X}_i\boldsymbol{\beta}(s) + \mathbb{Z}_i\gamma_i(s). \tag{1}$$

This yields estimates for fixed-effect regression coefficients ($\widehat{\boldsymbol{\beta}}(s)$, common across animals), random-effect coefficients ($\widehat{\gamma}_i(s)$, animal-specific), and $\widehat{\boldsymbol{\beta}}(s)$ *pointwise* 95% confidence intervals (CIs), as shown in the last row of *Figure 2A(1)*. We then smooth the $\widehat{\beta}_k$ across trial time-points for each covariate $k$, as illustrated in *Figure 2A(2)*. The smoothed *pointwise* 95% CI (dark gray) is constructed using a closed form covariance expression when $\mathbf{Y}_i(s) \,|\, \mathbb{X}_i, \mathbb{Z}_i, \gamma_i(s)$ is modeled as Gaussian, or through a bootstrap-based procedure for other distributions. To account for the multiple comparisons of inspecting coefficients across the entire trial, we construct a *joint* 95% CI (light gray), yielding the plots in *Figure 2A(3)*.

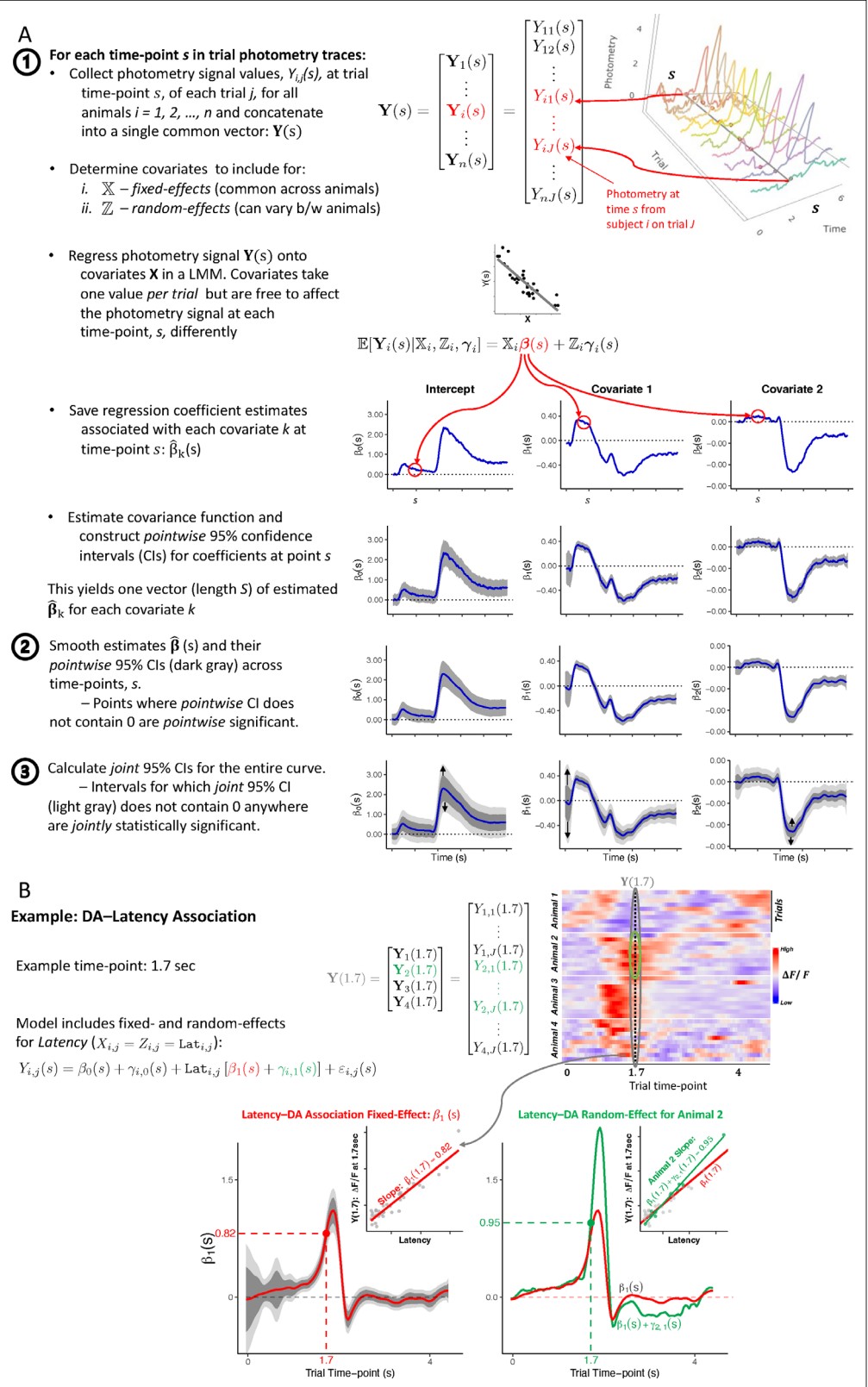

**Figure 2.** *Functional Linear Mixed Models* estimation. (**A**) General procedure. (**B**) Example analysis of `Latency`-signal (Latency-to-lick) association. To illustrate how the plots are constructed, we show the procedure at an example trial time-point ($s = 1.7$ sec), corresponding to values in the heatmap [Top right]. Each point in the FLMM $\beta_1(s)$ coefficient plot [Bottom Left] can be conceptualized as pooling signal values at time $s$ across trials/animals

*Figure 2 continued on next page*

*Figure 2 continued*

(a slice of the heatmap) and correlating that pooled vector, $\mathbf{Y}(1.7)$, against `Latency`, via a linear mixed model (LMM). [Bottom Right] shows how functional random-effects can be used to model variability in the `Latency`-dopamine (DA) slope across animals. The inset shows how at $s = 1.7$, the model treats an example animal's slope (green), $\beta_1(1.7) + \gamma_{2,1}(1.7) = 0.95$, to differ from the shared/common fixed-effect (red), $\beta_1(1.7) = 0.82$.

We detail the *joint* 95% CI estimation procedure, the subject of our statistical methodology contribution, in section **Functional mixed models** and the **Discussion**. By treating the signal as a 'function,' *FLMM* exploits the correlation across trial time-points to construct narrower *joint* CIs, thereby enabling one to identify more significant effects.

*Figure 2B* illustrates the *FLMM* output of an example analysis. The inset in *Figure 2B* [bottom left] shows how the effect at a given time-point is estimated and interpreted: when correlating (with an LMM) the signal values (pooled across trials and animals) measured at trial time-point $s = 1.7$ sec with the covariate, `Latency`, the slope of that line is $\widehat{\beta}_1(1.7) = 0.82$. More generally, the interpretation of the *FLMM* plot for covariate $k$ at time-point $s$, $\widehat{\beta}_k(s)$, is the 'average change in the photometry signal at trial time-point $s$ associated with a one unit increase in covariate $k$, holding other covariates constant.' *Figure 2B* [bottom right] shows estimated functional random-effects for an example animal and illustrates how these random-effects model individual differences in the `Latency`–signal association. Each *period* for which the *joint* 95% CI does not contain 0 anywhere denotes that the fixed-effect coefficients are *statistically significantly* different from 0 throughout the entire period. The plot conveys: (1) where effects are statistically significant; (2) the estimated effect magnitudes; and (3) *joint* 95% CIs, thereby providing a *complete*, interpretable, and simplified presentation of statistical results.

## A photometry study of the role of mesolimbic dopamine in learning

To demonstrate how our method can be used to answer scientific questions in photometry experiments, we reanalyzed data from a recent article proposing a new model for mesolimbic dopamine's role in learning (*Jeong et al., 2022*) (all analyses are on photometry data collected with dLight1.3b in the nucleus accumbens core). Note that this is a different study from the one described in *Figure 1*. We used this study for two reasons. First, the dataset exhibits many common characteristics of photometry data that can dilute effects, or even invalidate results if left unaccounted for. Second, the dataset contains data from multiple experiments, which allows us to illustrate how *FLMM* can be used to test hypotheses across a range of experimental designs. In this section, we discuss how *FLMM* handles those characteristics; in subsequent sections, we show how those hypotheses can be posed in our framework.

One of the most important characteristics of these data is that the dopamine (DA) measurements were collected in within-subject nested longitudinal experiments. For example, in the first behavioral task we discuss, mice were trained across multiple sessions. Each session involved the delivery of a sequence of 100 sucrose rewards. The authors analyzed average dopamine (AUC) during the 'reward period' time-window (aligned to the first lick after reward-delivery) as a trial (*Figure 3A*). Thus trials were nested in session, which were nested in animals (*Figure 3B*). Photometry experiments often exhibit this type of nested longitudinal structure, which can induce correlation patterns in the data within-animal, and obscure effects if not accounted for statistically. This structure can occur in both within-subjects and between-subjects designs, as illustrated in *Figure 3D*. For example, these data exhibited high within-animal correlations across sessions (*Figure 3E*), and across trials within-session (*Figure 3F*).

*FLMM* can model effect heterogeneity, as well as correlation patterns within- and across- trials, through the inclusion of functional random-effects. These allow one to estimate what is common across all animals, and what is unique to each. This is critical because photometry datasets often exhibit between-animal variability in covariate-signal associations. For example, in the task described above, the time between rewards, (`IRI`: 'inter-reward interval') varied between trials unpredictably (*Figure 3C*), and the authors reported that 'reward period' AUC was correlated with `IRI`. *Figure 3G* shows that the magnitude of this correlation varied considerably across animals and sessions. *FLMM* can model this variability in `IRI`–DA correlation magnitude through the inclusion of animal- and session-specific random effects. More broadly, by varying which covariates and random-effects are included in the model, *FLMM* can analyze data from experimental designs that include within- and

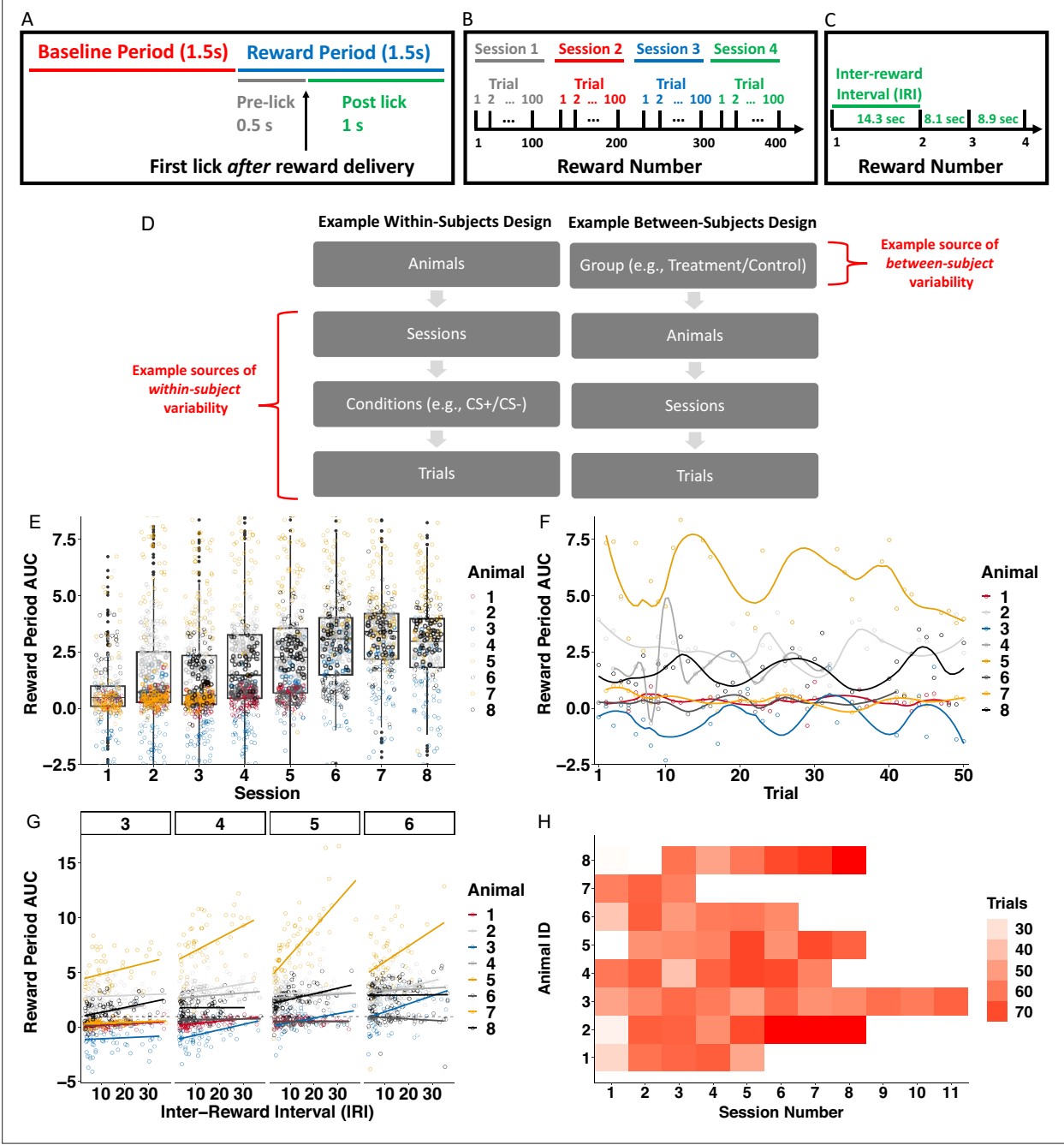

**Figure 3.** Nested longitudinal designs in photometry experiments can result in correlation patterns and missing data that dilute effects if not unaccounted for statistically. Descriptive statistics and figures pertain to data from *Jeong et al., 2022*, reanalyzed in section Using FLMM to test associations between signal and covariates throughout the trial. (**A**) Experiment trial time-windows used to construct photometry signal summary measures (Area Under the Curve, AUC). The reward was delivered at random times and signals were aligned to the first lick following reward delivery. Reward delivery may occur during the Baseline Period or Reward Period, depending on the lick time. (**B**) The Reward Number is defined as the cumulative number of rewards (interchangeably referred to as 'trials') pooled across sessions. Each session involved the delivery of 100 trials. (**C**) The time between two rewards (inter-reward interval or IRI) was a random draw from an exponential distribution (mean 14). (**D**) Examples of experimental designs that exhibit hierarchical nesting structure. Trials/sessions and conditions such as cue-type (e.g. CS+/CS-) contribute to variability within-animal. Between-subject variability can arise from, for example, experimental groups, photometry probe placement, or natural between-animal differences. (**E**) Reward Period AUC values are correlated *across sessions*. Each dot indicates the average reward period AUC value of one trial. *Between-session* correlation in AUC values can be seen *within-subject* since reward period AUC values are similar within-animal on adjacent sessions. *Between-session* correlation can be seen on average *across animals*: session boxplot medians are similar in adjacent sessions. (**F**) Temporal correlation within-subject on session 3, chosen because it is the only session common to all animals. Reward period AUC on each trial for any animal is similar on adjacent

*Figure 3 continued on next page*

Figure 3 continued

trials. (**G**) Lines show association (ordinary least square, OLS) between `IRI` and reward period AUC for each animal and session, revealing individual differences in association magnitude. The heterogeneity in line slopes highlights the need for random-effects to account for between-animal and between-session variability. (**H**) Number of sessions and trials per session (that meet inclusion criteria) included varies considerably between animals. For example, one animal's data was collected in sessions 1–11 while another's was collected in sessions 1–3.

between-subject contrasts. For example, one can implement functional versions of methods like correlations, or repeated measures ANOVA, and, more generally, model a wide range of dependence structures. Finally, *FLMM* can accommodate missing data and subjects with different sample sizes, which are often unavoidable characteristics of photometry experiments. In *Figure 3H* we show how this dataset included photometry recordings of behavioral sessions at various stages of training that differed across animals. If not accounted for statistically, this can obscure associations between the signal and covariates. For example, average reward period AUC levels were reported to increase across sessions. Thus animals with data collected only on early sessions may appear to have lower AUC levels than other animals, which can increase uncertainty in estimates of covariate effects.

## Using *FLMM* to test associations between signal and covariates throughout the trial

We first recreate an analysis of the experiment described in the previous section. *Jeong et al., 2022* reported, in section 'Tests 1 and 2', that the trial-level 'reward period' AUC was positively correlated with `IRI`. The authors fit separate Pearson correlations to data from each animal. To illustrate how to test this question in our framework, we show a recreation of their analysis in *Figure 4A, B*, and the *FLMM* analysis estimates of the `IRI`–DA association in *Figure 4C, D*. While *Jeong et al., 2022* assessed the `IRI`–AUC correlation in each animal separately, *FLMM* can test the `IRI`–DA association at each trial time-point and in all animals jointly, thereby improving our ability to detect effects. To implement a test most similar to the Pearson correlation, we fit an *FLMM* model with `IRI` as the (fixed-effect) covariate. Given the between-animal and between-session heterogeneity in signal profiles highlighted in *Figure 3*, we included nested random-effects to account for the variability across both subjects and sessions. The relationship is significantly positive throughout the time-window [–0.5, 1.5] sec.

These results corroborate the paper's findings, and provide finer-grained details inaccessible with standard analyses. For example, the temporal dynamics, revealed by *FLMM*, suggest that the neural signal associated with `IRI` may be composed of two distinct components. The first component rises rapidly starting at around –0.75 sec and decreases quickly after lick-onset (the first lick after reward-delivery). Since the association reaches its peak before lick-onset, the signal may reflect motivation, movement, or reward detection. The second component begins after lick-onset and rises and falls slower than the first component. The timing suggests it may track sucrose consumption. Importantly, the reward period AUC in the paper averages across these putative components. In contrast, *FLMM* is able to partially disentangle them through their distinct temporal dynamics, and offers insight into the role the components play in behavior.

*FLMM* can also identify results completely obscured by standard methods. We show this by recreating an analysis in which *Jeong et al., 2022* report that reward period AUC was positively correlated with that trial's `Reward Number`. *Figure 3B* illustrates the definition of `Reward Number`, the cumulative trials/rewards received across sessions. The finding that the `Reward Number`–AUC correlation was positive was controversial because it conflicted with the prediction of the Reward Prediction Error ('RPE') model, a prevailing hypothesis for the role of DA in reward learning. The authors argued that RPE predicts a negative `Reward Number`–DA association, while their model ('ANCCR') predicts a positive association. Using *FLMM* on the same data, we estimate that the Reward Number–DA association is in fact *negative* within-session. The discrepancy by findings arises because the analysis in *Jeong et al., 2022* pools trials *across sessions* in a manner that does not consider the session number that a reward was delivered on. *FLMM* finds the effect above by accounting for the nested session/trial task structure, and suggests why it occurs by estimating changes at each trial time-point.

The source of the conflicting results above is most clearly illustrated by fitting `Reward Number`–AUC correlations in each animal and session separately. In *Figure 5A* and *Appendix 4—figure 1* we show session-specific linear regression fits overlaid on the fits from the session-pooled analysis conducted

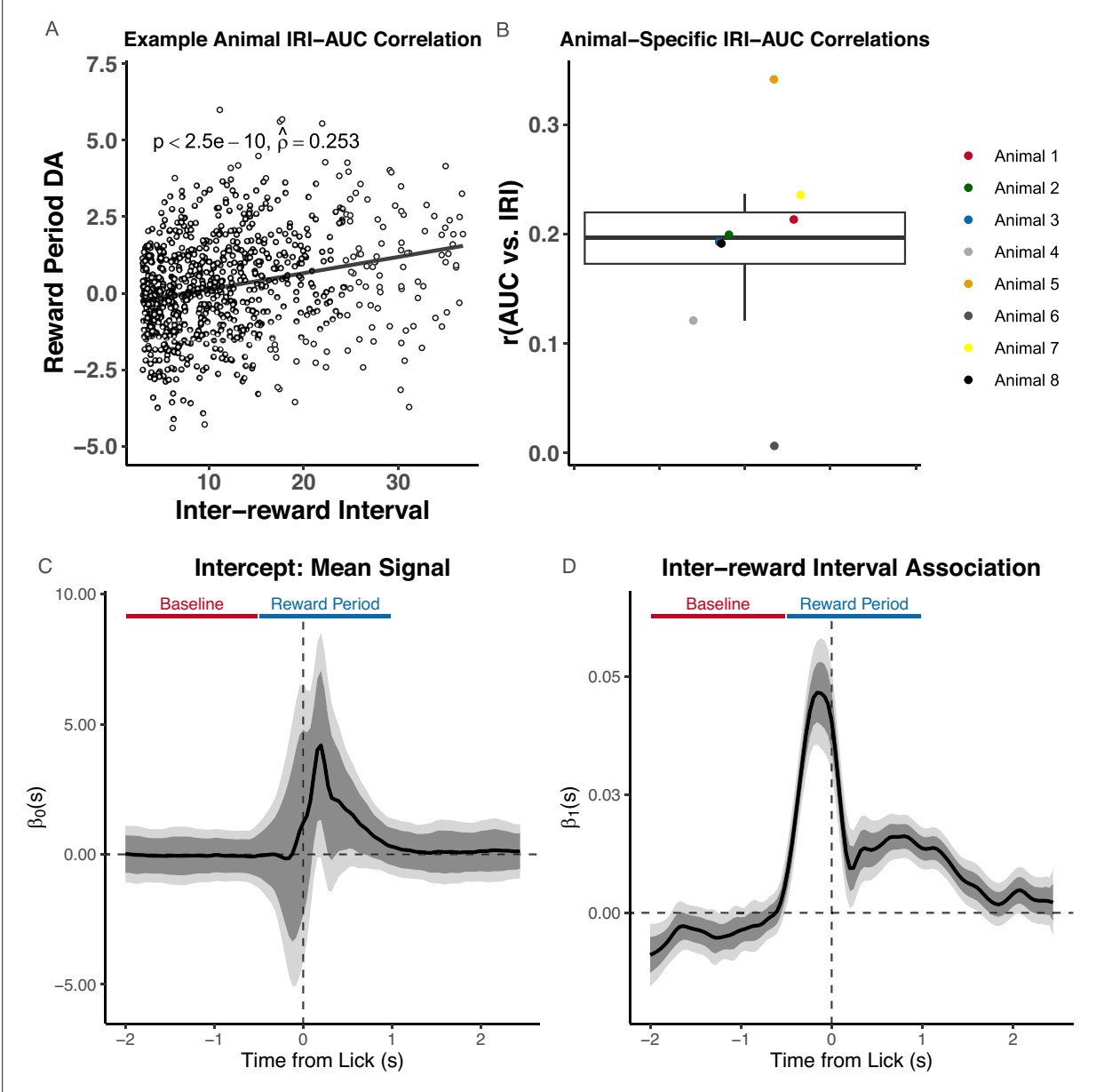

**Figure 4.** Functional Linear Mixed Models (*FLMM*) reveals distinct components obscured by summary measure analyses. (**A, B**) show a recreation of statistical analyses conducted by *Jeong et al., 2022* on the random inter-trial interval (IRI) reward delivery experiment, and (**C, D**) show our analyses. (**A**) Analysis of IRI–Area Under the Curve (AUC) correlation on all trials in an example animal, as presented in *Jeong et al., 2022*. (**B**) Recreation of boxplot summarizing IRI–AUC correlation coefficients from each animal. (**C,D**) Coefficient estimates from *FLMM* analysis of IRI–dopamine (DA) association: functional intercept estimate (**C**), and functional IRI slope (**D**). Although we do not use AUC in this analysis, we indicate the trial periods, 'Baseline' and 'Reward Period', that *Jeong et al., 2022* used to calculate the AUC. They quantified DA by a measure of normalized AUC of $\Delta F/F$ during a window ranging from 0.5 sec before to 1 sec after the first lick following reward delivery. All plots are aligned to this first lick after reward delivery. The IRI–DA association is statistically significantly positive in the time interval ~[–0.5, 1.75] sec.

by *Jeong et al., 2022*. We parameterized session-specific models to yield intercepts (highlighted as large black circles) with the interpretation 'the expected AUC value on the first trial of the session.' These intercepts tend to increase across sessions, ultimately resulting in a positive overall Reward Number–AUC correlation in the session-pooled analysis (see *Appendix 4—figure 1*). However, the majority of the within-session *slopes* are actually negative, indicating that the AUC decreases across trials *within-session*. The apparent disagreement between the positive Reward Number–AUC correlation estimated in the session-pooled analysis, and the negative Trial Number–AUC association

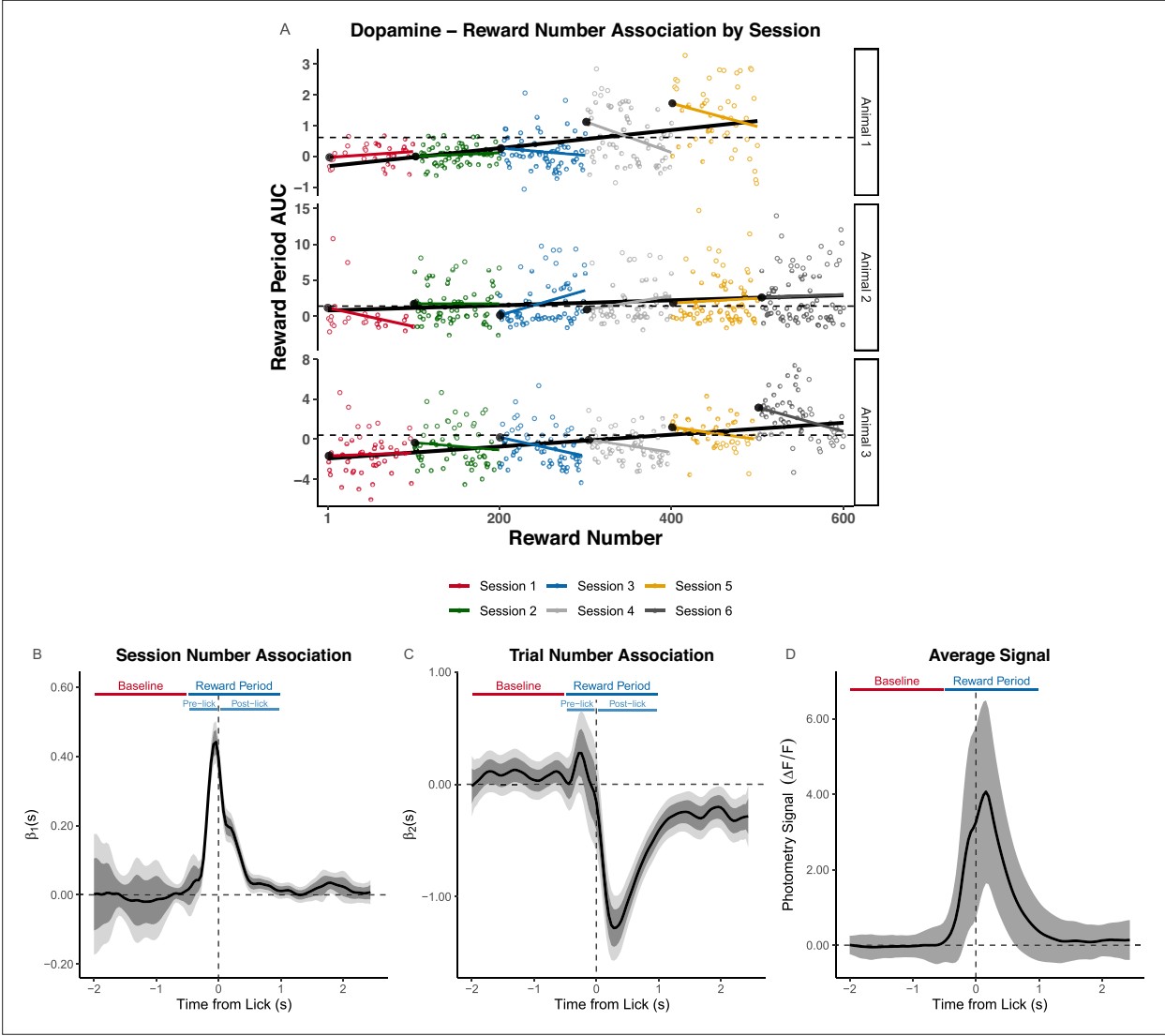

**Figure 5.** Functional Linear Mixed Models (*FLMM*) identifies within-session signal decreases obscured by standard analyses. (**A**) Visualization of the Simpson's paradox: the Area Under the Curve (AUC) decreases within-session, but increases across sessions. Plot shows Reward Number–AUC linear regressions fit to data pooled across sessions (black lines), or fit to each session separately (colored lines) in three example animals. Each colored dot is the AUC value for that animal on the corresponding session and trial. The black dots at the left of each color line indicate the intercept value of the session- and animal-specific linear regression model. Intercepts were parameterized to yield the interpretation as the 'expected AUC value on the first trial of the session for that animal.' The dotted lines indicate the animal-specific median of the intercepts (across sessions) and are included to visualize that the intercepts increase over sessions. (**B, C**) Coefficient estimates from FLMM analysis of the Reward Number–dopamine (DA) association that models Reward Number with Session Number and Trial Number (linear) effects to capture between-session and within-session effects, respectively. The plots are aligned to the first lick after reward-delivery. The Baseline and Reward Period show the time-windows used to construct AUCs in the summary measure analysis from *Jeong et al., 2022*. Pre-lick and post-lick time-windows indicate the portions of the Reward Period that occur before and after the lick, respectively. The Session Number effect is *jointly* significantly positive roughly in the interval [–0.25,0.5] sec, and peaks before lick-onset. This suggests DA increases across sessions during that interval. The Trial Number effect is briefly *pointwise* significantly positive around ~−0.3 sec and *jointly* significantly negative in the interval [0, 2.5] sec. This suggests DA decreases across trials within-session during the interval [0, 2.5] sec. (**D**) Average signal pooled across sessions and animals. Shaded region shows standard error of the mean.

identified in the within-session analysis, is an example of *Simpson's paradox*. This is caused by pooling data without taking into account its hierarchical structure.

To resolve this 'paradox,' we fit an *FLMM* that models the nested design by including *both* between-session and within-session fixed-effects, as well as nested random-effects. Estimating the Session Number and Trial Number functional coefficients reveals that *within-session* and *between-session* effects have distinct temporal dynamics. For example, *Figure 5D, E* shows that the *FLMM* estimates

that the mean signal decreases significantly *within-session* beginning immediately post-lick, but *increases across-sessions* for a brief interval around the lick. Importantly, the `Session Number`–DA association peaks and begins falling before the lick, while the `Trial Number`–DA correlation becomes significant only after the lick. Together these results suggest that DA increases across sessions in response to reward *anticipation*, but decreases within-session during reward *consumption*.

An alternative interpretation of the opposing within- and between-session effects is that DA rises within-session, but appears to decrease due to photobleaching. We assess this in **Appendix 4.3**. We repeat the above analyses on the signal aligned to reward-delivery, and analyze two other behavioral experiments, collected on the same animals. These show that, *within-session*, DA responses increase to reward-predictive cues, but decrease to reward-delivery (predicted and unpredicted). That we can detect both within-session increases and decreases to separate events (cue-onset and reward-delivery) occurring on the same trials, suggests that photobleaching is not hiding within-session DA increases. In personal communications, *Jeong et al., 2022* agree that while photobleaching cannot be definitively ruled out, a parsimonious interpretation of these findings is that there is a DA reduction to unpredicted rewards within-session, independent of photobleaching.

## Using *FLMM* to compare signal 'temporal dynamics' across conditions

We next describe an example comparing the signal 'temporal dynamics' across conditions, from section 'Tests 3–7' of *Jeong et al., 2022*. In this experiment, each trial consisted of a presentation of a 2 sec cue, followed by a reward 3 sec after the cue-onset. After many training sessions, the delay was lengthened to 9 sec (*Figure 6A–B*). On the first 9 sec delay session, the authors report that "dopaminergic cue responses [(Cue Period AUC)] showed no significant change," relative to trials from the last short-delay session. *Jeong et al., 2022* argue that their finding conflicts with what an RPE hypothesis of dopamine coding would anticipate (*Kobayashi and Schultz, 2008*). Like the authors, we compared the average signals between the last short-delay and the first long-delay sessions. However, we sought to test whether the signal 'dynamics' over trial time-points changed after lengthening the delay. To directly compare the difference in signal magnitudes between short- and long-delay sessions, at each time-point, we fit a *FLMM* model with a single binary covariate representing short/long-delays (similar to a functional paired t-test). The *FLMM* estimated mean DA was significantly lower in the latter parts of the initial cue-elicited DA response (~1–2.5 sec after cue-onset) on long- compared to short-delay trials (shown in *Figure 6H*). This effect may have been occluded because the AUC analyzed in *Jeong et al., 2022* was constructed by averaging signal values in a time window that contains opposing effects: the significant (relative) *reduction* in mean DA ~0.5–2.5 sec after cue-onset (marked as interval (2) in *Figure 6H*), identified by *FLMM*, is potentially diluted in the AUC by the (non-significant) *increase* in the cue response in the first ~0.5 sec (marked as interval (1) in *Figure 6H*). In our reanalyses we identified other experiments where effects were obscured by the use of AUCs averaging over opposite effects (see **Appendix 4.2**).

This example shows how *FLMM* can detect subtle effects that are difficult to identify by eye, and thus hard to construct appropriate summary measures for. For example, the delay-change effect is significant only during the falling-portion of the cue-elicited transient (0.5–2.5 sec after cue-onset). This small *relative* reduction (shown in *Figure 6H*) during a later portion of the cue-response is over-shadowed by the much larger *overall* DA response immediately following (0–0.5 sec) cue-onset (shown in *Figure 6G*). Finally, we show animal-specific functional random-effect estimates in *Figure 6E, F*. These provide intuition about how *FLMM* adjusts for between-animal differences in the 'temporal dynamics' of covariates effects.

We further analyze this dataset in **Appendix 2**, to compare *FLMM* with the approach applied in *Lee et al., 2019* of fitting *pointwise* LMMs (without any smoothing) and applying a Benjamini–Hochberg (BH) correction. Our hypothesis was that the *Lee et al., 2019* approach would yield substantially different analysis results, depending on the sampling rate of the signal data (since the number of tests being corrected for is determined by the sampling rate). The proportion of time-points at which effects are deemed statistically significant by *FLMM* joint 95% CIs is fairly stable across sampling rates. In contrast, that proportion is both inconsistent and often low (i.e. highly conservative) across sampling rates with the *Lee et al., 2019* approach. These results illustrate the advantages of modeling a trial signal as a function, and conducting estimation and inference in a manner that uses information across the entire trial.

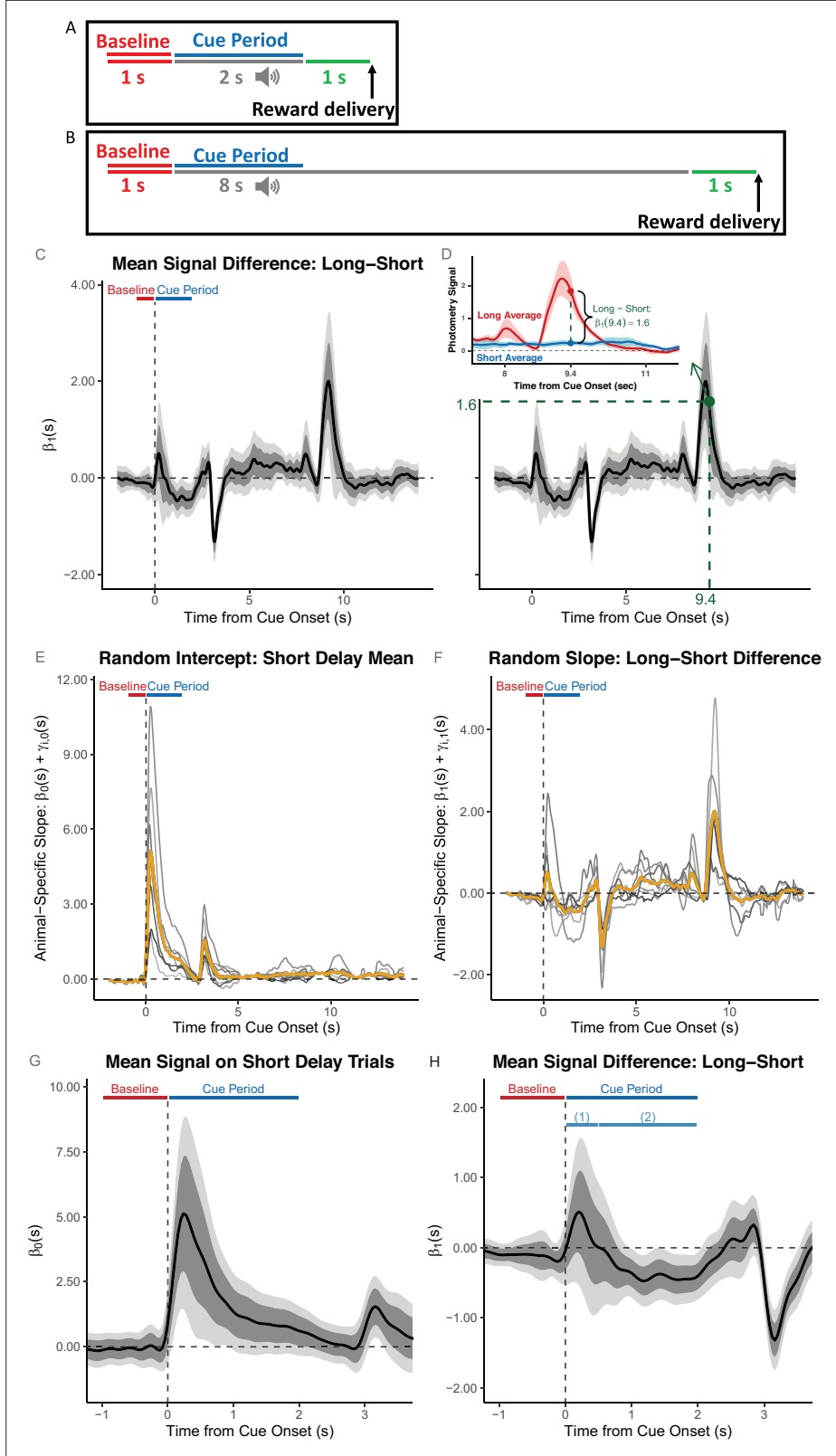

**Figure 6.** Functional Linear Mixed Models (*FLMM*) identifies significant temporal dynamics effects missed by summary measure analyses. The analysis of the Delay Length change experiment by *Jeong et al., 2022* used the following summary measure: the average `Cue Period AUC - Baseline AUC` (Area Under the Curves, AUCs in the windows [0,2] and [−1,0] sec, respectively, relative to cue onset). (**A**, **B**) Behavioral task design and Baseline/Cue

*Figure 6 continued on next page*

*Figure 6 continued*

Period are illustrated for short-delay (**A**) and long-delay (**B**) sessions. (**C-H**) These plots show coefficient estimates from *FLMM* re-analysis of the experiment. (**C**) The coefficient value at time-point $s$ on the plot is interpreted as the mean *change* in average dopamine (DA) signal at time-point $s$ between long- and short-delay trials (i.e. positive values indicate a larger signal on long-delay trials), aligned to cue onset. (**D**) Same Figure as in (**C**) but the inset shows the interpretation of an example time-point ($s = 9.4$): the difference in magnitude between the average traces (pooled across animals and trials) of long- and short- delay sessions. (**E, F**) Gold lines indicate the fixed-effect estimates and gray lines indicate animal-specific estimates (calculated as the sum of functional fixed-effect and random-effect estimates (Best Linear Unbiased Predictor)) for the random intercept, and random slope, respectively. (**G, H**) Fixed-effect coefficient estimates shown with expanded time axis. In (**H**), it is clear that long-delay trials exhibit average (relative) increases (sub-interval (1)) and decreases (sub-interval (2)) in the signal that would likely cancel out and dilute the effect, if analyzing with a summary measure (AUC) that averages the signal over the entire Cue Period.

## Simulation experiments

We conducted experiments to assess how *FLMM* performs on synthetic data based on the Delay Length experiment data from *Jeong et al., 2022* introduced in **Using FLMM to compare signal 'temporal dynamics' across conditions**. We simulated data in the small sample sizes typical in photometry experiments. We incorporated the key characteristics discussed earlier, namely trial-to-trial and session-to-session correlation, as well as animal-to-animal variability in photometry signal (i) magnitudes and (ii) levels of association with the covariates. We did this by treating the parameter estimates from a *FLMM* model, fit to the Delay Length dataset, as the 'true' parameter values. This ensured values fell within a realistic range. As the simulated data contained a single binary covariate, this allowed comparison with other statistical approaches discussed in the **Introduction**, namely the permutation method (*Jean-Richard-Dit-Bressel et al., 2020*), a paired samples t-test, and a (non-functional) LMM. The latter two only work with summary measures, so we applied them to the reward period AUC summary measure used in *Jeong et al., 2022*. *Figure 7A* shows average traces of the real data that the simulations are based on. *Figure 7B* presents simulated trials from one 'animal,' and *Figure 7C* presents simulated session averages from seven 'animals'.

The quantitative measures for comparison are *pointwise* and *joint* 95% CI coverage, and statistical power. *Figure 7D* shows that the *FLMM* achieves *joint* 95% CI coverage at roughly the nominal level for all sample sizes tested. The permutation method, *Perm*, (*Jean-Richard-Dit-Bressel et al., 2020*) provides only *pointwise* CIs and thus yields low *joint* coverage. We next analyzed the cue period to allow comparison between the average performance of *FLMM*, evaluated at time-points in that time-window, with the performance of standard methods applied to the AUC summary measure. *FLMM* achieved roughly the nominal pointwise coverage (*Figure 7E*), while *Perm* does not, likely because the cluster-bootstrap often performs poorly in small sample settings (*Ju, 2015*). *FLMM* yields substantially better statistical power than the other methods (*Figure 7F*). The LMM and t-test exhibit lower power, likely because effects are diluted by analyzing summary measures. In **Appendix 5.2**, we present additional simulation results that demonstrate that the performance of summary measure methods is highly sensitive to minor changes in the length of the time-interval that the AUC summarizes. These results show how *FLMM* improves *pointwise* coverage and statistical power compared to standard methods and provides reasonable *joint* coverage.

## Discussion

### Technical contribution

We introduced a Functional Linear Mixed Modeling framework for photometry data analysis that: (1) enables hypothesis testing at every trial time-point; (2) accounts for individual differences and trial-to-trial variability; and (3) allows modeling of longitudinal experimental designs. We extend the *FLMM* framework in *Cui et al., 2022* to enable parameter estimation with *pointwise* and *joint* 95% CIs for general random-effect specifications. Previous work provided *FLMM* implementations for either *joint* CIs for simple random-effects models (*Cui et al., 2022*; *Sergazinov et al., 2023*; *Crainiceanu et al., 2024*), or *pointwise* CIs for nested models (*Scheipl et al., 2016*). However, they do not allow computation of *joint* 95% CIs in the presence of general random-effects specifications, which is helpful for

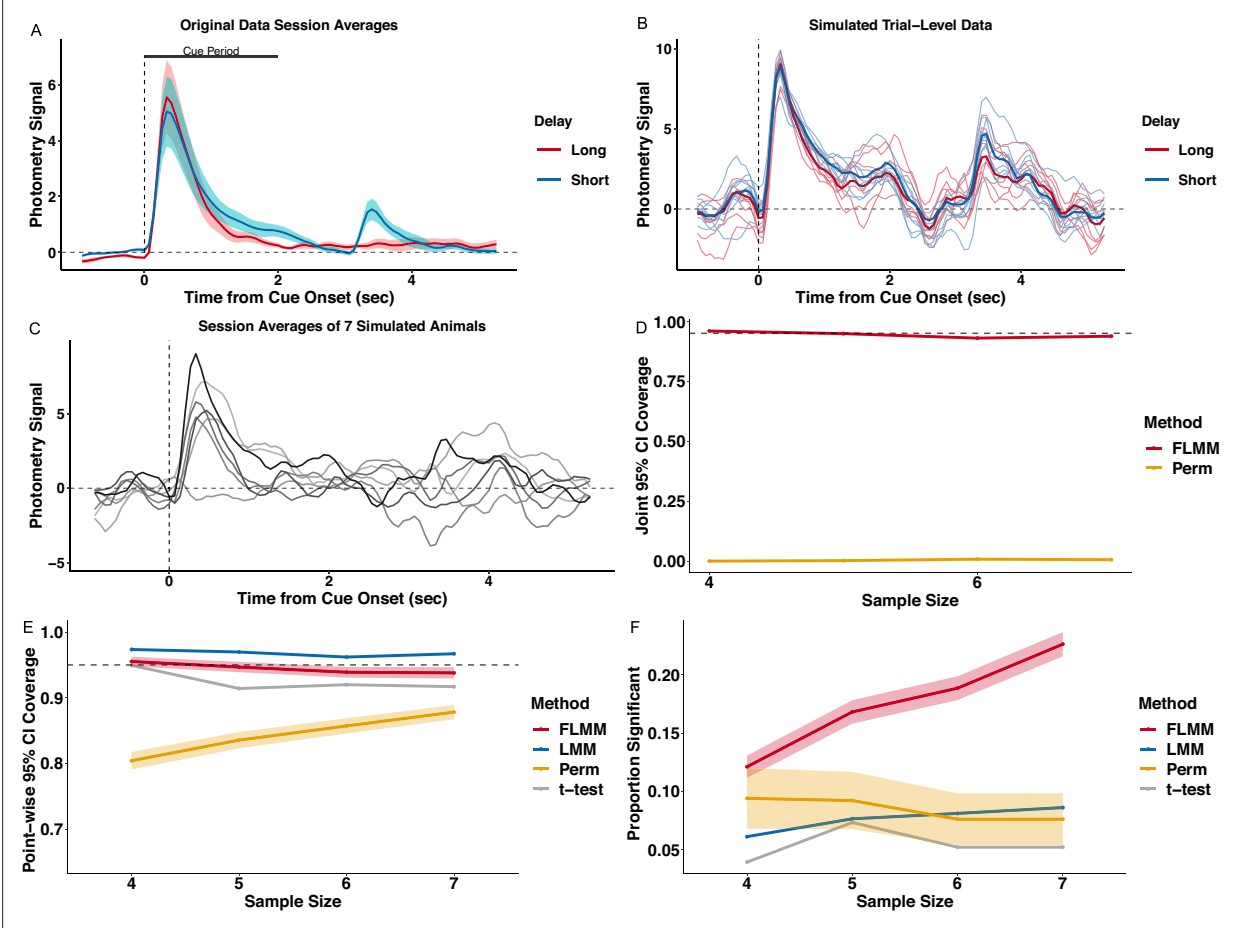

**Figure 7.** Realistic simulation experiments show that Functional Linear Mixed Models (*FLMM)* exhibits desirable statistical properties for photometry analyses. The simulation produces synthetic photometry data similar to *Jeong et al., 2022*, with the same sources of variability across trials and animals. (**A**) Lines show average traces from the original photometry data. The traces are averaged across trials and animals from the last short- and the first long-delay session. The bar shows the cue period analyzed in the paper and in our experiments. (**B**) Each thin line is the signal from a single simulated trial from the same 'animal'; bold lines show the average trace for each trial type. (**C**) Each line is the trial-averaged trace from one session for seven simulated 'animals.' (**D**) *FLMM* exhibits approximately correct *joint* 95% confidence interval (CI) coverage. *Perm* does not provide *joint* CIs and thus its *joint* coverage is low, as expected. (**E**) *FLMM* exhibits approximately correct *pointwise* 95% CI coverage. (**F**) *FLMM* improves statistical power during the cue period compared to standard methods at each sample size tested. Power is calculated for *Perm* based on the full consecutive threshold criteria. For figures (**E**) and (**F**) the linear mixed models (LMM) and t-test were fit on the cue period Area Under the Curve (AUC) and thus each replicate yields one indicator of CI inclusion and statistical significance. We represent the corresponding proportions with a line plot. For other methods, estimates are provided at each time-point. We therefore average performance across the cue period and then summarize the variability of these replicate-specific averages with a 95% confidence band.

modeling many characteristics of photometry datasets. When conducting the analyses in this paper, we found that nested random-effects specifications, enabled by our method, substantially improved model fits compared to models without nested random-effects. We describe *FLMM* further in the Methods section **Functional mixed models** and derive our methodology in **Appendix 3**.

## Evaluation of *FLMM* in real and synthetic datasets

We demonstrated the capabilities of *FLMM* by reanalyzing datasets from a recently published study *Jeong et al., 2022*. First, we showed that the main scientific questions asked by the authors can easily be answered by specifying covariates of interest, fitting an *FLMM* model, and referring to the coefficient estimate plots for results. Furthermore, we showed that *FLMM* allows direct comparison – and testing – of differences in signal time-courses between conditions, something that is difficult with standard methods relying on summary measure tests. Finally, we showed that *FLMM* can reveal significant effects that were tested by the authors, but obscured by summary measure analyses.

The specific effects identified with *FLMM* bear on one of the key questions in *Jeong et al., 2022*, namely which of two competing models – RPE and ANCCR – better describe changes in DA responses to unpredicted rewards over trials. For example, in personal communications, *Jeong et al., 2022* still argue that the observed DA increases *across-sessions* in the random `IRI` task may be inconsistent with RPE. However, as a result of our reanalyses, they now argue that the *within-session* decreases, identified by *FLMM*, conflict with the predictions of the initial ANCCR model presented in *Jeong et al., 2022*. The predictive/explanatory power of RPE and ANCCR as models for measured DA responses depend, however, on their respective model specifications (e.g. RPE reward function) and is thus beyond the scope of the present work. The difficulty in comparing the two models is further compounded by a number of alternative explanations for our findings that cannot be definitively ruled out. For example, satiation or habituation may contribute to within-session DA decreases. An initial exploration suggests that this may not be the sole cause, as the temporal dynamics of within-session reductions vary between-session (*Appendix 4—figure 2*), despite equal levels of food-restriction. The between-session variability in dynamics may, however, reflect learning-related changes in consummatory behavior, and thus may not rule out satiation as an explanation for the within-session DA reductions. Moreover, the authors suggested that the reward-delivery mechanism produces stimuli that could act as a reward-predictive cue. Some of the DA changes may, therefore, reflect learning about this cue. Finally, between-session DA increases could be due to elevations in indicator expression levels over training. This was, however, explored by *Jeong et al., 2022* and they reported that transients unrelated to reward did not increase in magnitude across sessions (see Figure S8F-H in *Jeong et al., 2022*). As the authors note, if reward-related DA increases across sessions were caused by higher indicator levels, one would expect non-reward-related DA activity to also rise.

In personal correspondence, the authors of *Jeong et al., 2022* offered the following possible explanations for the within-session DA decreases within the ANCCR framework. Specifically, while the ANCCR simulations described in *Jeong et al., 2022* did not consider the session a trial occurred on, there may be session boundary-effects in animal learning. For instance, animals may estimate the rate of reward to be lower at the start of a session, due to a degraded memory from prior sessions, or to generalization between the high reward rate within the head-fixed context and the low rate outside it. Then the systematic increase in animals' estimate of reward rate during the session would result in a corresponding decrease in the learning rate of the reward–reward predecessor representation (*Burke et al., 2023*). If so, the magnitude of the reward-reward predecessor representation increase at reward-delivery would reduce across sessions, and result in a negative `Session Number`–DA correlation.

To further demonstrate the utility of *FLMM* in answering common neuroscience questions, we reanalyze data in which the authors tested whether, across several sessions of Pavlovian learning, DA activity 'backpropagates' from reward delivery (3 sec after cue-onset) to the presentation of reward-predictive cues (see **Appendix 4.4**). We argue that *FLMM* is well-suited to assess questions like these: the method tests how the signal *timing* evolves both within- and across-sessions by providing a null hypothesis test (i.e. to assess statistical significance) at each time-point. Our results align with those reported by *Jeong et al., 2022*. In contrast, *Amo et al., 2022* reported fiber photometry data that they argue supports the backpropagation hypothesis, despite being collected from a similar brain region and behavioral task. If researchers apply *FLMM* to test the backpropagation hypothesis in the future, we recommend fitting models that account for between-animal heterogeneity. Our initial exploration suggested that if the backpropagation hypothesis holds, the timing of when that 'back-propagating hump' begins to transition may differ between animals. In order to thoroughly test that hypothesis, it may, therefore, be necessary to specify an *FLMM* mean model that explicitly accounts for this potential variability in timing.

Adjudicating between ANCCR and RPE is beyond the scope of this paper, but our findings suggest that future experiments may be needed to test the explanations proposed in light of our results. Even in the Delay Length change experiment, in which *FLMM* identified significant changes obscured by AUC analyses, it is not clear to us whether proposed versions of RPE and ANCCR would predict the specific change in dynamics observed. As such, modeling trial-level mesolimbic DA as a vector (e.g. see *Wärnberg and Kumar, 2023*, *Lee et al., 2024*) or function may improve theories for its role in reward learning. By estimating trial-level temporal dynamics, we hope that *FLMM* will make it possible to further refine these theories.

For further evaluation of *FLMM*, we reanalyzed a second study (***Coddington et al., 2023***, see **Appendix 6**) that measures calcium dynamics, showing the applicability of the method to different photometry sensors. There, we demonstrate additional capabilities to: (1) model how signal 'dynamics' early in training predict behavior later in training; and (2) estimate covariate interactions across both trial time-points and sessions. Finally, we carried out simulation experiments on realistic synthetic data to verify that, in small sample settings, the *FLMM* framework yields *joint* and *pointwise* CIs that achieve approximately 95% coverage, and improve statistical power versus the alternative approaches introduced. As *FLMM* commits type I errors at roughly nominal levels and outperforms methods that analyze trial-averaged data, we verify that *FLMM* is not overly susceptible to trial-level noise or spurious correlations.

## Benefits of applying *FLMM* to neural data

Beyond the technical capabilities of our method, we believe there are a number of ways in which it can enhance scientific practice. First, *FLMM* may reduce biases arising from summary statistic analyses. Photometry measures aggregate activity from a collection of neurons, which may contain sub-populations with different functions. By coarsening the temporal resolution, summary measures may obscure the heterogeneity of the target neural population, and bias the study of systems towards sub-populations that exhibit larger signals. Through analyzing each time-point, *FLMM* may help disentangle separate signal components arising from different neural sub-populations. Second, we found that *FLMM* made it easier to identify model misspecification, as estimate plots often provided obvious visual indications (e.g. CI magnitude varying enormously across sub-intervals of the trial). We recommend fitting a few models and selecting one based on fit criteria (e.g. AIC/BIC), without considering statistical significance. Finally, we hope *FLMM* will improve reproducibility by removing the need for some signal pre-processing steps. Any data pre-processing will have down-stream consequences on the analysis results and interpretations from most statistical methods. In addition to possibly drowning out effects, some pre-processing steps (e.g. signal smoothing) can reduce reproducibility if the approaches differ across labs. *FLMM* can be used to eliminate some common pre-processing steps by including them as part of the model, or make their effect on the analysis results clearer, as we summarize in the next paragraph.

*FLMM* can help model signal components unrelated to the scientific question of interest, and provides a systematic framework to quantify the additional uncertainty from those modeling choices. For example, analysts sometimes normalize data with trial-specific baselines because longitudinal experiments can induce correlation patterns across trials that standard techniques (e.g. repeated measures ANOVA) may not adequately account for. Even without many standard data pre-processing steps, *FLMM* provides smooth estimation results across trial time-points (the 'functional domain'), has the ability to adjust for between-trial and -animal heterogeneity, and provides a valid statistical inference approach that quantifies the resulting uncertainty. For instance, session-to-session variability in signal magnitudes or dynamics (e.g. a decreasing baseline within-session from bleaching or satiation) could be accounted for, at least in part, through the inclusion of trial-level fixed or random effects. Similarly, signal heterogeneity due to subject characteristics (e.g. sex, CS+ cue identity) could be incorporated into a model through the inclusion of animal-specific random effects. Inclusion of these effects would then influence the width of the confidence intervals. By expressing one's 'beliefs' in an *FLMM* model specification, one can compare models (e.g. with AIC). Even the level of smoothing in *FLMM* is largely selected as a function of the data, and is accounted for directly in the equations used to construct confidence intervals. This stands in contrast to 'trying to clean up the data' with a pre-processing step that may have an unknown impact on the final statistical inferences.

*FLMM* is applicable in a wide range of experimental settings. We provide an analysis guide and open source package, `fastFMM` (our package is available on `CRAN` and also at [https://github.com/gloewing/fastFMM](https://github.com/gloewing/fastFMM), which has links to our analysis guide and examples of using the package in both `R` and `python`), that is the first, to our knowledge, to provide *joint* CIs for functional mixed models with nested random-effects. For demonstration, we selected a dataset collected on a behavioral paradigm, photometry sensor, and signaling pathway that are well characterized, making it easier to evaluate the framework. Although selecting summary measures and time-windows to quantify is relatively easy here, this may not be the case when collecting data under new conditions. In such cases, *FLMM* can characterize the association between neural activity and covariates without strong assumptions about

how to summarize signals. Importantly, the package is fast and maintains a low memory footprint even for complex models and relatively large datasets (see **Functional mixed models package** for an example). The *FLMM* framework may also be applicable to techniques like electrophysiology and calcium imaging. For example, our package can fit functional generalized LMMs with a count distribution (e.g. Poisson). Additionally, our method can be extended to model time-varying covariates. This would enable one to estimate how the level of association between signals, simultaneously recorded from different brain regions, fluctuates across trial time-points. This would also enable modeling of trials that differ in length due to, for example, variable behavioral response times (e.g. latency-to-press). In this paper, we specified *FLMM* models with linear covariate–signal relationships *at a fixed trial time-point* across trials/sessions, to compare the *FLMM* analogue of the analyses conducted in *Jeong et al., 2022*. However, our package allows the modeling of covariate–signal relationships with non-linear functions of covariates, using splines or other basis functions. One must consider, however, the tradeoff between flexibility and interpretability when specifying potentially complex models, especially since *FLMM* is designed for statistical inference.

To conclude, we believe research on statistical methods for fiber photometry is necessary, given the widespread use of the technique and the variability of analysis procedures across labs. We hope that our proposed *FLMM* framework and software further enable investigators to extract the rich information contained in photometry signals.

## Methods

The framework we introduced is based on *FLMM*, which is a combination of linear mixed modeling and functional regression. In this section, we introduce both of these prior to describing the *FLMM* approach. We then describe the analyses and modeling methods we used to reanalyze data from *Jeong et al., 2022* as well as details of the simulation scheme. We describe our strategy for the analysis of data after pre-processing steps such as the calculation of $\Delta F/F$.

### Linear mixed models

Linear mixed modeling (LMM, also known as multilevel or hierarchical modeling) is a method for testing the association between covariates (i.e. 'predictor' or 'independent' variables) and an outcome ('dependent') variable for repeated measures data (or multilevel/longitudinal). We provide a brief description of mixed models below, but see *Aarts et al., 2014*; *Yu et al., 2022*; *Magezi, 2015*; *Barr et al., 2013*; *Barr, 2013*; *Baayen et al., 2008* for detailed descriptions of their use in neuroscience and psychology.

LMM enables hypothesis testing in repeated measures designs, and can account for and characterize individual differences through the inclusion of 'random-effects.' Conceptually, random-effects serve as animal-specific regression coefficients (one may include more than just animal-specific random-effects (e.g. session-level random-effects)). This allows the relationship between a covariate of interest (e.g. locomotion) and the outcome (e.g. lever pressing) to vary between-animals in the model, while still pooling data from all animals to estimate parameters. For example, they can be used to model between-animal differences in photometry signal magnitudes on baseline conditions (e.g. on a control condition). This can allow the model to capture how variables of interest (e.g. behavior) are associated with the signal after 'adjusting' for the fact that some animals may, on average, exhibit lower signal magnitudes on all conditions. While LMMs may not feel intuitive initially, they actually share connections to many familiar hypothesis testing procedures. Just as ANOVAs, t-tests, and correlations can be cast as special cases of linear models, repeated measures versions of these tests (e.g. paired sample t-tests, repeated measures ANOVAs, and MANOVAs) have similar connections to linear *mixed* models. We recommend sections 1.2–1.3 of *Fitzmaurice, 2008* for more explanation of these connections. This reference also includes descriptions of the capacity of LMM (and thus FLMM) to accommodate: (1) unbalanced designs, (2) varying sample sizes across individuals, and (3) more complicated correlation structures than methods like repeated measures ANOVA.

In order to provide better intuition about LMM in the photometry context, we illustrate the role of random-effects in *Figure 8*. In panel A, we plot data from five animals (photometry data from *Coddington et al., 2023*, with behavioral data simulated for illustrative purposes). On multiple trials, to examine the association between the signal and behavior on a test session. Averaging the signal

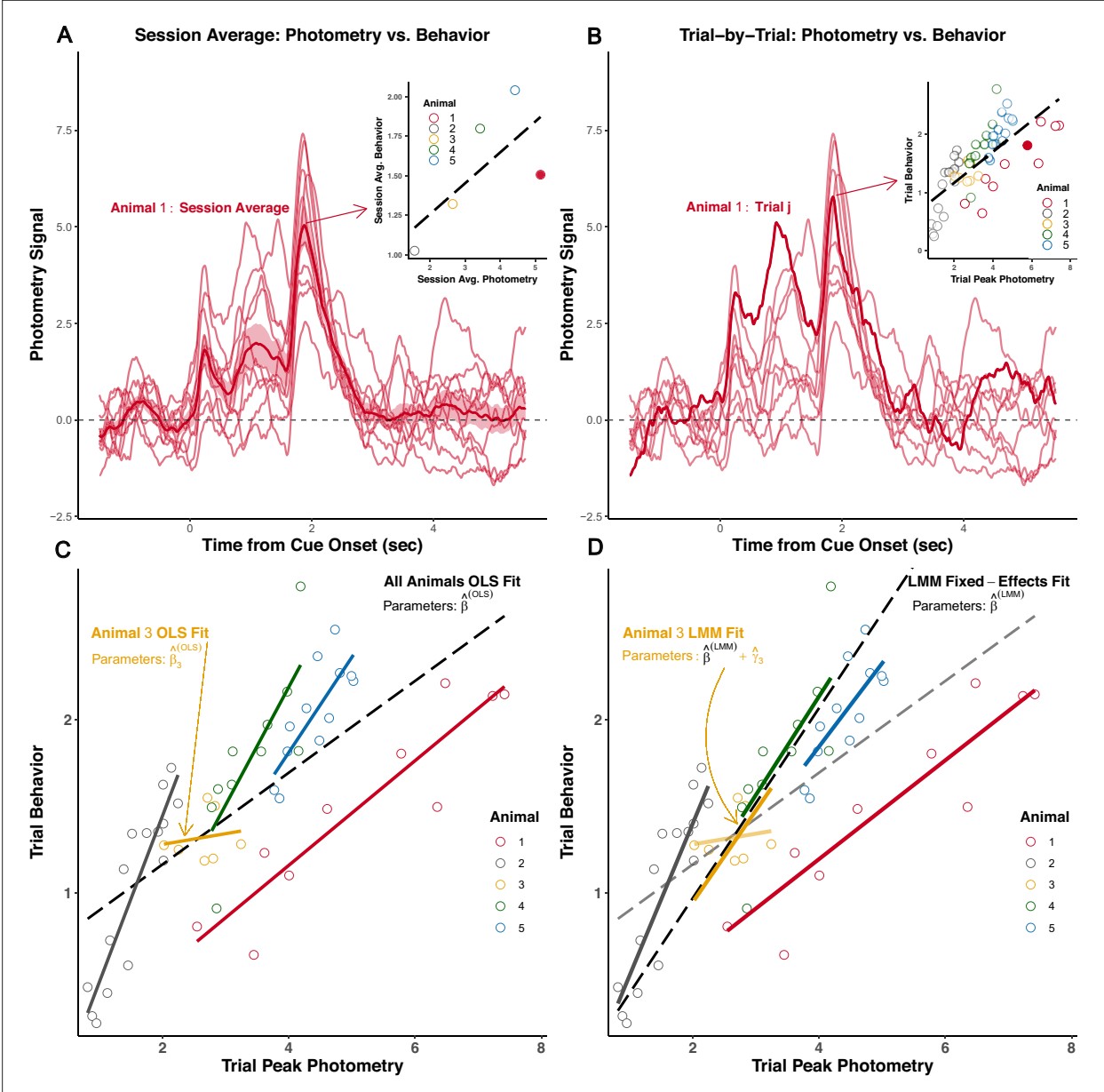

**Figure 8.** Example repeated measures data from a single test session. (**A**) *Session-average approach*: photometry signals from all trials for Animal 1 are averaged across trials and summarized (peak amplitude). The session-average summary is then plotted (inset) against the session-average behavior for each of five animals. Example animal's session average is the filled circle. (**B**) *Trial-level approach*: the trial-level signal summary measure (peak amplitude) is pooled across animals and correlated with trial-level behavior. An example signal from one trial for Animal 1 is highlighted in the trace plot. That example trial is represented as the filled circle in the inset. Each dot in the inset is one trial from one animal; dot color indicates the animal ID. (**C**) Inset from (**B**) is magnified. Linear regressions (OLS: Ordinary least squares) fit separately to each animal's trial-level data. A global regression fit to the trial-level data pooled across animals is displayed as the dotted black line. (**D**) Linear mixed modeling (LMM) strikes a balance between one model common to all animals and fitting many animal-specific models. The 'global' fixed-effects fit (from $\widehat{\boldsymbol{\beta}}^{(LMM)}$) and the fits including the subject-level random-effect estimates (Best Linear Unbiased Predictor) are displayed. Subject-specific fit for animal $i$ is calculated from: $\widehat{\boldsymbol{\beta}}^{(LMM)} + \widehat{\boldsymbol{\gamma}}_i^{(LMM)}$. Note the fixed-effects, $\widehat{\boldsymbol{\beta}}^{(LMM)}$, and random-effects, $\widehat{\boldsymbol{\gamma}}_i^{(LMM)}$, are estimated in the same model.

across trials and correlating its peak amplitude with average session behavior results in only five observations, shown in the inset. In *Figure 8B*, we show the alternative approach that correlates a summary of *each trial* (e.g. peak amplitude) with the trial-level behavioral variable. The linear regression is fit to many more observations than in the session average strategy, but the standard regression hypothesis testing procedure may now be invalid, as we violated the independence assumption by pooling

repeated observations within animals. Moreover, it ignores that the number of samples (trials) differ between animals. We magnify the inset from *Figure 8B* in *Figure 8C* to emphasize how within-subject correlation arises, at least partially, from animal-to-animal variability. Indeed, the linear regressions (Ordinary Least Squares or 'OLS') fit to each animal separately differs from the fit obtained if we first pool all animals' trial-level data ('All Animals OLS Fit'). This is expected since (1) each animal may have a different 'true' regression line (i.e. individual differences), and (2) the estimated lines may differ 'by chance' alone due to sampling error. An LMM allows us to pool across animals and trials but still account for individual differences through 'random-effects,' thereby striking a *balance* between (i) the line fit to the trial-level data pooled across animals, and (ii) the animal-level fits. In *Figure 8D* we show the fit estimated from the LMM (determined by $\widehat{\beta}^{(LMM)}$), as well as the subject-specific fits that include the random-effects (defined by the *sum*: $\widehat{\beta}^{(LMM)} + \widehat{\gamma}_i$ for subject $i$). In *Figure 8C* the *All Animals OLS Fit* is 'pulled' towards Animal 1's data because we failed to account for the fact that more trials were recorded from Animal 1. In *Figure 8D*, the LMM fit accounts for differing sample sizes across animals and is no longer unduly influenced by Animal 1: it is more similar to the majority of the other animals' subject-specific LMM fits than the OLS fit. Moreover, we see that Animal 3 in *Figure 8C* has an OLS fit that differed considerably from all other animals. The LMM subject-specific fit is, however, 'shrunk' towards the population fit, determined by $\widehat{\beta}^{(LMM)}$.

The LMM used to estimate the fits presented in *Figure 8D* is

$$Y_{i,j} = \beta_0^{(LMM)} + \gamma_{0,i} + X_{i,j}\left(\beta_1^{(LMM)} + \gamma_{1,i}\right) + \epsilon_{i,j} \tag{2}$$

where $Y_{i,j}$ and $X_{i,j}$ are the outcome variable (e.g. trial behavior) and covariate (e.g. photometry signal peak amplitude), respectively, associated with subject $i$ on trial $j$. For short hand, we concatenate the 'fixed' effects, random-effects and covariates into vectors, $\boldsymbol{\beta}^{(LMM)} = [\beta_0 \; \beta_1]^T$, $\boldsymbol{\gamma}_i = [\gamma_{0,i} \; \gamma_{1,i}]^T$ and $\mathbf{X}_{i,j} = [1 \; X_{i,j}]^T$. $\boldsymbol{\beta}^{(LMM)}$ is the population regression coefficient estimate that is *shared* across all subjects, and $\boldsymbol{\gamma}_i$ is subject $i$'s random-effect, its deviation from the shared population slope $\boldsymbol{\beta}^{(LMM)}$. Thus the slope that determines the fit specific to subject $i$ is $\boldsymbol{\beta}^{(LMM)} + \boldsymbol{\gamma}_i$. That is, the LMM estimates: (i) what is common among all animals; and (ii) what is unique to each animal. This imbues LMM (and its generalizations) with a flexibility that enables one to account for individual differences and to model correlated data arising from a wide range of complex experimental designs. LMM still requires, however, one to analyze summary measures (e.g. AUCs), and thus their application in photometry analyses leads to discarding of within-trial signal information across time-points the downsides of summary measure analyses are well-characterized in the statistics community (e.g. see section 1.2 of *Fitzmaurice, 2008*).

## Functional linear regression

We now describe function-on-scalar regression, a functional data analysis method where the outcome variable is a *function*, rather than a scalar value. We treat the photometry time series from one trial as a 'function', which allows us to test the association between any trial-level experimental variables of interest (e.g. cue type, latency-to-press) and the photometry signal value *at each time-point in the trial*. For example, if photometry data for two groups is collected on one trial, one could fit a separate t-test at *every* time-point in the trial. The t-test at time $s$ then assesses the mean difference between the signal magnitude of group 1 at time $s$ and the signal magnitude of group 2 at time $s$. Plotting the test statistics vs. time yields a *curve* representing how differences between groups change over time. This provides the intuition behind the functional version of a *t-test*, and is a special case of the *functional linear model*. This functional version of the t-test differs from the common approach of comparing the groups' signals using a single summary measure (e.g. peak amplitude in the trial).

In essence, the functional linear model (here, we describe the *function-on-scalar* linear model but use the functional linear model as short hand) is a linear regression with a functional coefficient $\beta(s)$ across time-points $s$:

$$Y_i(s) = \mathbf{X}_i\boldsymbol{\beta}(s) + \epsilon_i(s) \tag{3}$$

where $Y_i(s) \in \mathbb{R}$ and $\epsilon_i(s) \in \mathbb{R}$ are the photometry value and error term of animal $i$ at time-point $s$ in the signal. The $p \times 1$ vector, $\boldsymbol{\beta}(s)$, is the regression coefficient at time-point $s$, applied to the covariates, $\mathbf{X}_i \in \mathbb{R}^p$. This functional linear regression framework allows for testing the effects of (scalar) variables

at every time-point, adjusting for multiple covariates (continuous or discrete) and their interactions. Functional regression thus exploits *within-trial* signal information. However, it cannot be directly used to analyze multiple trials per animal, since it assumes independence across observations of the signal *between-trials* (see Chapter 5 of *Crainiceanu et al., 2024*).

## Functional mixed models

Our framework is based on Functional Linear Mixed Models (*FLMM*), which is a form of function-on-scalar regression that combines the benefits of linear mixed effects modeling and functional linear regression. We provided an informal presentation of the approach in the Results section **Functional Linear Mixed Models (FLMM)**, with the goal of enabling researchers with a wide range of statistical backgrounds to understand its application and results. Here, we provide a high-level overview of the methodology, with further technical details provided in **Appendix 3**. *FLMM* models each trial's signal as a function that varies smoothly across trial time-points (i.e. along the 'functional domain'). It is thus a type of non-linear modeling technique over the functional domain, since we do not assume a linear model (straight line). *FLMM* and other functional data analysis methods model data as functions, when there is a natural ordering (e.g. time-series data are ordered by time, imaging data are ordered by x-y coordinates), and are assumed to vary smoothly along the functional domain (e.g. one assumes values of a photometry signal at close time-points in a trial have similar values). Functional data analysis approaches exploit this smoothness and natural ordering to capture additional information during estimation and inference. Our *FLMM* approach is based on an estimation procedure that was first proposed in *Cui et al., 2022*. We selected this method because (1) it allows for calculation of *joint* 95% CIs, (2) it is readily scalable for the dataset sizes and random-effect specifications needed for neuroscience experiments, and (3) it uses common syntax for model specification from the well-known `lme4` package.

The construction of *joint* CIs in the context of functional data analysis is an important research question; see *Cui et al., 2022* and references therein. Each *point* at which the *pointwise* 95% CI does not contain 0 indicates that the coefficient is *statistically significantly* different from 0 at that point. Compared with *pointwise* CIs, *joint* CIs take into account the autocorrelation of signal values across trial time-points (the functional domain). Therefore, instead of interpreting results at a specific time-point, *joint* CIs enable *joint* interpretations at multiple locations along the functional domain. This aligns with interpreting covariate effects on the photometry signals across time-intervals (e.g. a cue period) as opposed to at a single trial time-point. Previous methodological work has provided functional mixed model implementations for either *joint* 95% CIs for simple random-effects models (*Cui et al., 2022*), or *pointwise* 95% CIs for nested models (*Scheipl et al., 2016*), but to our knowledge, do not provide explicit formulas or software for computing *joint* 95% CIs in the presence of general random-effects specifications. We found nested random-effects specifications substantially improved model fits in the neuroscience experiments we reanalyzed, and helped model sophisticated behavioral designs (see **Appendix 3.3** for more details). We, therefore, derived an extension of the estimator for the covariance of the random-effects (*Cui et al., 2022*), based on a Method of Moments approach described by *Greven et al., 2010*.

Mixed effects modeling (and thus *FLMM*) requires one to specify the random-effects in the modeling process. In essence, this amounts to selecting which covariate effects (i.e. regression coefficients) might vary across, for example, animals or sessions (see **Appendix 3.3** for more details on specification and interpretation of random-effects). While this may seem unfamiliar at first, in practice, most experimental designs (and the appropriate model formulations) fall under a few categories that can be modeled with extensions of familiar methods (e.g. *FLMM* analogues of repeated measures ANOVA, correlations, etc.). Our method is fully implemented by our package and uses the same syntax as the common mixed effects package `lme4`, thereby allowing one to easily fit the models and plot the results. Thus the rich literature on linear mixed model selection and model syntax can be used for *FLMM*, as many of the same principles apply (e.g. see reviews of LMM and applications in neuroscience and psychology in *Aarts et al., 2014*; *Yu et al., 2022*; *Magezi, 2015*; *Barr et al., 2013*; *Barr, 2013*; *Baayen et al., 2008*). Users may improve their analyses by evaluating a few candidate models and selecting the one with the best model fit criteria. To that effect, AIC, BIC, and cAIC (*Säfken et al., 2018*) are already provided automatically in our software implementation. We provide the code to fit models with our package below and we include all code used for this paper (e.g. data pre-processing,

model-fitting) on the GitHub page: https://github.com/gloewing/photometry_FLMM (copy archived at *Loewinger, 2024*). Finally, since fiber photometry sums photon counts over many neurons (the activity of which may be modeled as only weakly dependent given model parameters), we appeal to the central limit theorem to motivate the adoption of a Gaussian-likelihood for the conditional distribution of $Y_{i,j}(s) \mid \mathbf{X}_{i,j}, \gamma_{i,j}(s)$. However, when photon counts are low, our package can be used to fit functional generalized LMMs with a count distribution like a Poisson (see *Cui et al., 2022*).

## Figure 1 methods and analyses

*Figure 1* was generated from data (*Coddington et al., 2023*) available at *Dudman, 2023*. All signals are measurements of calcium dynamics in mesolimbic dopamine cells (DAT-Cre::ai32 transgenic mice were injected with a Cre-dependent jRCaMP1b virus across the ventral midbrain) recorded from fibers in the nucleus accumbens. In this experiment, head-fixed mice were exposed to a 0.5 sec stimulus, followed by a reward 1 sec after cue-offset. Signals are aligned to cue-onset.

*Figure 1A* was generated from trial-averaging Lick+ trials from session 8 for control animals that received no optogenetic stimulation. To ensure the animals were sufficiently trained we selected session 8, because it is the latest session in which all control animals have recorded data available in this dataset. *Figure 1B* was generated from trial-averaging Lick+ and Lick- trials and then plotting the pointwise difference. *Figure 1C* was generated arbitrarily from the first control animal in the dataset (i.e. animal ID 1), which we selected without inspecting other animals' data to avoid biases. The data plotted are a subset of Lick+ trials (a sequence that starts from the first and ends on the last trial and takes every 10th trial) on the final training session for that animal (session 11). We selected this session to ensure the animal was as well trained as possible. *Figure 1D* shows session averages on Lick+ trials from sessions 8–11.

We made *Figure 1E and F* for explanatory purposes, and the specific choices we made in creating these figures are not meant to reflect any specific photometry analysis procedures. We nevertheless include a description of how we generated them for completeness. *Figure 1E* was the trial-averaged trace from session 8 for Lick- trials for Animal 1. *Figure 1F* was the trial-averaged trace from session 8 for Lick- and Lick+ trials separately for Animal 1. The inset contains an individual point for each animal from session 8. A given point was calculated by trial-averaging Lick+ and Lick- for each animal separately, and then finding the maximum difference.

## Data reanalysis methods

Our primary experimental results in the **Results** section come from a reanalysis of the data from *Jeong et al., 2022*. Here, we provide a high-level description of the methods used, with additional details in **Appendix 4**. We describe the models we implemented and where possible, we explain the authors' hypotheses, data pre-processing, and analysis procedures. We often center all covariates to be mean zero to allow an interpretation of the intercept as the mean signal for a trial when all covariates are at their average values. For each of the following analyses, we fit a collection of models and selected the final one based on which exhibited the best AIC and BIC. Finally, the R scripts used to replicate the analyses or derive information from data (e.g. extracting consummatory lick bouts) can be found on the GitHub page: https://github.com/gloewing/photometry_FLMM (copy archived at *Loewinger, 2024*).

### Notation

We denote $\mathbb{E}[Y_{i,j,l}(s) \mid \mathbf{X}_{i,j,l}, \mathbf{Z}_{i,j,l}, \gamma_{i,j,l}(s)]$ as the *FLMM* mean model of $Y_{i,j,l}(s) \in \mathbb{R}$, the photometry signal at trial time-point $s$, for animal $i$, on trial $j$ of session $l$. We use the general notation $\mathbf{X}_{i,j,l}$ as the $p \times 1$ vector of covariates for the fixed-effects and $\mathbf{Z}_{i,j,l}$ as the $q \times 1$ vector of covariates for the random-effects. The full form of the covariates that fully specify the columns of the associated design matrices can be quite complicated (e.g. when we include animal-specific random-effects for each session). On the right-hand side of the mean model, we specify the covariates by name when possible (e.g. $\text{IRI}_{i,j,l}$ as the inter-reward interval for subject $i$ on trial $j$ on session $l$), with the understanding that the vectors $\mathbf{X}_{i,j,l}$ and $\mathbf{Z}_{i,j,l}$ contain these covariates where applicable (e.g. this covariate may be included in $\mathbf{X}_{i,j,l}$, $\mathbf{Z}_{i,j,l}$, or both). As mentioned in the main text, the entry in these vectors can change from trial-to-trial but, unlike the outcome, the entries do not change within-trial and, for that reason, we do *not* include the notation $(s) : X_{i,j,l}(s)$. Finally, in some models the covariates change from session-to-session but

not across trials *within* a session. We indicate how the variables change (e.g. trial-to-trial vs. session-to-session but not trial-to-trial) through outcome, covariate, and random-effect subscripts (e.g. $\mathbf{Y}_{i,l}(s)$, $\texttt{Delay}_{i,l}$, $\gamma_{i,l}(s)$). Finally, we denote $\gamma_{i,j,l}(s)$ as the $q \times 1$ vector of random effects for subject $i$ on trial $j$ of session $l$ at time-point $s$.

## Reanalysis methods: Using *FLMM* to test associations between signal and covariates throughout the trial

We analyzed the same set of sessions that were analyzed by the authors (*Jeong et al., 2022*) in the section 'Tests 1 and 2 (unpredicted rewards)'. As noted previously, the set of sessions analyzed by the authors differed across animals, and the number of trials per session differed between sessions. The sessions analyzed and the characteristics of these data are presented in *Figure 3*. All analyses are on data collected with the photometry dopamine dLight1.3b sensor (AAVDJ-CAG-dLight1.3b virus). The virus was injected and the optical fiber was implanted in the nucleus accumbens core. The authors quantified dopamine (DA), measured with normalized AUC of $\Delta F/F$ during a window of 0.5 sec before to 1 sec after the first lick that occurred after reward-delivery. In 'Data Analysis: Experiment 1' of the Supplement (p.3), the authors describe that "The [Area Under the Curve (AUC)] during 1.5 sec time window before reward period was subtracted from AUC during reward period to normalize base-line activity. To test dynamics of dopamine response to reward, Pearson's correlation was calculated between dopamine response and Reward Number, or dopamine response and inter-reward interval (IRI) from the previous reward." In all of our analyses in this section, we tested *linear effects* of all covariates of interest, given that the authors conducted Pearson correlations in their tests and we wanted to implement the most similar model in our framework (e.g. see Fig. S8 in *Jeong et al., 2022*). We followed the methods for excluding trials based on behavioral criteria, and for extracting consummatory lick bouts described in 'Data Analysis: Experiment 1' of the Supplement (p.3–4) of the original manuscript. We describe this in more detail in **Appendix 4.2**.

### IRI model

We tested the association between the DA response to sucrose reward and $\texttt{IRI}$ ($\texttt{IRI}_{i,j,l}$), as shown in *Figure 4*. We first compared models that included, for example, subject- and session-specific random intercepts, random slopes for $\texttt{Trial Number}$, random slopes for $\texttt{Lick Latency}$ (the time between reward-delivery and first lick), and random slopes for $\texttt{IRI}$. We considered this collection of models to account for trial-to-trial variability in learning (e.g. $\texttt{Lick Latency}$ grows faster with learning), within-session effects such as satiation (e.g. $\texttt{Trial Number}$), between-session heterogeneity in $\texttt{IRI}$ slopes, and baseline signal dynamics (e.g. the functional random-intercept). The best model included subject-specific (indexed by $i$) and session-specific (indexed by $l$) random functional intercepts and random functional slopes for $\texttt{Lick Latency}$ ($\texttt{Lick}_{i,j,l}$):

$$\mathbb{E}\left[Y_{i,j,l}(s) \,\middle|\, \mathbf{X}_{i,j,l}, \mathbf{Z}_{i,j,l}, \gamma_{i,l}(s)\right] = \beta_0(s) + \beta_1(s)\texttt{IRI}_{i,j,l} + \gamma_{0,i}(s) +$$

$$\gamma_{1,i,l}(s) + \texttt{Lick}_{i,j,l}\left[\gamma_{2,i}(s) + \gamma_{3,i,l}(s)\right].$$

This can be fit with our package using the code:

```
model_fit = fui(photometry ~ IRI + (lick_latency | id/session),
                data = photometry_data,
                subj_ID = "id")
```

This models $\gamma_{i,l}(s) \overset{\text{iid}}{\sim} \mathcal{N}\left(\mathbf{0}, \Sigma_\gamma(s)\right)$, where $\gamma_{i,l}(s) = \left[\gamma_{0,i}(s)\ \gamma_{1,i,l}(s)\ \gamma_{2,i}(s)\ \gamma_{3,i,l}(s)\right]^T$.

### Reward Number model

Similarly we tested the association between the DA response to sucrose reward and $\texttt{Reward Number}$ by modeling it with $\texttt{Session Number}$ ($\texttt{SN}_{i,l}$), and $\texttt{Trial Number}$ (indexed by $j$ but written out as $\texttt{TN}_{i,j,l}$ when used as a covariate), as shown in *Figure 5*. Our final model included subject-specific (indexed by $i$) and session-specific (indexed by $l$) random functional intercepts and random functional slopes for $\texttt{Lick Latency}$ ($\texttt{Lick}_{i,j,l}$):

$$\mathbb{E}\left[Y_{i,j,l}(s)\,\big|\,\mathbf{X}_{i,j,l},\mathbf{Z}_{i,j,l},\gamma_{i,l}(s)\right] = \beta_0(s) + \beta_1(s)\mathtt{SN}_{i,l} + \beta_2(s)\mathtt{TN}_{i,j,l} + \gamma_{0,i}(s) +$$

$$\gamma_{1,i,l}(s) + \mathtt{Lick}_{i,j,l}\left[\gamma_{2,i}(s) + \gamma_{3,i,l}(s)\right].$$

We fit the above model with the code:

```
model_fit = fui(photometry ~ trial + session + (lick_latency | id/session),
                data = photometry_data,
                subj_ID = "id")
```

This models $\boldsymbol{\gamma}_{i,l}(s) \overset{\text{iid}}{\sim} \mathcal{N}\left(\mathbf{0}, \Sigma_\gamma(s)\right)$, where $\boldsymbol{\gamma}_{i,l}(s) = \left[\gamma_{0,i}(s)\ \gamma_{1,i,l}(s)\ \gamma_{2,i}(s)\ \gamma_{3,i,l}(s)\right]^T$.

## Reanalysis methods: Using *FLMM* to compare signal 'temporal dynamics' across conditions

We analyzed the data described in the section 'Experiment 3: Tests 4–5' (*Jeong et al., 2022*). The authors noted that "[W]hen a learned delay between cue onset and reward (3 seconds) is extended permanently to a new, longer delay (9 seconds), [Reward Prediction Error] predicts that as animals learn the longer delay…there will be a concomitant reduction in the dopamine cue response due to temporal discounting (46). By contrast, [their model] predicts little to no change in the dopamine cue response as the structure of the task is largely unchanged…Experimentally, [they] observed that although animals learned the new delay rapidly, dopaminergic cue responses showed no significant change" (*Jeong et al., 2022*).

Their analysis of DA cue-response was based on a baseline subtracted AUC (AUC [0,2 sec] - [–1,0 sec] AUC relative to cue onset). They write in their supplement (p. 4) that "Experiments 2–5: [T]o analyze learning-dependent dynamics of dopamine response, AUC of ΔF/F during 2 [sec] from CS +onset was normalized by AUC during baseline period."

Our final model included a random slope for the long-delay (9 sec) indicator ($\mathtt{Delay}_{i,l}$), where $\mathtt{Delay}_{i,l} = 1$ indicates long-delay trials:

$$\mathbb{E}\left[Y_{i,j,l}(s)\,\big|\,\mathbf{X}_{i,l},\mathbf{Z}_{i,l},\gamma_i(s)\right] = \beta_0(s) + \gamma_{0,i}(s) + \mathtt{Delay}_{i,l}\left[\beta_1(s) + \gamma_{1,i}(s)\right].$$

We analyzed all trials on the last session of the short-delay (2 sec) and all trials on the first session of the long-delay (8 sec). Because the experimental design included the delay length switch *between* sessions, the covariate $\mathtt{Delay}_{i,l}$ does not include a trial index $j$ (i.e. all trials on session $l$ have the same delay length). This model assumes $\boldsymbol{\gamma}_i(s) \overset{\text{iid}}{\sim} \mathcal{N}\left(\mathbf{0}, \Sigma_\gamma(s)\right)$, where $\boldsymbol{\gamma}_i(s) = [\gamma_{0,i}(s)\ \gamma_{1,i}(s)]^T$. This can be fit with the code:

```
model_fit = fui(photometry ~ delay + (delay | id), data = photometry_data)
```

### Functional mixed models package

We provide the fastFMM package that implements our proposed methods. The package can be downloaded on `CRAN`, or on the GitHub page: https://github.com/gloewing/fastFMM. This GitHub also includes our analysis guide, and examples of calling the package from python. We also recommend `lme4`'s thorough package resources, since our package is based on the `lme4` software and model syntax (e.g. see Table 2 in *Bates, 2014*, Table 1 in *Barr et al., 2013* and *Bates, 2010*). **Appendix 3.4** contains more information about functional data analysis modeling approaches.

Our `fastFMM` package scales to the dataset sizes and model specifications common in photometry. The majority of the analyses presented in the **Results** included fairly simple functional fixed and random effect model specifications because we were implementing the *FLMM* versions of the summary measure analyses presented in *Jeong et al., 2022*. However, we fit the following *FLMM* to demonstrate the scalability of our method with more complex model specifications:

$$\mathbb{E}\left[Y_{i,j,l}(s)\,\big|\,\mathbf{X}_{i,j,l},\mathbf{Z}_{i,j,l},\gamma_i(s)\right] = \beta_0(s) + \gamma_{0,i}(s) + \mathtt{SN}_{i,l}\left[\beta_1(s) + \gamma_{1,i}(s)\right] +$$

$$\mathtt{TN}_{i,j,l}\left[\beta_2(s) + \gamma_{2,i}(s)\right] + \mathtt{IRI}_{i,j,l}\left[\beta_3(s) + \gamma_{3,i}(s)\right] +$$

$$\mathtt{Lick}_{i,j,l}\left[\beta_4(s) + \gamma_{4,i}(s)\right] + \mathtt{TL}_{i,j,l}\left[\beta_5(s) + \gamma_{5,i}(s)\right].$$

We use the same notation as the `Reward Number` model in **Reanalysis methods: Using *FLMM* to test associations between signal and covariates throughout the trial**, with the additional variable $\mathtt{TL}_{i,j,l}$ denoting the `Total Licks` on trial $j$ of session $l$ for animal $i$. In a dataset with over 3200 total trials (pooled across animals), this model took ~1.2 min to fit on a MacBook Pro with an Apple M1 Max chip with 64 GB of RAM. Model fitting had a low memory footprint. This can be fit with the code:

```
model_fit = fui(photometry ~ session + trial +iri + lick_time + licks +
                  (session + trial + iri + lick_time + licks | id),
                parallel = TRUE,
                data = photometry_data)
```

## Simulation experiments

For the experiments reported in **Using *FLMM* to compare signal 'temporal dynamics' across conditions**, we generated synthetic photometry signals based on the Delay Length data and model described in **Reanalysis methods: Using *FLMM* to compare signal 'temporal dynamics' across conditions**. We simulated the photometry signals using the functional linear mixed effects model.

$$Y_{i,j,l}(s) = \beta_0(s) + \gamma_{0,i}(s) + \mathtt{Delay}_{i,j,l}\left[\beta_1(s) + \gamma_{1,i}(s)\right] + \epsilon_{i,j,l}(s)$$

where $\beta_0(s)$ is the (functional) intercept at time-point $s$ on all trials, $\gamma_{0,i}(s)$ is the random (functional) intercept of subject $i$ at time-point $s$ on all trials. The (functional) coefficient $\beta_1(s)$ represents the average change in the mean signal at trial time-point $s$ on long-delay trials (compared to short-delay trials). $\epsilon_{i,j,l}(s)$ is the error term and $\gamma_{k,i}(s) \perp\!\!\!\perp \epsilon_{i,j,l}(s)$ for all $i \in \{1, 2, ..., n\}$, $j \in \{1, 2, ..., J\}$, $k \in \{0, 1\}$, $l \in \{1, 2\}$ and $s \in \{1, 2, ..., S\}$. All other notation is described in **Notation** and **Reanalysis methods: Using *FLMM* to compare signal 'temporal dynamics' across conditions**.

For each simulation replicate, we drew $\gamma_{k,i} \stackrel{iid}{\sim} \mathcal{N}_T(\mathbf{0}, \widehat{\Sigma}_k^\gamma)$ where $\widehat{\Sigma}_k^\gamma$ is the $S \times S$ covariance matrix of random-effect $k$ across trial time-points, estimated in the analysis described in **Reanalysis Methods: Using FLMM to compare signal 'temporal dynamics' across conditions**. For each $s$, we set $\beta(s) = \widehat{\beta}(s)$, where $\widehat{\beta}(s)$ was the estimated coefficient vector, at $s$, from the model described in **Reanalysis methods: Using *FLMM* to compare signal 'temporal dynamics' across conditions**. We drew $\epsilon_{i,j,l} \stackrel{iid}{\sim} \mathcal{N}_T(\mathbf{0}, \Sigma_i^\epsilon)$ where $\epsilon_{i,j,l}$ represents the $S \times 1$ vector of errors across trial time-points. The errors were drawn independently *across* different trials and subjects. To induce additional correlation across time-points within a trial, we set $\Sigma_i^\epsilon = 5 * \widehat{\Sigma}_m^\epsilon$, where $\widehat{\Sigma}_m^\epsilon$ is the estimated covariance matrix of the model residuals fit to the observed photometry data of animal $m$. For each *simulated* subject, $i$, the index $m$ was drawn uniformly from the observed animal indices $\{1, ..., 7\}$ without replacement when the simulated sample size $n \leq 7$. Otherwise, we set the first 7 to the indices $\{1, ..., 7\}$, and the remaining $n - 7$ were drawn without replacement from the indices $\{1, ..., 7\}$. This induced some correlation between signals from different subjects when $n > 7$, which had a small but predictably deleterious effect on method performances.

For comparisons in the cue period window, we assessed the average performance of *FLMM* evaluated at time-points in that window, against the performance of standard methods that analyzed a summary measure of the window (cue period average-signal, which we denote as AUC) as the outcome variable. The LMM was applied to trial-level AUCs, the paired t-test was applied to subject- and condition-level AUCs. The performances of the *Perm* and *FLMM* were quantified according to the average pointwise performance during the cue period. We present the full consecutive threshold criteria of *Perm* in **Figure 7** and show the performance of the 1/2 consecutive threshold criteria in **Appendix 5.2** for completeness. We implemented the consecutive threshold criteria method, adapted to our simulated sampling rate of 15 Hz.

## Acknowledgements

This research was supported by the Intramural Research Program of the National Institute of Mental Health (NIMH), project ZIC-MH002968 and the National Institute on Alcohol Abuse and Alcoholism (NIAAA), project ZIAAA000416. This study utilized the high-performance computational capabilities of the Biowulf Linux cluster at the National Institutes of Health, Bethesda, MD (http://biowulf.nih.gov). First, we would like to thank the corresponding authors of 'Mesolimbic dopamine release conveys

causal associations,' Drs. Vijay Namboodiri and Huijeong Jeong, for sharing their data, helping us conduct analyses, and interpret the results. This work would not have been possible without their generosity, commitment to open science, and scientific rigor. We also thank Dr. Luke Coddington, the corresponding author of and 'Mesolimbic dopamine adapts the rate of learning from action' for generously sharing his data, and being so willing to engage in detailed and open discussions of our analyses. We thank Al Xin for their contributions to the fastFMM package. We would also like to thank Drs. Kauê Costa, Filipe Rodrigues, Guohong Cui, Guillermo Esber, Matthew Gardner, Charles Zheng, Sofia Beas, Adrina Kocharian, Geoff Schoenbaum, Hugo Tejeda, Mario Penzo, Dave Lovinger's Lab, Joe Cheer's lab, and the NIMH Machine Learning Core for helpful feedback on our manuscript.

## Additional information

### Funding

| Funder | Grant reference number | Author |
| --- | --- | --- |
| National Institute of Mental Health | ZIC-MH002968 | Gabriel Loewinger Francisco Pereira |
| National Institute on Alcohol Abuse and Alcoholism | ZIAAA000416 | David Lovinger |

The funders had no role in study design, data collection and interpretation, or the decision to submit the work for publication.

### Author contributions

Gabriel Loewinger, Conceptualization, Resources, Software, Formal analysis, Visualization, Methodology, Writing - original draft, Writing - review and editing; Erjia Cui, Software, Writing - review and editing; David Lovinger, Validation, Writing - review and editing; Francisco Pereira, Conceptualization, Supervision, Visualization, Writing - original draft, Writing - review and editing

### Author ORCIDs

Gabriel Loewinger ⓘ https://orcid.org/0000-0002-0755-8520

Reviewer #1 (Public review): https://doi.org/10.7554/eLife.95802.3.sa1
Reviewer #2 (Public review): https://doi.org/10.7554/eLife.95802.3.sa2
Reviewer #3 (Public review): https://doi.org/10.7554/eLife.95802.3.sa3
Author response https://doi.org/10.7554/eLife.95802.3.sa4

## Additional files

### Supplementary files

MDAR checklist

### Data availability

The current manuscript is a computational study reanalyzing existing public data, so no data have been generated for this manuscript. Modeling code is available on Github and CRAN.

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

## Appendix 1

### Photometry citations

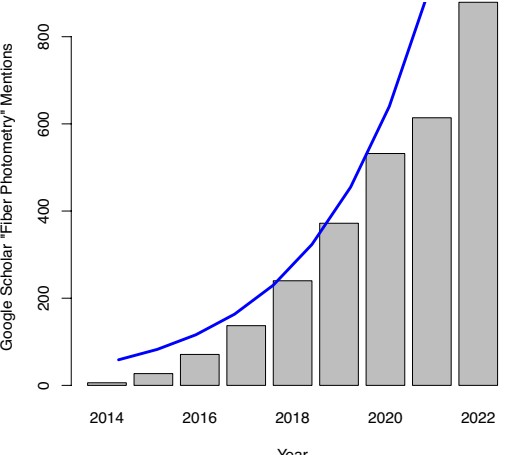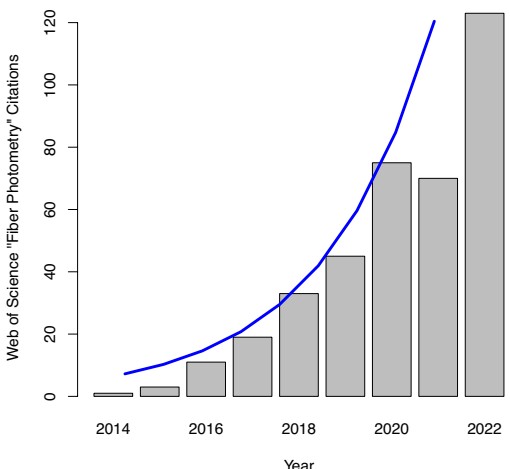

**Appendix 1—figure 1.** Photometry citations by year. [Left] Google scholar mentions of the string 'Fiber Photometry' by year. There were 549 mentions between January 1, 2023 and June 22, 2023. [Right] Web of Science citations of papers that include the string 'Fiber Photometry' by year. There were 50 citations between January 1, 2023 and June 22, 2023. Blue lines indicate fitted values from an exponential fit to the data: $\texttt{Citations}_i = \alpha * \exp[\beta * \texttt{Year}_i]$, where $\alpha$ and $\beta$ were estimated with the `nls` package in `R`. The 1500 references to photometry described in the main text refers to Google Scholar mentions in the 12 months prior to June 2023.

## Appendix 2

### BH correction

The goal of this appendix is to illustrate the advantages of *FLMM* compared to fitting LMMs to each trial time-point separately, and then applying a multiple-comparisons correction (as proposed in *Lee et al., 2019*). We specifically show that, in contrast to the approach applied in that work, *FLMM* yields far less conservative and more stable inference across different sub-sampling rates. We analyzed the Delay Length experiment (shown in *Figure 6*) data sub-sampled at a range of sampling rates (data were sub-sampled at evenly spaced intervals). We fit either a collection of separate LMMs followed by a Benjamini–Hochberg (BH) correction, or *FLMM* with statistical significance determined from both Pointwise and Joint 95% CIs. To avoid introducing more notation, we show the *FLMM* model, with the understanding that the functional notation can be interpreted pointwise for the approach applied in *Lee et al., 2019*:

$$\mathbb{E}\left[Y_{i,j,l}(s) \,\middle|\, \mathbf{X}_{i,l}\, \mathbf{Z}_{i,l}\, \gamma_i(s)\right] = \beta_0(s) + \gamma_{0,i}(s) + \texttt{Delay}_{i,l}\left[\beta_1(s) + \gamma_{1,i}(s)\right].$$

As shown in *Appendix 2—tables 1 and 2* the proportion of time-points, $s$, at which $\widehat{\beta}_0(s)$ is statistically significant with *FLMM* Joint CIs is fairly stable across sampling rates. In contrast, the percentage is highly inconsistent with the BH approach. For example, the BH approach identifies ~30% of time-points as significant at a sampling rate of 20 Hz, less than 1% at 25 Hz, ~30% at 30 Hz and then drops again to ~1% at 40 Hz. As the sampling rate grows towards 125 Hz (the sampling rate of the online dataset), the number of statistical comparisons grows, and the proportion of points that are significant drops to below 1%.

For the $\widehat{\beta}_1(s)$, the BH approach identifies no time-points that achieve statistical significance, even for time-points when the animal is consuming the reward. In contrast, the Pointwise and Joint *FLMM* CIs identify a relatively stable proportion of time-points that are statistically significant. The $\widehat{\beta}_1(s)$ is arguably a far more important point of comparison between these two statistical approaches, as the scientific question that motivated this analysis focuses on $\widehat{\beta}_1(s)$, rather than on $\widehat{\beta}_0(s)$.

Qualitatively speaking, the coefficient estimates, the widths of the 95% CIs, and the time intervals of statistical significance appear stable across the sampling rates, emphasizing how *FLMM* yields consistent inference and is not overly sensitive to the sub-sampling rate. We note that a multiple comparisons correction may yield more stable results if one first smooths regression coefficient point and variance estimates. To the best of our understanding, such smoothing was not conducted by *Lee et al., 2019*. Moreover, such a strategy would essentially be a functional mixed model using a multiple comparisons correction, instead of a joint CI.

**Appendix 2—table 1.** Functional intercept $\widehat{\beta}_0(s)$.
Percentage of time-points in which the coefficient estimates $\widehat{\beta}_0\left(s\right)$ are statistically significant. We compare the Benjamini–Hochberg (BH) correction applied to pointwise linear mixed models (LMM) models, the Functional Linear Mixed Models (*FLMM*) pointwise 95% CIs (Pointwise), and the *FLMM* joint 95% CIs (Joint). The proportion of points that are significant with the Benjamini–Hochberg (BH) approach jump around between 20-40Hz and then dramatically decrease as the sampling rate increases. In contrast, the *FLMM* Pointwise and Joint CIs are relatively stable and show only slight reductions in the proportion of points that are significant as the sampling rate increases.

| Sampling rate (Hz) | BH | Pointwise | Joint |
|---|---|---|---|
| 20 | 31.00 | 23.58 | 9.17 |
| 25 | 0.73 | 23.27 | 9.09 |
| 30 | 30.23 | 22.67 | 8.72 |
| 40 | 1.09 | 22.66 | 8.50 |
| 50 | 0.73 | 22.38 | 8.28 |
| 125 | 0.80 | 22.02 | 7.63 |

**Appendix 2—table 2.** Functional slope $\widehat{\beta}_1(s)$.

Percentage of time-points in which the coefficient estimates $\widehat{\beta}_1(s)$ are statistically significant. We compare the Benjamini–Hochberg (BH) correction applied to pointwise linear mixed models (LMM) models, the Functional Linear Mixed Models (*FLMM)* pointwise 95% CIs (Pointwise), and the *FLMM* joint 95% CIs (Joint). The BH identifies no statistically significant effects at any time-points, whereas the the *FLMM* Pointwise and Joint CIs identify significant effects at a relatively consistent proportion of time-points.

| Sampling rate (Hz) | BH | Pointwise | Joint |
|---|---|---|---|
| 20 | 0 | 46.72 | 17.90 |
| 25 | 0 | 44.36 | 16.73 |
| 30 | 0 | 43.31 | 15.41 |
| 40 | 0 | 43.57 | 15.90 |
| 50 | 0 | 43.46 | 14.39 |
| 125 | 0 | 42.15 | 13.66 |

# Appendix 3

## Functional mixed models methods

For a thorough introduction to the Functional Mixed Model fitting framework presented in our manuscript, please refer to *Cui et al., 2022*. However, for completeness, we provide a short description of the estimation details and then provide a brief derivation of our proposed estimator. The majority of the details presented here can be found in greater depth in section '3.1. Analytic Inference for Gaussian Functional Data' of *Cui et al., 2022* but we believe we have included a sufficient description of the statistical estimation scheme to explain the derivation of our proposed estimator (presented in **Appendix 3.2**). For clarity, we focus on functional *linear* mixed models. However, our package also provides the capability to fit the wider class of functional generalized linear mixed models for the distributions supported by the `R lme4` package.

## 3.1 Functional linear mixed models

We focus on the functional linear mixed model,

$$\mathbf{Y}_i(s) = \mathbb{X}_i \boldsymbol{\beta}(s) + \mathbb{Z}_i \boldsymbol{\gamma}_i(s) + \boldsymbol{\epsilon}_i(s) \tag{4}$$

where $\mathbf{Y}_i(s) \in \mathbb{R}^{J_i}$ is the vector of observations of functional outcomes at time-point $s$ for subject $i$, $J_i$ is the number of functional observations of subject $i$, and $s \in \{1, 2, ..., S\}$, where $S$ is the number of points on the grid of the functional domain (i.e. the number of trial time-points or trial photometry samples). We take $n$ to be the number of subjects (e.g. animals) and $N = \sum_{i=1}^n J_i$. We denote $\boldsymbol{\beta}(s) \in \mathbb{R}^p$ to be the vector of functional fixed-effects at point $s$. $\boldsymbol{\gamma}_i(s) \in \mathbb{R}^q$ and $\boldsymbol{\epsilon}_i(s) \in \mathbb{R}^{J_i}$ are vectors of mutually independent random-effects and error terms for time-point $s$, with multivariate Gaussian distributions. We denote $\mathbb{X}_i \in \mathbb{R}^{J_i \times p}$ and $\mathbb{Z}_i \in \mathbb{R}^{J_i \times q}$ as (design) matrices of subject $i$ containing the covariates for the fixed- and random-effects terms, respectively. Denote $\mathbb{Z} \in \mathbb{R}^{N \times nq}$ to be a block diagonal matrix where the $i^{th}$ block entry is $\mathbb{Z}_i$ and let $\mathbb{X} \in \mathbb{R}^{N \times p}$ refer to the design matrix from row concatenating the subject-specific matrices $\mathbb{X}_i$. Estimates for the fixed-effects $\boldsymbol{\beta}(s)$ are correlated across trial time-points (i.e. the functional domain), which is incorporated into *joint* inference by assuming $\text{Cov}\left(\boldsymbol{\gamma}(s_1), \boldsymbol{\gamma}(s_2)\right) = \mathbb{G}(s_1, s_2) \in \mathbb{R}^{nq \times nq}$ and $\text{Cov}\left(\boldsymbol{\epsilon}(s_1), \boldsymbol{\epsilon}(s_2)\right) = 0$ for all $s_1 \neq s_2$, where $\boldsymbol{\gamma}(s) = [\boldsymbol{\gamma}_1^T(s) \; \boldsymbol{\gamma}_2^T(s) \, ... \, \boldsymbol{\gamma}_n^T(s)]^T \in \mathbb{R}^{nq}$ and $\boldsymbol{\epsilon}(s) = [\boldsymbol{\epsilon}_1^T(s) \; \boldsymbol{\epsilon}_2^T(s) \, ... \, \boldsymbol{\epsilon}_n^T(s)]^T \in \mathbb{R}^N$.

We used a two-stage approach for fitting functional linear mixed models. In the first step, we fit pointwise linear mixed models at each time-point $s$, yielding fixed-effect estimates which we denote as $\tilde{\boldsymbol{\beta}}(s)$. The pointwise estimator admits the closed form expression, $\tilde{\boldsymbol{\beta}}(s) = [\mathbb{X}^T \mathbb{V}^{-1}(s) \mathbb{X}]^{-1} \mathbb{X}^T \mathbb{V}^{-1}(s) \mathbf{Y}(s)$, where $\mathbb{V}(s) = \mathbb{Z}\mathbb{H}(s)\mathbb{Z}^T + \tilde{\mathbb{R}}(s)$, and $\mathbb{H}(s)$ and $\tilde{\mathbb{R}}(s)$ are the covariance matrices of $\boldsymbol{\gamma}(s)$ and $\boldsymbol{\epsilon}(s)$, respectively. The variance of the pointwise estimator is $\text{Var}\left(\tilde{\boldsymbol{\beta}}(s)\right) = [\mathbb{X}^T \mathbb{V}^{-1}(s) \mathbb{X}]^{-1}$. The correlation across trial time-points can be seen more explicitly through the following expression

$$\text{Cov}\left(\tilde{\boldsymbol{\beta}}(s_1), \tilde{\boldsymbol{\beta}}(s_2)\right) = \left[\mathbb{X}^T \mathbb{V}^{-1}(s_1) \mathbb{X}\right]^{-1} \mathbb{X}^T \mathbb{V}^{-1}(s_1) \mathbb{W}(s_1, s_2) \mathbb{V}^{-1}(s_2) \mathbb{X} \left[\mathbb{X}^T \mathbb{V}^{-1}(s_2) \mathbb{X}\right]^{-1}$$

where $\mathbb{W}(s_1, s_2) = \mathbb{Z}\mathbb{G}(s_1, s_2)\mathbb{Z}^T$. While estimates of $\mathbb{H}(s)$ and $\tilde{\mathbb{R}}(s)$ are provided by standard mixed modeling software, $\mathbb{G}(s_1, s_2)$ must be estimated separately. We detail the estimation scheme below as it plays a critical role in *joint* inference and constitutes the main topic of our statistical contribution. Note that the estimates of $\mathbb{G}(s_1, s_2)$, $\mathbb{H}(s)$, and $\tilde{\mathbb{R}}(s)$ can be smoothed using, for example, fast bivaraite P-splines (*Xiao et al., 2013*) along the functional domain (that is, trial time-points) to reduce variability; and then any negative eigenvalues of the smoothed matrices can be trimmed at 0 to ensure the resulting covariance matrix estimates are positive semi-definite (*Cui et al., 2022*; *Greven et al., 2010*).

Our package provides the option to use an array of smoothing approaches for the regression coefficient estimates, $\tilde{\boldsymbol{\beta}}(s)$, but defaults to using penalized splines (*Ruppert et al., 2003*). When using penalized splines to smooth the pointwise estimates, $\tilde{\boldsymbol{\beta}}(s)$, the covariance matrix for the final fixed-effects estimates admits a closed form expression, and allows for fast calculation of *joint* confidence band estimates. To see this, denote $\widehat{\boldsymbol{\beta}}(s)$ as the final fixed-effect estimates after smoothing the raw pointwise estimates, $\tilde{\boldsymbol{\beta}}(s)$, directly provided by mixed model software. Then a smoothed estimator for the $t^{th}$ fixed-effect is $\widehat{\boldsymbol{\beta}}^{(t)} = \mathbb{S}_t \tilde{\boldsymbol{\beta}}^{(t)}$, where $\tilde{\boldsymbol{\beta}}^{(t)} = \left[\tilde{\beta}^{(t)}(s_1), ..., \tilde{\beta}^{(t)}(s_S)\right]^T \in \mathbb{R}^S$. Thus, $\text{Cov}\left(\widehat{\boldsymbol{\beta}}^{(t)}\right) = \mathbb{S}_t \text{Cov}\left(\tilde{\boldsymbol{\beta}}^{(t)}\right) \mathbb{S}_t$, where $\mathbb{S}_t = \mathbb{B}_t(\mathbb{B}_t^T \mathbb{B}_t + \lambda_t \mathbb{P}_t)^{-1} \mathbb{B}_t^T$, $\mathbb{B}_t$ is a $K$-dimensional spline basis matrix, $\lambda_t$ is a smoothing parameter, and $\mathbb{P}_t$ is a penalty matrix. Typically the number of knots $K \ll S$,

but as long as a sufficient number of knots are specified, the smoother and exact $K$ selected appear to have little impact on the final coefficient estimates or 95% CIs in practice. To be conservative, we used a high number of knots ($K \approx S/4$). In practice, the specific number could be altered depending on the sampling rate of the photometry data analyzed and the specific sensor (since the kinetics of neurochemical signaling systems can vary widely). The results presented in the paper were calculated based upon implementation of our method using thin-plate splines since these performed well in practice.

Our package leverages the well-known `lme4` package in `R` for fitting the pointwise mixed effects models (**Bates, 2010**; **Bates, 2014**). This precludes, however, the specification of structure on the covariance matrix of the errors, $\epsilon_i(s)$ across trials (for a fixed trial time-point $s$) in Gaussian models, functionality provided by, for example, the `nlme` in `R` (**Pinheiro and Bates, 2023**). Future work could explore functional linear mixed models that allow for this functionality. However, in our experimentation analyzing photometry data summary measures (e.g. AUCs) from a range of experiments that observed multiple trials and sessions per animal, we found that model fit criteria (e.g. AIC, BIC, cAIC **Säfken et al., 2018**) were usually much better when accounting for correlation across trials within-animal through random-effect specifications as opposed to placing structure on the covariance of the errors within-subject across trials. Moreover, we found that models that specified the types of error covariance structures applicable to photometry experiments were often slow to fit which is a drawback in our approach to fitting functional mixed models given that it requires fitting many pointwise models. For those reasons, we set our package's default to use the `lme4` package thereby providing both speed and modeling flexibility to users.

We now provide a brief description of our proposed estimator for $\mathbb{G}(s_1, s_2)$. The critical role that this plays in inference and in neuroscience applications arises from the nested designs common in the sophisticated behavioral experiments commonly used alongside fiber photometry. We briefly describe this in **Appendix 3.3**.

## 3.2 Covariance estimator

We begin with a high-level description of the estimator for the covariance matrix, $\mathbb{G}(s_1, s_2)$, and include details below. Denote $Y_{i,j}(s_1) \in \mathbb{R}$ as the functional outcome at time-point $s_1$, for subject $i$ on trial $j$. The method of moments estimator proposed by **Greven et al., 2010** and applied to our functional mixed model estimation procedure was presented in section 3.1, equation 4 from **Cui et al., 2022**,

$$\mathbb{E}\left[ \{Y_{i,k}(s_1) - \mathbf{X}_{i,k}^T \boldsymbol{\beta}(s_1)\} \{Y_{i,j}(s_2) - \mathbf{X}_{i,j}^T \boldsymbol{\beta}(s_2)\} \right] = \sum_{t=1}^{q} \sum_{v=1}^{q} Z_{i,k,t} Z_{i,j,v} \text{Cov}\left( \gamma_{i,t}(s_1), \gamma_{i,v}(s_2) \right) \quad (5)$$

where $t, v$ are random-effect covariate indices. This expression suggests an estimator that regresses the residual products $[Y_{i,k}(s_1) - \mathbf{X}_{i,k}^T \boldsymbol{\beta}(s_1)][Y_{i,j}(s_2) - \mathbf{X}_{i,j}^T \boldsymbol{\beta}(s_2)]$ onto the random-effect covariate products $\{Z_{i,k,t} Z_{i,j,v} : j, k = 1, 2, ...J_i\}$. Concatenating these covariate products into a design matrix, $\tilde{\mathbb{Z}}$, and concatenating the residual products into an 'outcome vector,' $\tilde{\mathbf{Y}}(s_1, s_2) \in \mathbb{R}^N$, allows us to express the estimator as a solution to the least squares problem for each pair $(s_1, s_2)$

$$\hat{\boldsymbol{\alpha}}(s_1, s_2) \in \underset{\boldsymbol{\alpha}(s_1, s_2)}{\operatorname{argmin}} \left\| \tilde{\mathbf{Y}}(s_1, s_2) - \tilde{\mathbb{Z}} \boldsymbol{\alpha}(s_1, s_2) \right\|_2^2 \quad (6)$$

where $\tilde{\mathbb{Z}} \in \mathbb{R}^{N \times \tilde{q}}$ is a matrix of the covariate products, $Z_{i,k,t} Z_{i,j,v}$, described above. After calculating $\hat{\boldsymbol{\alpha}}(s_1, s_2)$, $\widehat{\mathbb{G}}(s_1, s_2)$ is obtained by re-organizing the elements of the vector $\hat{\boldsymbol{\alpha}}(s_1, s_2)$ into the entries of the matrix $\widehat{\mathbb{G}}(s_1, s_2)$. This estimator is flexible and in principle is agnostic to the random-effect specification, but in practice requires extension for general random-effects specifications, which we derive here.

Before discussing the extension to this estimator, we provide additional necessary notation. Since a single random-effect can require many columns of $\mathbb{Z}$ to encode the corresponding covariate (e.g. if there are many session-specific random intercepts), separate indices are needed to distinguish between a single random-effect distribution and the (potentially many) columns of the random-effect design matrix associated with draws from that random-effect distribution.

The $\tilde{\mathbb{Z}} \in \mathbb{R}^{N \times \tilde{q}}$, described above, is a block diagonal matrix where the $i^{th}$ block entry is $\tilde{\mathbb{Z}}_i$. The number of columns, $\tilde{q}$, will vary depending on the random-effect specification. For example, if one

specifies a model that includes session-specific random-intercepts, this would require one column in $\mathbb{Z}$ (and thus multiple columns in $\tilde{\mathbb{Z}}$) for each animal and session to encode the corresponding animal- and session-specific covariates associated with these random-intercepts.

Let $q^*$ be the number of unique random-effects distributions for a given model, where $q^* \leq q$. Take as an example a model that includes: (1) a subject-specific random intercept, $\gamma_{0,i}(s)$, and (2) subject- and session-specific random intercepts, $\gamma_{1,i,l}(s)$ for participant $i$ on session $l$. If we assume that $\gamma_{1,i,l}(s) \overset{iid}{\sim} \mathcal{N}(0, \tilde{\sigma}^2) \ \forall \ i, l$, then $q^* = 2$, but $q > 2$ (and thus $\tilde{q} > 2$) since encoding the subject- and session-specific random intercepts, $\gamma_{1,i,l}(s)$ would require many indicator variables (entered as columns of $\mathbb{Z}_i$). More generally, for random-effect covariate, $r$, denote $\mathcal{I}_r$ as the set of column indices of $\mathbb{Z}$ that encode random-effect covariate $r$. Although we describe $\gamma_{r,i,l}(s)$ as if it were drawn from a univariate Gaussian for explanatory purposes, neither the estimator nor the modeling software assume independence between random-effects from *different* covariates.

Expanding section 3.1, equation 4 from **Cui et al., 2022** to be in terms of the column indices of the design matrix $\mathbb{Z}_i$,

$$\sum_{t=1}^{q} \sum_{v=1}^{q} Z_{i,k,t} Z_{i,j,v} \text{Cov}(\gamma_{i,t}(s_1), \gamma_{i,v}(s_2))$$

$$= \sum_{r=1}^{q*} \sum_{m=1}^{q*} \sum_{w \in \mathcal{I}_r} \sum_{b \in \mathcal{I}_m} Z_{i,k,w} Z_{i,j,b} \text{Cov}(\gamma_{i,w}(s_1), \gamma_{i,b}(s_2))$$

$$= \sum_{r=1}^{q*} \sum_{m=1}^{q*} \left\{ \sum_{w \in \mathcal{I}_r} Z_{i,k,w} \left[ \sum_{b \in \mathcal{I}_m} Z_{i,j,b} \text{Cov}(\gamma_{i,w}(s_1), \gamma_{i,b}(s_2)) \right] \right\}.$$

Now recall that $\gamma_{r,i,w}(s) \overset{iid}{\sim} \mathcal{N}(0, \tilde{\sigma}_r^2)$ for all $w \in \mathcal{I}_r$. Thus, $\text{Cov}\left(\gamma_{r,i,w}(s_1), \gamma_{r,i,b}(s_2)\right) = \tilde{\rho}_{r,m}$ for a covariance value, $\tilde{\rho}_{r,m}$, that is equal across all $w \in \mathcal{I}_r$ and $b \in \mathcal{I}_m$. It then follows that we can simplify the above as,

$$\sum_{r=1}^{q*} \sum_{m=1}^{q*} \left\{ \sum_{l \in \mathcal{I}_r} Z_{i,k,l} \left[ \sum_{b \in \mathcal{I}_m} Z_{i,j,b} \text{Cov}(\gamma_{i,l}(s_1), \gamma_{i,b}(s_2)) \right] \right\}$$

$$= \sum_{r=1}^{q*} \sum_{m=1}^{q*} \left[ \sum_{l \in \mathcal{I}_r} Z_{i,k,l} \right] \left[ \sum_{b \in \mathcal{I}_m} Z_{i,j,b} \right] \text{Cov}(\gamma_{i,r}(s_1), \gamma_{i,m}(s_2)).$$

This expression suggests an estimation strategy that takes the product $[\Sigma_{l \in \mathcal{I}_r} Z_{i,k,l}][\Sigma_{b \in \mathcal{I}_m} Z_{i,j,b}]$ as a 'covariate' in the OLS-based estimator described in **Equation 5**. Organizing these products into the columns of $\tilde{\mathbb{Z}}$ does not result, however, in a full rank matrix for all random-effect specifications. In such cases, the solution to the problem described in **Equation 6** is not unique. We solve for the 'ridgeless regression' solution to the problem described in **Equation 6** because it yields the minimum $\ell_2$ norm estimator and exhibits desirable statistical properties (**Hastie et al., 2019**). This can be expressed as the solution to the optimization problem,

$$\min_{\alpha(s_1, s_2)} \|\alpha(s_1, s_2)\|_2^2$$
$$\text{s.t.} \quad \alpha(s_1, s_2) = \underset{\alpha(s_1, s_2)}{\text{argmin}} \left\| \widehat{\epsilon}(s_1, s_2) - \tilde{\mathbb{Z}} \alpha(s_1, s_2) \right\|_2^2 \tag{7}$$

where $\widehat{\epsilon}(s_1, s_2)$ are the residual products (i.e. with entries $\widehat{\epsilon}_{i,j,k}(s_1, s_2) = [Y_{i,k}(s_1) - \mathbf{X}_{i,k}^T \boldsymbol{\beta}(s_1)][Y_{i,j}(s_2) - \mathbf{X}_{i,j}^T \boldsymbol{\beta}(s_2)])$ and $\boldsymbol{\alpha}(s_1, s_2) \in \mathbb{R}^{\tilde{q}}$. We construct $\widehat{\mathbb{G}}(s_1, s_2)$ through reorganizing the elements of $\widehat{\boldsymbol{\alpha}}(s_1, s_2)$ into the matrix $\widehat{\mathbb{G}}(s_1, s_2)$. The $\boldsymbol{\alpha}(s_1, s_2)$ can be estimated with the closed form expression,

$$\widehat{\boldsymbol{\alpha}}(s_1, s_2) = \left( \tilde{\mathbb{Z}}^T \tilde{\mathbb{Z}} \right)^+ \tilde{\mathbb{Z}}^T \tilde{\mathbf{Y}}(s_1, s_2) = \mathbb{M} \tilde{\mathbf{Y}}(s_1, s_2),$$

where $(\cdot)^+$ denotes the Moore-Penrose pseudoinverse inverse (*Hastie et al., 2019*). Thus, while the estimator requires one to solve $S(S+1)/2$ problems of the form above (i.e. one for each unique $\{s_1, s_2\}$ pair), in practice we only need to calculate $\mathbb{M}$ once for all $\{s_1, s_2\}$ pairs. This allows for estimation of $\mathbb{G}$ (i.e. for all unique $\{s_1, s_2\}$ pairs) for datasets with many observations, complex random-effects specifications, and large $S$ within a couple of seconds in total. We found that this approach performed well in practice, and it is the default in our package. All results presented in the main text apply to this version.

The above estimator allows for a fast, flexible extension of the work in *Cui et al., 2022*, thereby allowing for 95% CI calculation for general random-effects specifications. This statistical contribution is critical for neuroscience, since studies often exhibit nested experimental designs that require sophisticated random-effect models to properly capture the rich information contained in photometry signals.

## Alternative covariance estimators

We also explored a collection of related strategies. These are motivated by the fact that one can regress the residual products onto $\{Z_{i,k,v} Z_{i,j,t} : v, t = 1, ..., q\}$, instead of onto the products of the sums (across indices $t$, $v$), as proposed above. Denote $\widehat{\alpha}_{v,t}$ as the coefficient estimate associated with the 'covariate' $Z_{i,j,v} Z_{i,k,t}$ in this regression. We exploit the fact that $\mathrm{Cov}\left(\gamma_{i,w}(s_1), \gamma_{i,b}(s_2)\right) = \tilde{\rho}_{r,m}$ for a covariance value, $\tilde{\rho}_{r,m}$, common across all $w \in \mathcal{I}_r$ and $b \in \mathcal{I}_m$. Solving for the desired quantity then yields the estimator

$$\widehat{\mathrm{Cov}}\left(\gamma_{i,r}(s_1), \gamma_{i,m}(s_2)\right) = \frac{1}{|\mathcal{I}_r|\,|\mathcal{I}_m|} \sum_{w \in \mathcal{I}_r} \sum_{b \in \mathcal{I}_m} \widehat{\alpha}_{w,b}. \tag{8}$$

We found this approach (available in our package by setting the argument `MoM = 2`) was slower, more memory intensive, and yielded confidence interval coverage comparable to the approach proposed above. We also explored the performance of two estimators for $\mathbb{G}$ based on mathematical programs that enforce non-negativity of variance components (before eigenvalue trimming). The first mathematical program is

$$\begin{aligned}
&\min_{\boldsymbol{\alpha}(s_1,s_2)} \|\boldsymbol{\alpha}(s_1,s_2)\|_2^2 \\
&\text{s.t.} \quad \boldsymbol{\alpha}(s_1,s_2) = \operatorname*{argmin}_{\boldsymbol{\alpha}(s_1,s_2)} \left\|\widehat{\boldsymbol{\epsilon}} - \tilde{\mathbb{Z}}\boldsymbol{\alpha}(s_1,s_2)\right\|_2^2 \\
&\quad \alpha_m \geq 0 \quad \text{for } m \in \mathcal{M}
\end{aligned} \tag{9}$$

where $\mathcal{M}$ denotes the set of indices corresponding to variance elements.

*Equation 8* suggests that one could alternatively constrain the sum, as opposed to each element in the summation $\sum_{l \in \mathcal{I}_r} \sum_{b \in \mathcal{I}_m} \widehat{\alpha}_{l,b}$. Thus we also explored performance of the estimator based on the mathematical program,

$$\begin{aligned}
&\min_{\boldsymbol{\alpha}(s_1,s_2)} \|\boldsymbol{\alpha}(s_1,s_2)\|_2^2 \\
&\text{s.t.} \quad \boldsymbol{\alpha}(s_1,s_2) = \operatorname*{argmin}_{\boldsymbol{\alpha}(s_1,s_2)} \left\|\widehat{\boldsymbol{\epsilon}} - \tilde{\mathbb{Z}}\boldsymbol{\alpha}(s_1,s_2)\right\|_2^2 \\
&\quad \sum_{l \in \mathcal{I}_m} \alpha_l \geq 0 \quad \text{for } m \in \mathcal{M}.
\end{aligned} \tag{10}$$

We found that, in practice, the performance of the estimators based on the mathematical programs *Equation 7*, *Equation 9*, and *Equation 10*, performed comparably in simulations. This may be because the eigenvalue trimming applied to the solutions to the above optimization problems yields similar final estimates. Our software automatically implements estimators based on all approaches described above, but defaults to the first estimator proposed above, which is based on the solution to mathematical program *Equation 7*.

In simulations, we observed that the regression coefficient estimate 95% CIs calculated using our proposed $\mathbb{G}$ estimator achieved nearly nominal *joint* coverage even with model specifications that yield reduced rank $\tilde{\mathbb{Z}}$. Therefore, the above estimator allows for a fast, flexible extension of the work in *Cui et al., 2022*, thereby allowing for 95% CI calculation for general random-effects specifications. This statistical contribution is important for neuroscience since studies often exhibit

nested experimental designs that require sophisticated random-effect models to properly capture the rich information contained in photometry signals.

## Reduced rank problem

We point out that the method of moments estimators applied here (proposed in *Greven et al., 2010*) exhibit a non-identifiability property. While this did not substantially influence the resulting 95% confidence interval coverage in our simulations, we argue that future attention to this issue is warranted. Such settings arise anytime one includes, for example, nested random-effect structures. This is a drawback because these types of random-effect specifications are critical to properly model the sophisticated behavioral designs common in neuroscience which often exhibit multiple layers of nesting. For instance, suppose a collection of animals are trained on multiple sessions, and within each session are trained on multiple trials. Then trial is nested in session, which is nested in animal/subject. Inclusion of animal- and session-specific random intercepts, a common random-effect specification for such experimental designs, will exhibit the challenges above.

In order to make the issue concrete, we describe perhaps the simplest case where the above problem presents. Specifically, we show how the original estimator requires extension in a simple *FLMM* model that is analogous to the functional analog of the paired samples t-test: a model that includes a single binary covariate such as a treatment indicator (i.e. $X_{i,j} \in \{0,1\}$), a random subject-specific intercept, and a random subject-specific slope. Suppose we observe a collection of trials indexed by $j$, on a collection of animals indexed by $i$. Then the above model can be expressed as

$$\mathbb{E}\left[Y_{i,j}(s) \,\middle|\, \mathbf{X}_{i,j}, \mathbf{Z}_{i,j}, \boldsymbol{\gamma}_i(s)\right] = \beta_0(s) + \gamma_{0,i}(s) + X_{i,j}\left[\beta_1(s) + \gamma_{1,i}(s)\right]. \tag{11}$$

This model yields the vector of random-effect covariates $\mathbf{Z}_{i,j}^T = [1\ X_{i,j}]$, and the associated matrix $\tilde{\mathbb{Z}}_i$ is

$$\tilde{\mathbb{Z}}_i = \begin{bmatrix} 1 & 2X_{1,1} & X_{1,1}^2 \\ 1 & 2X_{1,2} & X_{1,2}^2 \\ \vdots & \vdots & \vdots \\ 1 & 2X_{1,J_1} & X_{1,J_1}^2 \end{bmatrix} = \begin{bmatrix} 1 & 2X_{1,1} & X_{1,1} \\ 1 & 2X_{1,2} & X_{1,2} \\ \vdots & \vdots & \vdots \\ 1 & 2X_{1,J_1} & X_{1,J_1} \end{bmatrix}.$$

The above equality follows trivially because $X_{i,j}^2 = X_{i,j}$ since $X_{i,j}$ is binary. The columns, therefore, exhibit the linear dependence $2\tilde{\mathbb{Z}}_{.,2} = \tilde{\mathbb{Z}}_{.,3}$ and thus $\tilde{\mathbb{Z}}$ is not full rank. To our knowledge, no re-coding of the binary variable $X_{i,j}$ resolves the above linear dependence problem without changing the contrast e.g., one could fit a model without a random intercept and include the covariates $[X_{i,j}\ (1 - X_{i,j})]$.

More generally, the reduced rank problem presents whenever a subject-specific random intercept is included (because it is encoded by a column of ones, $\mathbb{1}$, in $\tilde{\mathbb{Z}}$) alongside any binary covariate in $\mathbb{Z}$. As shown in the example above, this is because the column of ones used for random intercepts, when multiplied by binary covariates to construct the matrix $\tilde{\mathbb{Z}}$, will create linear dependence in the columns of $\tilde{\mathbb{Z}}$. This is problematic because binary covariates are needed to encode dichotomous and factor covariates (e.g. in the functional analog of t-tests and ANOVAs), as well as, for example, session-specific random intercepts in nested designs. As such, resolving this issue is critical to properly model experiments that regularly arise in neuroscience.

## 3.3 Random-effects structures and interpretations for neuroscience experiments

Functional random intercepts that are unique to each trial and/or sessions can be used to account for animal-to-animal, session-to-session, and trial-to-trial heterogeneity in the signal and is one example where one might include a covariate in the random-effects (that is, included in $\mathbb{Z}_i$) but not in the main effects terms, $\mathbb{X}_i$. Inclusion of these session- and trial-level random-effects is an example of one way to account for correlation within an animal across trials and sessions (photometry signal on trial $j$ for animal $i$ is likely similar to the signal on trial $j + 1$). Indeed, many neuroscience studies involving behavioral paradigms that involve multiple sessions and trials (and potentially multiple groups) may require complicated random-effects structures to account for animal-to-animal, session-

to-session, and trial-to-trial heterogeneity in the photometry signal. For this reason, we explored the performance of many models to learn how to best model variability in the data.

*FLMM* yields interpretable regression coefficient estimates. Conveniently, they can be interpreted conditional on the functional random-effects or marginally with respect to the random-effects since both interpretations are numerically equivalent in the linear setting. For an intuitive introduction to this equivalence for linear mixed models, we recommend sections 2.2 and 7.4 of *Fitzmaurice, 2008*. We present the marginal interpretation here because it is often more intuitive for photometry analyses. However, the conditional version has its own advantages.

## 3.4 Functional regression model classes

**Appendix 3—table 1.** Regression model classes based on functional vs scalar response variables and for longitudinal (repeated measures) vs cross-sectional data.
We use 'functional mixed models' as a short-hand for function-on-scalar mixed models. We use 'functional regression' as a short-hand for single-level function-on-scalar regression.

|  | Longitudinal | Cross-sectional |
|---|---|---|
| Functional | Functional mixed models | Functional regression |
| Scalar | Generalized linear mixed-effects (GLMM) | Generalized linear models (GLM) |

**Appendix 3—table 2.** Cross-sectional regression model classes based on functional vs. scalar predictor variables (i.e. covariates) and functional vs. scalar outcome variables.
We take the FoFR, FoSR, and SoFR to be the single-level (non-longitudinal) versions of these methods.

|  | Functional outcome | Scalar outcome |
|---|---|---|
| Functional predictors | Function-on-function regression (FoFR) | Scalar-onfunction regression (SoFR) |
| Scalar predictors | Function-onscalar regression (FoSR) | Generalized linear models (GLM) |

## Appendix 4

### Reanalysis figures and methods

In the following sections, we provide additional details for the reanalyses (*Jeong et al., 2022*) presented in the main text. In cases where it is helpful, we provide quotes from the original paper (*Jeong et al., 2022*) that informed our modeling decisions. These were taken from the main text of the paper, figure captions, and the supplement. In some places, we omitted minor portions of text to make the quotations more concise (e.g. figure references or citation numbers).

4.1 'Using *FLMM* to test associations between signal and covariates throughout the trial' reanalyses

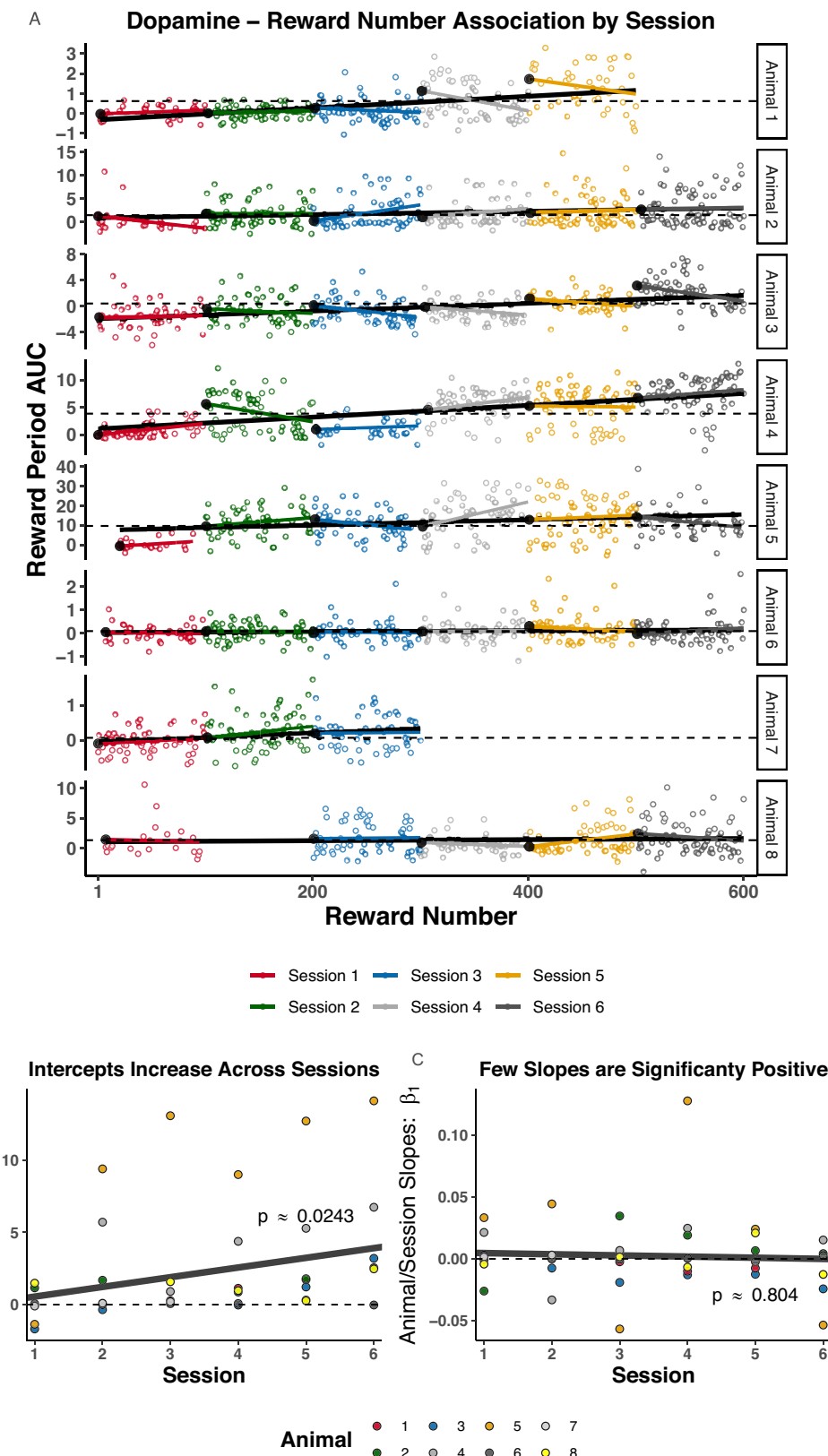

**Appendix 4—figure 1.** `Reward Number`–Area Under the Curve (AUC) correlation within-session and across-session. (**A**) Color indicates session number, rows denote animal number. `Trial Number` is the within-session `Reward Number` and ranges from 1 to 100 for each session. The black line that spans across sessions is a `Reward Number`–AUC linear regression fit, while the session color lines indicate a within-session `Trial Number`–AUC linear regression fit. The large black circles on the left side of each session-specific fit is the intercept, paramterized to yield the interpretation as the 'expected AUC on the first trial of the corresponding session.' Dotted horizontal lines are set at the median of the intercepts to facilitate comparison. The intercepts tend to rise across sessions, while few slopes are significantly positive. (**B, C**) Each dot indicates the estimated intercept value (**B**) or slope (**C**) from the fits shown in (**A**). Lines and p-values were calculated in an linear mixed model (LMM) that was fit to the session-specific linear regression slopes, $\widehat{\beta}_1$, and intercepts, $\widehat{\beta}_0$, shown in (**A**). The LMM included animal-specific random intercepts and slopes. These plots quantify the trend observed in (**A**): the estimated intercepts significantly increase across sessions, but the slopes are mostly negative.

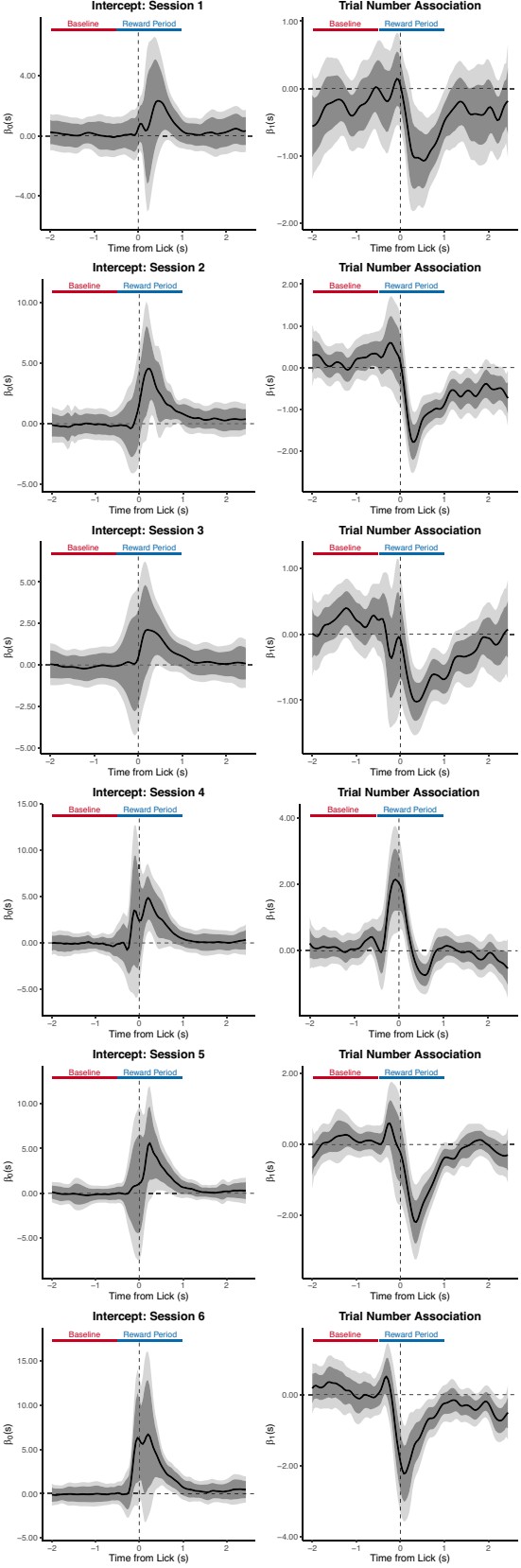

**Appendix 4—figure 2.** `Trial Number`–dopamine (DA) correlation within-session on the random inter-reward interval (`IRI`) task. Row indicates session number. The intercept is paramterized to yield the interpretation as the
*Appendix 4—figure 2 continued on next page*

*Appendix 4—figure 2 continued*
'expected signal magnitude on the first trial of the corresponding session.' Effects are aligned to the first lick after reward delivery. This is the session-by-session version of the analysis presented in ***Figure 5J–K***.

## 4.2 Lick bout correlation

We present further analyses that illustrate how *FLMM* reveals effects obscured by standard methods. The authors note that the `Reward Number`–DA association, reported in their analyses, could arise from `Lick Rate` increases that also correlate with `Reward Number`. They tested this alternative explanation by applying a Pearson correlation between `Lick Rate` and DA. While they reported no significant association, the plot for the `Lick Rate` covariate from *FLMM* (***Appendix 4—figure 3***), shows `Lick Rate` exhibits (1) a significant positive association with DA before Lick-bout onset, and (2) a negative association after Lick-bout onset (reaching *joint* significance at ~1 sec). These results suggest that the correlation analysis of reward period AUC in the paper missed this effect because the AUC summarized a time-window that contained opposing effects. The underlying association was likely diluted by averaging over time-points when `Lick Rate` is both positively correlated with DA (the first 0.5 sec of the reward period, visible in positive *FLMM* coefficient estimates) and negatively correlated with DA (the final 1 sec of the 1.5 sec time-window, with negative *FLMM* coefficient estimates). Identifying this effect with a summary measure would have required selecting these time intervals perfectly a priori; this is not necessary with *FLMM*.

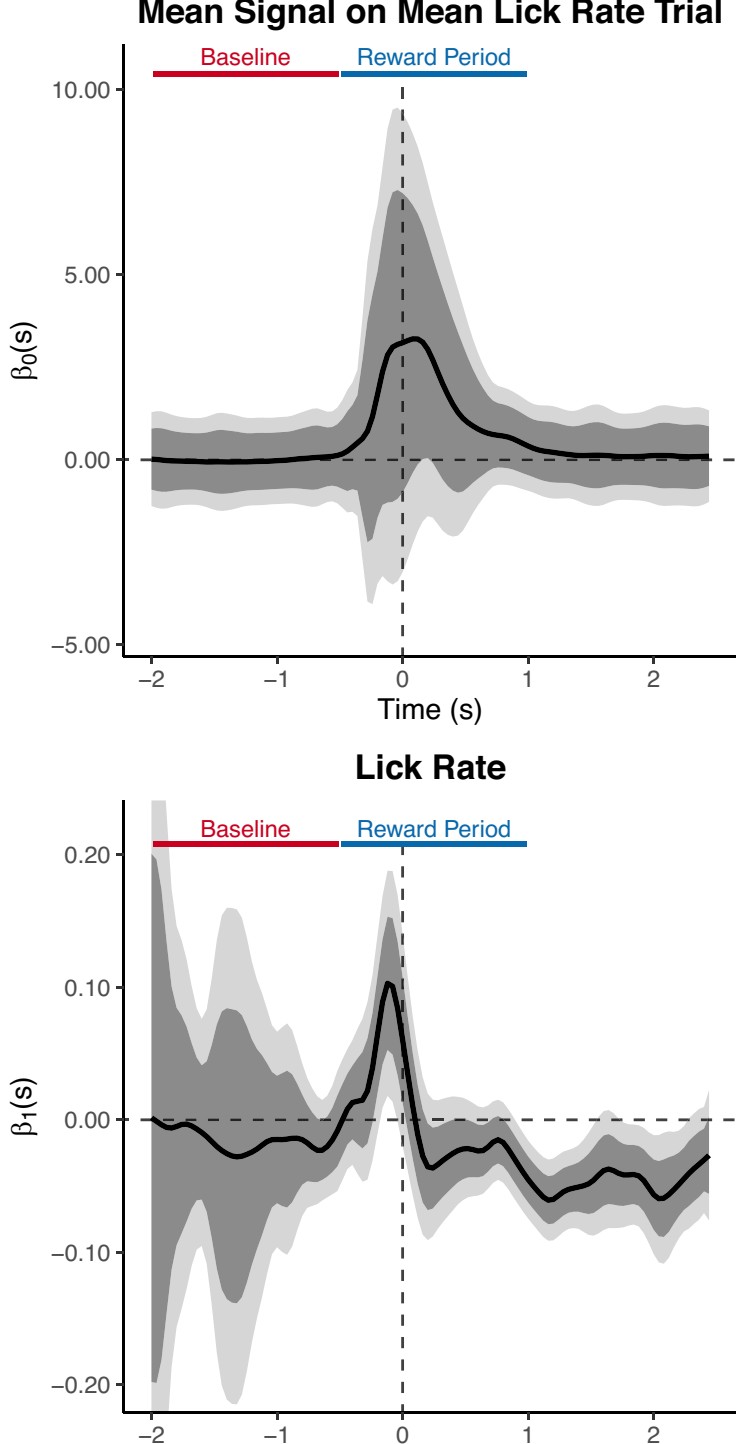

**Appendix 4—figure 3.** Functional Linear Mixed Models (*FLMM*) reveals details occluded by summary measure analyses. Coefficient estimates from an *FLMM* analysis of the random inter-trial interval (`IRI`) reward delivery experiment. The top row contains the intercept term plots where the title provides an interpretation of the intercept: the average dopamine (DA) signal on trials when `Lick Rate` is at its average value. The bottom row shows the coefficient estimate plot of the covariate in the model. The 'Baseline' and 'Reward Period' bars show the trial period that the original authors used to calculate the summary measure (Area Under the Curve, AUC). Specifically, they quantified DA by a measure of normalized AUC of $\Delta F/F$ during a window 0.5 sec before to

*Appendix 4—figure 3 continued*

1 sec after the first lick following reward delivery. All plots are aligned to this first lick after reward delivery. The interpretation of the y-value of the bottom plot at any time-point $s$: the mean change in the dopamine signal at $s$ for a one unit change in `Lick Rate`. Association between DA and `Lick Rate` aligned to lick bout onset. Time-points when `Lick Rate` was negatively associated with DA (negative coefficient estimates in the final 1 sec of the 1.5 sec window) may have diluted time-points when they were positively associated (positive coefficient estimates in the first 0.5 sec of the reward period).

## `Lick Rate` model

We conducted an *FLMM* reanalysis of the analyses shown in Figure S8 C-D of *Jeong et al., 2022*. The figure caption notes that "The consummatory lick rate is not correlated with dopamine reward response." We used the methods described in the quoted paragraphs above, and provide further details in **Appendix 4.2**. We fit the lick rate ($LR_{i,j,l}$) *FLMM*

$$\mathbb{E}\left[Y_{i,j,l}(s) \mid \mathbf{X}_{i,j,l}, \mathbf{Z}_{i,j,l}, \gamma_i(s)\right] = \beta_0(s) + \beta_1(s)LR_{i,j,l} + \gamma_{0,i}(s) +$$
$$\gamma_{1,i,l}(s) + TN_{i,j,l}\left[\gamma_{2,i}(s) + \gamma_{3,i,l}(s)\right]$$

where `Trial Number` is denoted as $TN_{i,j,l}$. This can be fit with the code:

```
model_fit = fui(photometry ~ lick_rate + (trial | id/session),
                data = photometry_data,
                subj_ID = "id")
```

## Consummatory lick-bout extraction

As stated in 'Data Analysis: Experiment 1' of the Supplement (p.3–4), consummatory Lick-bouts were defined as follows:

"To test whether lick rate affects dopamine reward response, we first classified licks into consummatory and non-consummatory licks. A group of licks with less than 1 [sec] interval was defined as a lick bout...Every lick in the first lick bout after reward delivery was defined as a consummatory lick, and all other licks were defined as nonconsummatory licks...To avoid any influence of the previous reward on the dopamine response or behavior to the current reward, we excluded rewards with less than 3 [sec] IRI from the previous reward (22.6 ± 2.4%) for the above analyses. Rewards without lick until the next reward (5.7 ± 0.1%) were also excluded from analyses."

For an `IRI` that was less than 3 sec, we excluded the trial before and after that `IRI` so as to avoid any influence on either trial's signal. Our code implementations of the above methods can be found on the github page: https://github.com/gloewing/photometry_fLMM (copy archived at *Loewinger, 2024*).

The reward period time-window and methods were described in the Supplement of *Jeong et al., 2022*:

"Dopamine response to reward was defined as the normalized area under curve (AUC) of ΔF/F during reward period. Reward period was defined as -0.5 to 1 [sec] from the first lick after reward delivery. We defined the window with reference to the first lick time, not reward delivery time, because the response latency to reward differs across trials. Also, we used a window starting from 0.5 [sec] ahead the first lick because dopamine response started to increase even before an animal made the first lick (as they get better at sensing reward delivery in late sessions). The AUC during 1.5 [sec] time window before reward period was subtracted from AUC during reward period to normalize baseline activity."

## 4.3 Photobleaching

In main text section **Using FLMM to test associations between signal and covariates throughout the trial**, we conducted analyses that showed DA decreased within-session during the post-lick interval of the reward period ([0, 1] sec from Lick-onset): `Trial Number` was negatively correlated with signal magnitude during the post-lick period. We include a portion of main *Figure 5* here to assist in the photobleaching discussion.

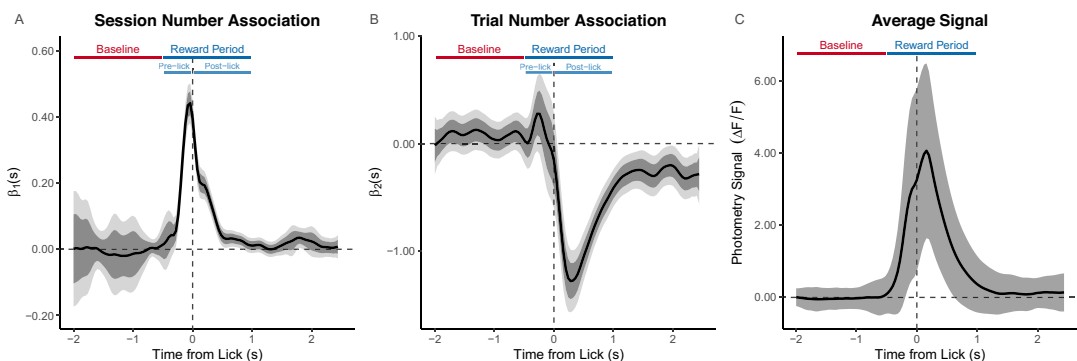

**Appendix 4—figure 4.** Random inter-reward interval (`IRI`) experiment aligned to Lick-onset: `Reward Number`–dopamine (DA) association analyzed as within-session (`Trial Number`) and between-session (`Session Number`) linear effects. The average reward signal shows the average trace with standard error of the mean indicated by the shaded region. The `Trial Number Session Number` effects are functional linear mixed model (*FLMM*) coefficient estimate plots.

We reasoned that if photobleaching caused the within-session decrease post-lick, indications of photobleaching would also be evident in other time-windows in the trial when the average signal was high (e.g. the pre-lick period). We assumed this would hold even if photobleaching depended on light intensity in a non-linear fashion (*Serra and Terentjev, 2008*), since we reasoned the relationship would still be monotonic. The signal–`Trial Number` association is, however, actually slightly positive pre-lick (albeit non-significant), indicating that the signal does not decrease on average within-session pre-lick. Moreover, if the degree of photobleaching scales with signal magnitude, then we would expect to see a negative signal–`Trial Number` association only during the [0,0.5] sec time-window when the mean signal is higher than it was during the pre-lick period (since the pre-lick period does not exhibit any negative signal–Trial Number correlation). Instead, the negative correlation is evident in a time-window when the mean signal has returned to close to zero [0, 2.5] sec. In *Appendix 4—figure 2*, we show the same analyses conducted on each session separately. *FLMM* estimates a large positive `Trial Number` effect pre-lick and post-lick on session 4, despite the fact that the average signal was higher than in sessions 1–3 when the `Trial Number` is negative. Since there are sessions on which it is possible to detect positive `Trial Number` effects, we reasoned that photobleaching would not have occluded any true DA increases within-session. Finally, we repeated the above analyses while adjusting for lick frequency and found that controlling for various behavioral engagement summaries did not impact the `Trial Number` effects.

In *Appendix 4—figure 5*, we conducted a similar analysis but aligned the signal to reward-delivery. If photobleaching were the main cause of the within-session reductions described above, one might expect the `Trial Number` coefficient to be most negative when the average signal is most positive. Instead, these analyses show that the within-session signal decrease (i.e. the negative `Trial Number` effect) and the average signal exhibit distinct temporal dynamics. For example, the peak average signal occurs around 0.75 sec post-reward, while the `Trial Number` coefficient is most negative around 1.25–1.5 sec. Moreover, while the average signal magnitude has returned to nearly 0 around 1.75 sec, the `Trial Number` coefficient remains significantly negative until at least 2.5 sec after reward-delivery.

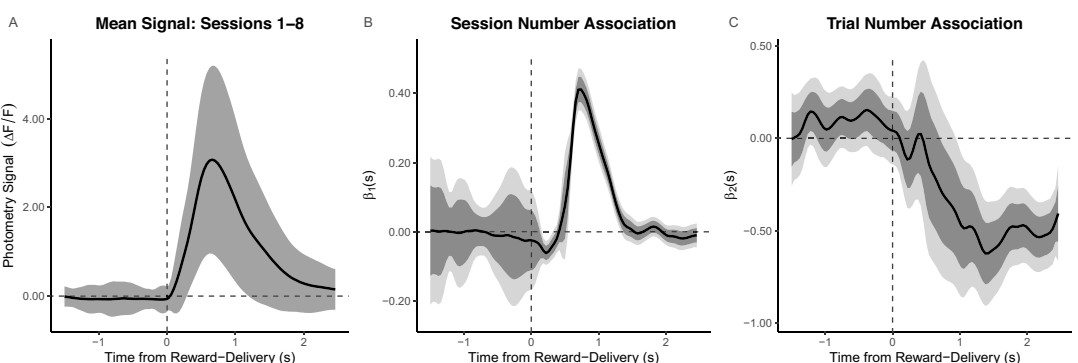

**Appendix 4—figure 5.** Random inter-reward interval (`IRI`) experiment aligned to reward-delivery: `Reward Number`–dopamine (DA) association analyzed as within-session (`Trial Number`) and between-session (`Session Number`) linear effects. The average reward signal shows the average trace with standard error of the mean indicated by the shaded region. The `Trial Number Session Number` effects are functional linear mixed model (*FLMM*) coefficient estimate plots.

### 4.3.1 Photobleaching: Delay length experiment

We next sought to determine whether within-session decreases occurred in a different reward learning task collected from the same mice. We analyzed cue responses on the data from the Delay Length experiments (presented in main text section **Using FLMM to compare signal 'temporal dynamics' across conditions**). We used data from the final short-delay session because the animals were well trained on the Pavlovian task at that stage. During personal communications, *Jeong et al., 2022* suggested these analyses because it provided an opportunity to analyze stabilized event-triggered DA responses in a different task from the same animals. *Appendix 4—figure 6* shows the peak magnitude of the average signal at cue-onset (of the delay length data) is about 4.5 $\Delta F/F$ units (see the Intercept plot), which is about 15% *higher* than the peak magnitude of the average signal during the post-lick period of the experiment descried above (5 shows that the peak magnitude of the average signal was about 4 $\Delta F/F$ units). Because the signal is higher and photobleaching is thought to exert a greater effect on larger signals, one would expect photobleaching to have a *larger* effect in this experiment (i.e. a more negative `Trial Number` effect). However, *Appendix 4—figure 6* shows that the within-session signal *increases* significantly over trials during the cue period (i.e. a positive `Trial Number` effect). *Appendix 4—figure 6* also shows that the signal decreases within-session across trials at reward-delivery (3 sec after cue-onset) despite exhibiting a substantially lower average signal than during the Cue Period. This echos the within-session reductions observed around reward-consumption on the random `IRI` task above. We note additional analyses that adjusted for anticipatory licking and other indications of behavioral engagement did not noticeably impact the `Trial Number` effects. Analyses that pooled together multiple short-delay sessions yielded similar results. Taken together, these analyses of data from the same animals provide additional evidence against a photobleaching explanation as the main contributor for the within-session reduction described in *Figure 5*. One potential caveat is that there may be additional unmeasured factors (e.g. change in motivation) that increase across trials within a session, thereby occluding an effect of photobleaching. However, we reason that the parsimonious explanation in this case is that photobleaching is not a significant concern.

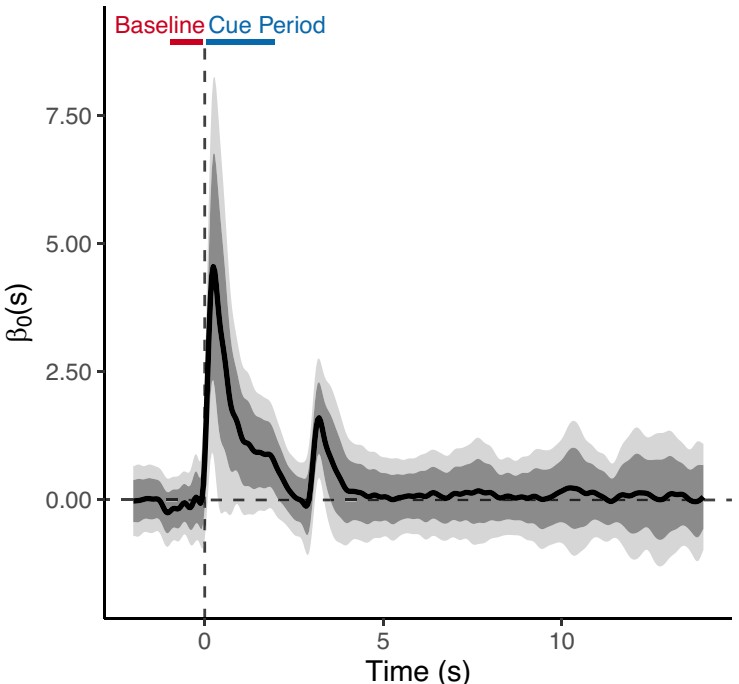

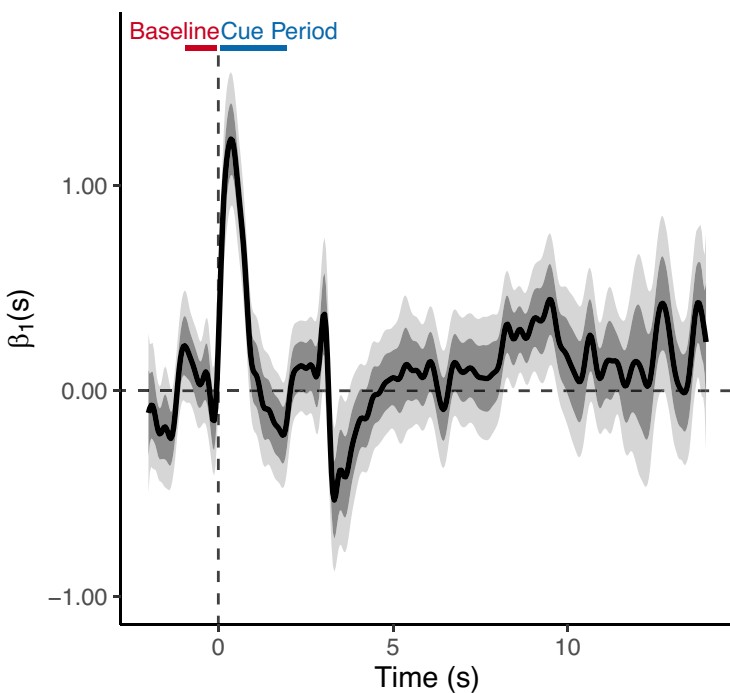

**Appendix 4—figure 6.** Functional Linear Mixed Models (*FLMM*) identifies how the signal increases across trials during the Cue Period and decreases across trials after reward-delivery (3 sec).

## 4.3.2 Photobleaching: Background reward experiment

Next, we show results from analyses suggested during personal communications with ***Jeong et al., 2022*** to further rule out the possibility that the within-session decrease was a result of

photobleaching. We specifically reanalyzed 'Test 7' in *Jeong et al., 2022* (original analysis results shown in *Figure 4M–P* of their manuscript). They described the experiment as follows:

"To test whether the significant positive dopamine responses ollowing extinction reflect a retrospective association between the cue and reward, we selectively reduced the retrospective association without reducing the prospective association. We maintained the fixed reward following the cue but added unpredictable rewards during the inter-trial interval. In this experiment, not all rewards are preceded by the cue (i.e. retrospective association is weak), but all cues are followed by reward (i.e. prospective association is high). ANCCR predicts a rapid drop in dopamine cue response whereas RPE predicts no change in cue response if TDRL only considers the cue-reward 'trial period' (Test 7, fig. S10). The dopamine cue response remained significantly positive but decayed across trials faster than during extinction."

This experiment was like long-delay sessions in the Delay Length experiment (i.e. a CS+ followed by reward-delivery 9 sec later), but it also included rewards delivered randomly without any predictive stimuli during the inter-trial interval. These 'background rewards' were similar to the reward-delivery schedule in the random `IRI` experiment presented in *Figure 5* and, therefore, provide a critical point of comparison. Similar to our random `IRI` experiment analyses, we analyzed the same 'reward period' time window aligned to the first lick after reward-delivery. We removed background reward 'trials' that were too close to each other to avoid signal bleed-over from adjacent trials (e.g. trials with inter-reward intervals that were too short), and trials for which no licks occurred between two successive reward-deliveries (background or reward-predicted) to avoid 'double-counting' a trial.

*Appendix 4—figure 7* shows *FLMM* estimates a negative `Trial Number` effect. The magnitude of this negative coefficient is comparable to that seen in the random rewards experiments post-lick (i.e. after the first lick following reward-delivery). However, the within-session effect is most negative pre-lick, unlike what we observed in the random `IRI` task presented in *Figure 5*. It seems unlikely that this would be solely explained through satiation since the animals cannot be consuming the reward in the pre-lick period. Finally, we note additional analyses that adjust for lick frequency and other indications of behavioral engagement did not impact the `Trial Number` effects. Thus, while these results might be expected if photobleaching was causing the within-session signal reductions shown in *Figure 5*, we argue that the results suggest that it is unlikely that photobleaching is the sole contributor to these within-session reductions.

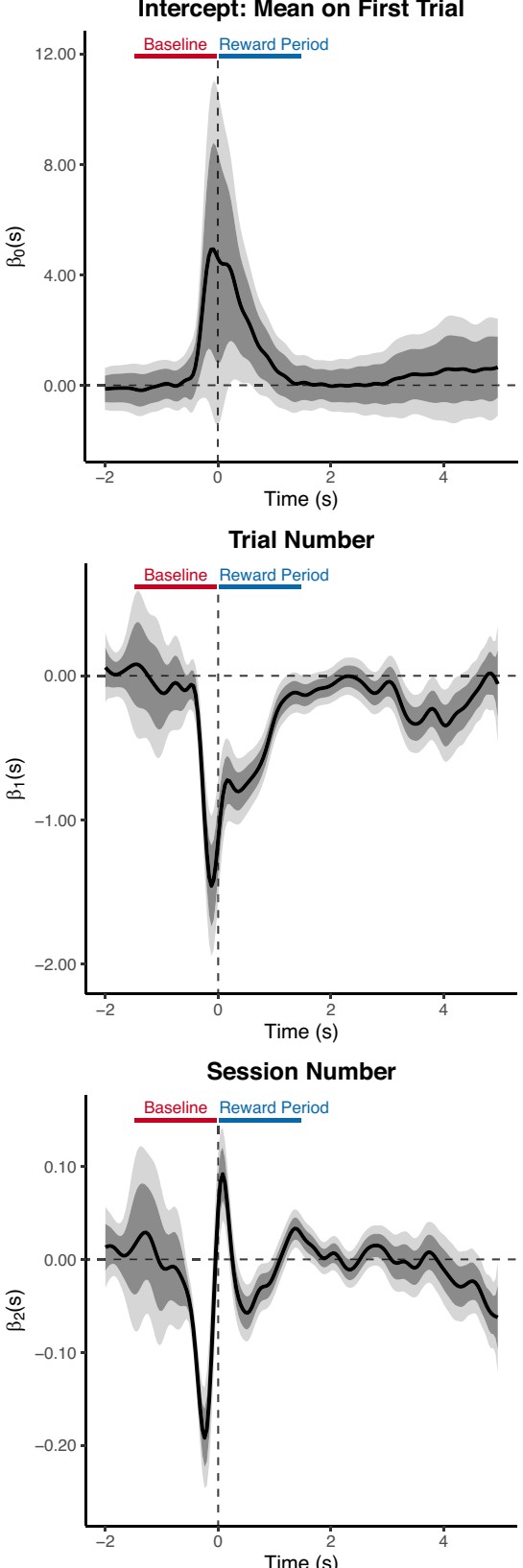

**Appendix 4—figure 7.** Background reward experiment analyses.

## 4.4 Using *FLMM* to test signal changes within- and across-trials

We show how *FLMM* enables hypothesis testing of signal changes within- and across-trials. In the experiment in section 'Tests 9–11' of *Jeong et al., 2022*, the authors test whether, across several sessions of Pavlovian learning, DA activity 'backpropagates' from reward delivery (3 sec after cue-onset) to the presentation of reward-predictive cues. They analyzed a summary measure defined as the difference between the average signal during pre-reward-delivery ('Late') and cue-onset ('Early') time-windows.

We tested the 'backpropagation' question with a *FLMM* model with session binary indicators as covariates, similar to a functional repeated measures ANOVA. This yields estimates of mean signal changes, at each time-point, between pairs of sessions. We did not observe significant 'Late' period changes, consistent with the authors' findings (*Appendix 4—figure 8*; see *Appendix 4—figure 9* for individual-animal fits). This analysis likely cannot be used to definitively rule out the existence of the 'backpropagation' phenomenon, but emphasizes how *FLMM* is well-suited to answer these types of questions. Nevertheless, we identified an additional effect that would be hard to find with summary measure analyses: the average (peak) size of cue-elicited DA, exhibited later in training, is similar to the degree that reward-delivery DA decreased. This is evident by comparing the symmetry between the magnitudes of peak increases during the 'Early' period and peak decreases during the post–'Late' period on the last session (*Appendix 4—figure 8*). This illustrates the capability in *FLMM* of directly testing effects visible in graphs, instead of having to perform hypothesis tests on a summary-of-summaries (e.g. a ratio or difference of AUCs).

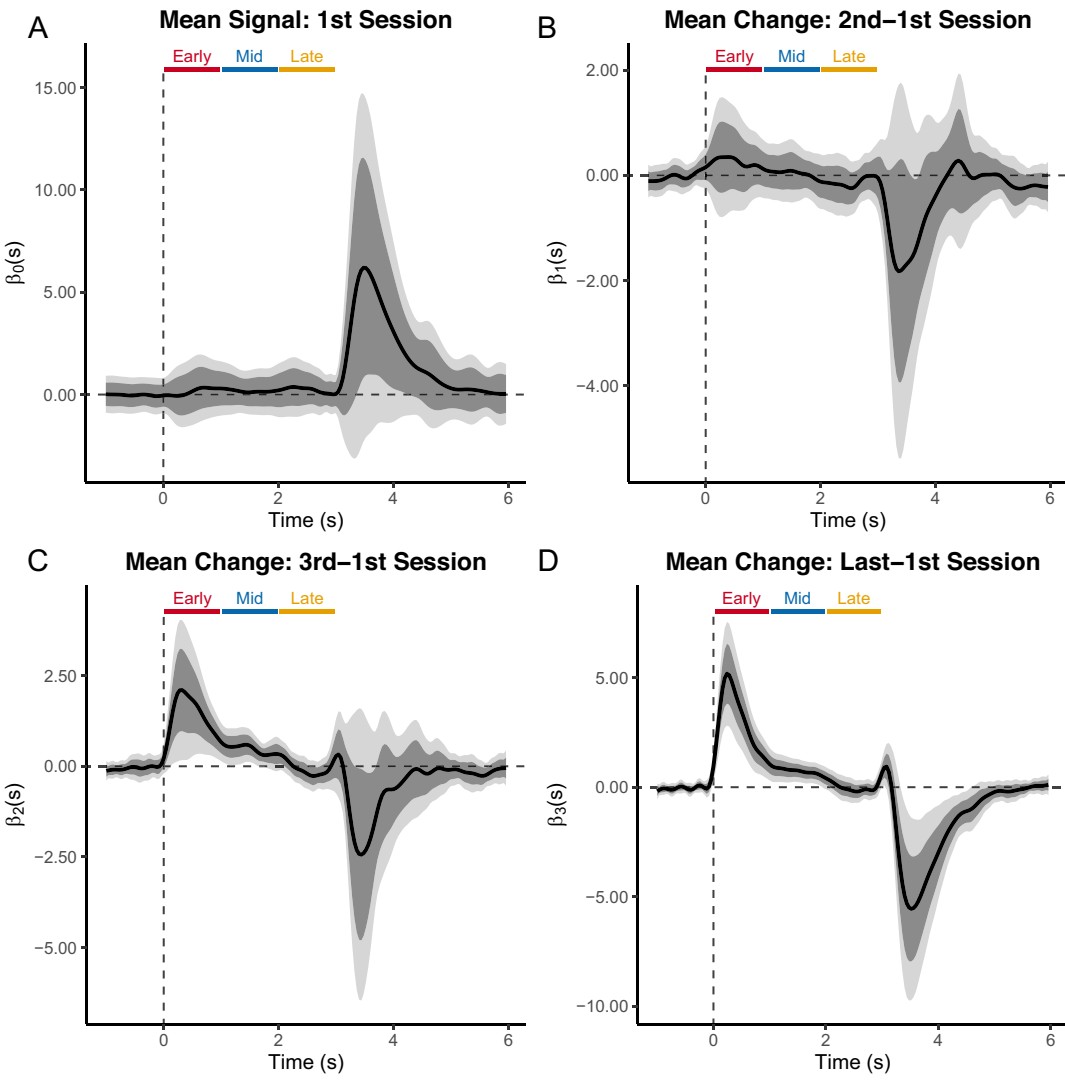

**Appendix 4—figure 8.** Functional Linear Mixed Models (*FLMM*) identifies how the signal evolves across trial time-points, and how the temporal location of transients progresses across sessions in a statistically significant manner. The panels show coefficient estimates from *FLMM* analyses of the 'backpropagation' experiment. Panel (**A**) contains the intercept term plot corresponding to the average signal on the first session of training. The 'Early' (0–1 sec), 'Mid' (1–2 sec), and 'Late' (2–3 sec) bars show the trial time-periods that the original authors used to calculate summary Area Under the Curve (AUC) measures. Trials are aligned to cue onset (cues lasted 2 sec) and rewards were delivered at 3 sec. Panels (**B**-**D**) show the coefficient estimates corresponding to the mean change in signal values from the second, third, or, fourth sessions, respectively, *compared to the first session* (positive values indicate an increase from the first session). Plots are aligned to cue onset.

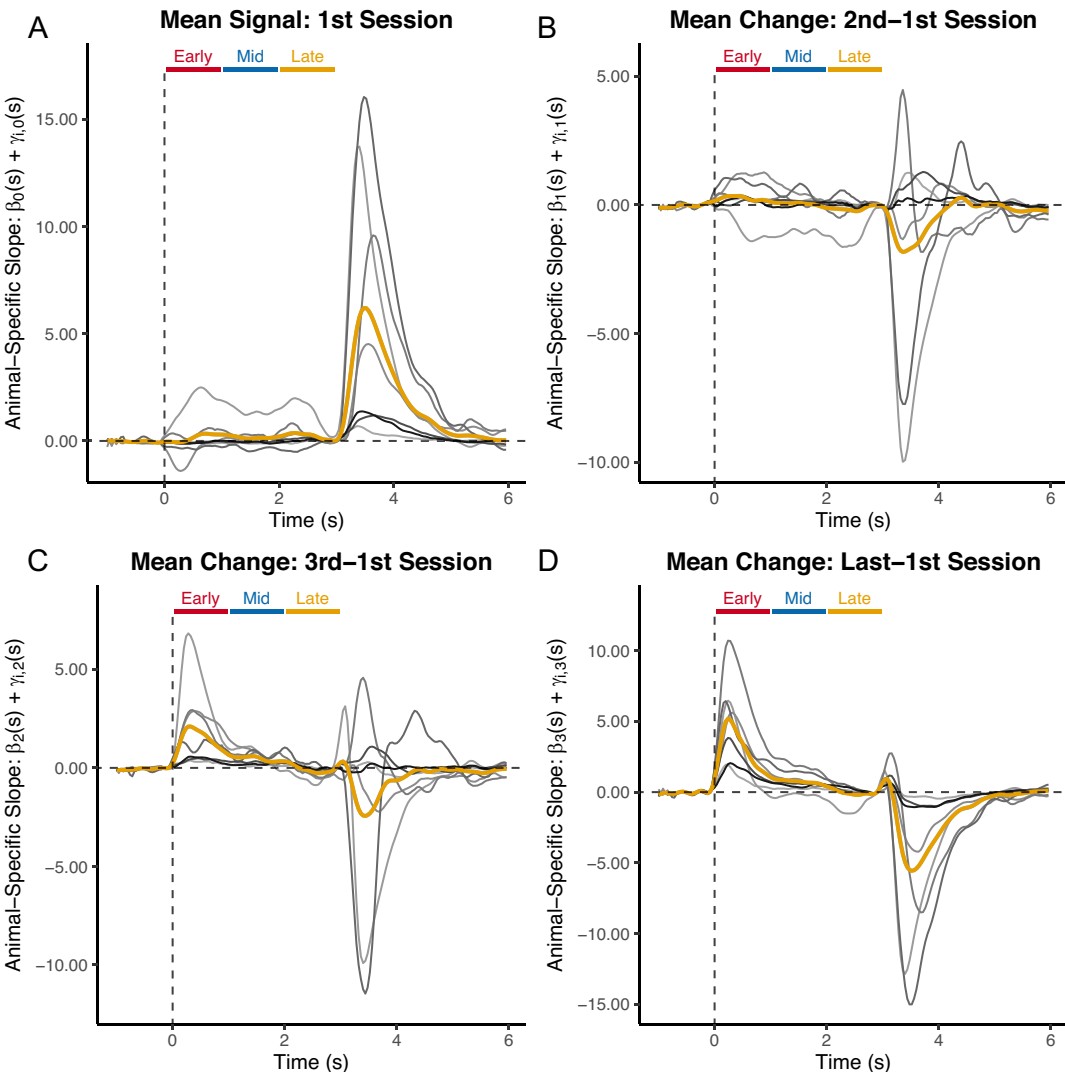

**Appendix 4—figure 9.** Individual-level coefficient estimates from functional linear mixed model (*FLMM)* analyses of the 'backpropagation' experiment: gold lines indicate the fixed-effect estimates and gray lines indicate animal-specific functional random-effect estimates (Best Linear Unbiased Predictor). Panel (**A**) contains the intercept term plot where the title provides an interpretation: the average signal on the first session of training. The 'Early' (0–1 sec), 'Mid' (1–2 sec), and 'Late'' (2–3 sec) bars show the trial time-periods that the original authors used to calculate summary Area Under the Curve (AUC) measures. Trials are aligned to cue onset and rewards were delivered at 3 sec. Panels (**B**-**D**) show the coefficient estimates which are interpreted as the change in mean signal from the second, third, or fourth sessions, respectively, compared to the first session (positive values indicate an increase from the first session). Plots are aligned to cue onset.

## 4.4.1 Reanalysis methods: Using *FLMM* to test signal changes within- and across-trials

We reanalyzed data presented in the section 'Tests 9–11 (Backpropagation within a trial)' of *Jeong et al., 2022*. We sought to evaluate the author's hypothesis described in the following paragraph in the main text:

"[in the] TDRL RPE account (...) dopamine responses drive value learning of the immediately preceding state. We tested three predictions of this central postulate that are each inconsistent with ANCCR. The first is that during the acquisition of trace conditioning, dopamine response systematically back propagates from the moment mmediately prior to reward to the cue onset (50) (Test 9, Fig. 6A). Unlike TDRL RPE, ANCCR does not make such a prediction since delay periods are not broken into states in ANCCR (...) Our observations were not consistent with a backpropagating

bump of activity and were instead consistent with an increase in cue response over trials of learning (Fig. 6B)"

We analyzed data from sessions 1–3, and the final session for each animal, as these were the sessions where we noticed the greatest changes. We fit a random slope model using indicator variables of the first, second, third, and subject-specific final sessions as covariates. We discarded data from other sessions and thus the interpretation of the intercept is the mean signal on the first session. Our final random slope model was

$$\mathbb{E}\left[Y_{i,l}(s) \mid \mathbf{X}_{i,l}, \mathbf{Z}_{i,l}, \boldsymbol{\gamma}_i(s)\right] = \beta_0(s) + \gamma_{0,i}(s) + \mathbb{1}(l = 2)\left[\beta_1(s) + \gamma_{1,i}(s)\right]$$

$$+ \mathbb{1}(l = 3)\left[\beta_2(s) + \gamma_{2,i}(s)\right] + \mathbb{1}(l = \mathsf{S}_i)\left[\beta_3(s) + \gamma_{3,i}(s)\right].$$

where $l$ denotes the session number, $\mathsf{S}_i$ is subject $i$'s final session (which can differ between animals), and $\mathbb{1}(l = s)$ is an indicator variable for session $s$. We show the code from our package to fit the above model (note that the period variable below is a factor variable):

```
model_fit = fui(photometry ~ period + (period | id), data=photometry_data)
```

## Appendix 5

### Simulations

#### 5.1 Simulation scheme
As described in the main text, we simulated data from the model

$$Y_{i,j}(s) = \beta_0(s) + \gamma_{0,i}(s) + \texttt{Delay}_{i,j}\left[\beta_1(s) + \gamma_{1,i}(s)\right] + \epsilon_{i,j}(s).$$
(12)

We take the simulated $\beta(s)$ in **Appendix 4.7** to be equal to the estimated coefficients from the model above fit to the real data in *Jeong et al., 2022*. To 'simulate' the covariates (i.e. just the $\texttt{Delay}$ indicator vector), we randomly draw a subset of animal IDs and concatenate all of their *observed* covariates from the (*Jeong et al., 2022*). That is for each simulation replicate, $r$, of sample size $n \in \{4, 5, ..., 8\}$, we randomly draw a sample of $n$ animal subject IDs, denoted as $n_r$ from the set $\{1, 2, ..., 7\}$ with uniform probability without replacement. For each animal ID in the sample, $n_r$, we concatenate the covariates (i.e. the design matrix) of the corresponding subjects in the *observed* data to be the 'simulated' covariates. That is, the design matrix $\mathbb{X}_r$ for simulation replicate $r$, is the row concatenation of each $\mathbb{X}_i$ for $i \in n_r$.

#### 5.2 Additional simulation results
To explore how analyzing summary measures can drown-out effects, we compared method performances on the same analyses across different cue period lengths (2 sec, 2.5 sec, or 3 sec from cue onset), which we visualize in *Appendix 5—figure 1A*. These relatively small adjustments substantially influenced estimation performance (see *Appendix 5—figure 1B*), and statistical power (see *Appendix 5—figures 1D and 2*). The pointwise 95 % CI coverage (see *Appendix 5—figure 1C*) of the summary measure approaches (t-test and LMM) is also very sensitive to the specified length of the cue period because they analyze a summary that averages over a time-window that contains heterogeneous effects. However, *FLMM* and *Perm* CI coverages (averaged over the cue period) remain stable to differing time-window lengths, because they evaluate each time-point. Notably, *Perm* exhibits low coverage at smaller sample sizes and the t-test yields poor coverage in all settings tested, likely because of the animal-to-animal variability in the signal magnitude and Delay Length change effect (the data were simulated from a LMM that specifies independence between observations conditional on random-effects. Since the t-test does not include random-effects, it may not achieve the nominal coverage because it relies on a different conditional independence assumption). Finally, *Appendix 5—figure 3* shows *FLMM* fits in around 10 seconds for datasets with about 800 trials (pooled across animals). Taken together, these data-driven simulations demonstrate that at low sample sizes, and in the presence of individual-differences, the *FLMM* consistently (1) achieves roughly nominal *pointwise* and *joint* coverage, (2) improves statistical power, and (3) can be fit quickly.

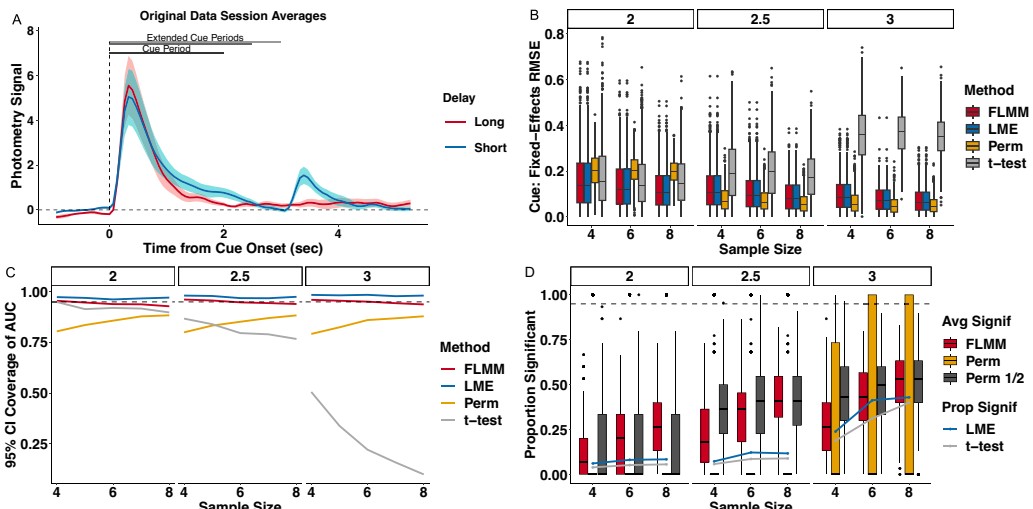

**Appendix 5—figure 1.** Summary measure analyses are highly sensitive to minor changes in the summary time-window. (**A**) Average short/long-delay data. Bars show cue period length used in (2 sec) and 'extended' delays analyzed in additional simulations. (**B**) Estimation error (RMSE), $\frac{1}{\sqrt{R}}\left\|\bar{\boldsymbol{\beta}}_1(\boldsymbol{s}) - \widehat{\bar{\boldsymbol{\beta}}}_1(\boldsymbol{s})\right\|_2$ where

$\bar{\beta}_1(s) = \frac{1}{|\mathcal{S}|}\sum_{s\in\mathcal{S}}\beta_k(s)$, associated with a mean difference during the cue periods (panels) and $n$ on the x-axis. Lower numbers indicate more accurate estimates. (**C**) pointwise 95% CI coverage is associated with a mean difference during cue period (panels) and $n$ on the x-axis. Higher values indicate better CI coverage. (**D**) Statistical power during cue period. The linear mixed model (LMM) and t-test were fit on the signal averaged over the cue period and thus each simulation replicate yields a single indicator of CI inclusion or statistical significance, which we represent with a line plot. For other methods, estimates are provided at each time-point, and performance is averaged across the time-points. We summarize these simulation replicate-specific averages with a boxplot.

Here, we define the estimation error (RMSE) of statistical methods for the average difference in photometry signal amplitude during the cue period as a function of how long that cue period was (2 sec, 2.5 sec, 3 sec). The error was defined as $\frac{1}{\sqrt{R}}\left\|\bar{\beta}_1(s) - \hat{\bar{\beta}}_1(s)\right\|_2$ where $\bar{\beta}_1(s) = \frac{1}{|\mathcal{S}|}\sum_{s\in\mathcal{S}}\beta_1(s)$. That is, the average of the coefficients for covariate $k$ across time-points $s$ in a fixed interval (indexed by $\mathcal{S}$). Since the model only contains a single binary covariate, we can compare the average of functional coefficients for the slope parameter in a *FLMM* model with the slope coefficient estimate in a scalar LMM applied on the outcome: $\bar{Y}_{i,j} = \frac{1}{|\mathcal{S}|}\sum_{s\in\mathcal{S}}Y_{i,j}(s)$ and a paired-samples t-test using the outcome: $\bar{Y}_i = \frac{1}{|\mathcal{J}_i|}\frac{1}{|\mathcal{S}|}\sum_{s\in\mathcal{S}}\sum_{j\in\mathcal{J}_i}Y_{i,j}(s)$, where $\mathcal{J}_i$ is the set of trials observed for subject $i$.

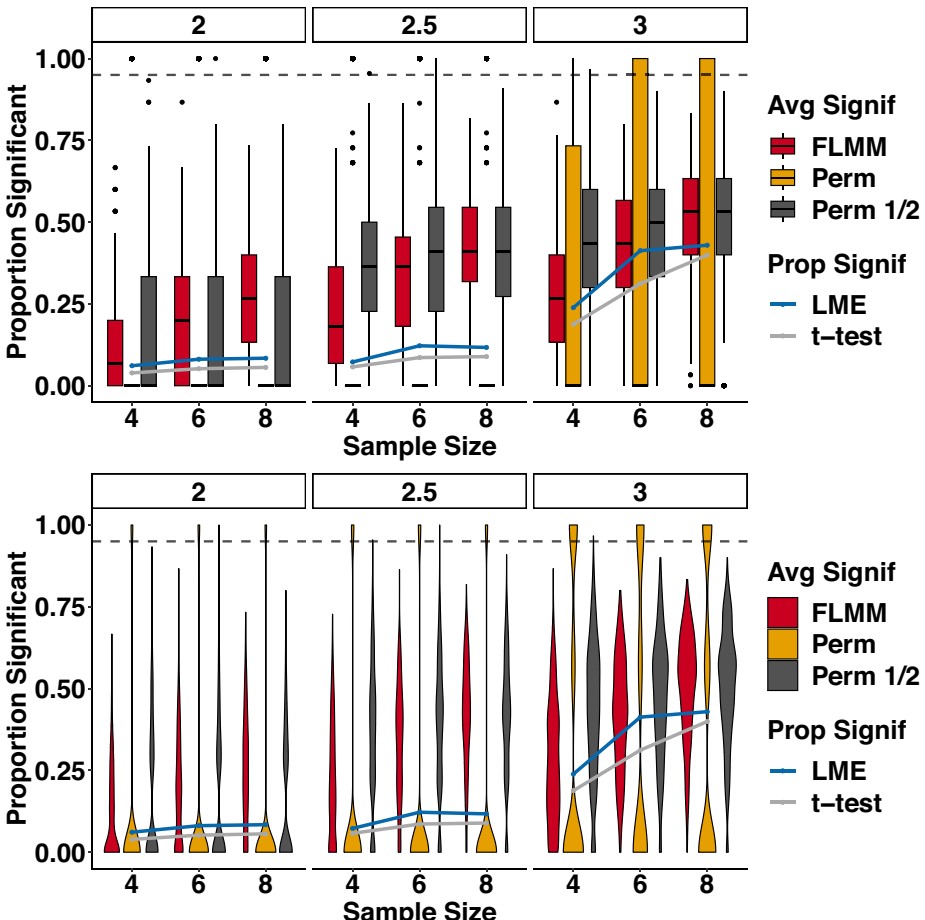

**Appendix 5—figure 2.** Statistical power associated with mean difference defining the cue period as 2 sec, 2.5 sec, and 3 sec (panels) and sample sizes (numbers of animals) on the x-axis. The two panels presents the same data in either violin or boxplot forms. Higher numbers indicate better power. For functional linear mixed model (*FLMM*) and *Perm*, power is averaged across the time-points in the cue period whereas the others assess the power using the average signal (across the cue period) as the outcome. Since each simulation replicate takes the proportion of significant time-points in the cue period for *FLMM* and *Perm*, these are presented as boxplots (or violin plots), *Appendix 5—figure 2 continued on next page*

*Appendix 5—figure 2 continued*

whereas the rest are simply presented as the proportion of simulation replicates that identified the mean signal during the cue as statistically significant (either 0 or 1 for each replicate).

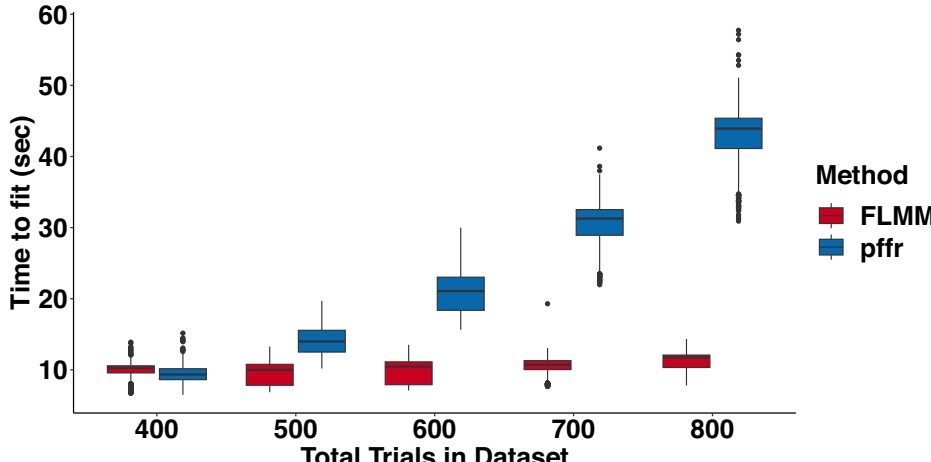

**Appendix 5—figure 3.** Time to fit functional linear mixed model (*FLMM*) fit with our software (using a closed-form variance calculation) on simulated data (each data point represents one replicate). pffr shows the time to fit the functional linear mixed model (with the same model specification) with the `pffr()` function in the `refund` package. Number of animals in simulations shown in plots ranges from 4 to 8 (i.e. $n = trials/100$).

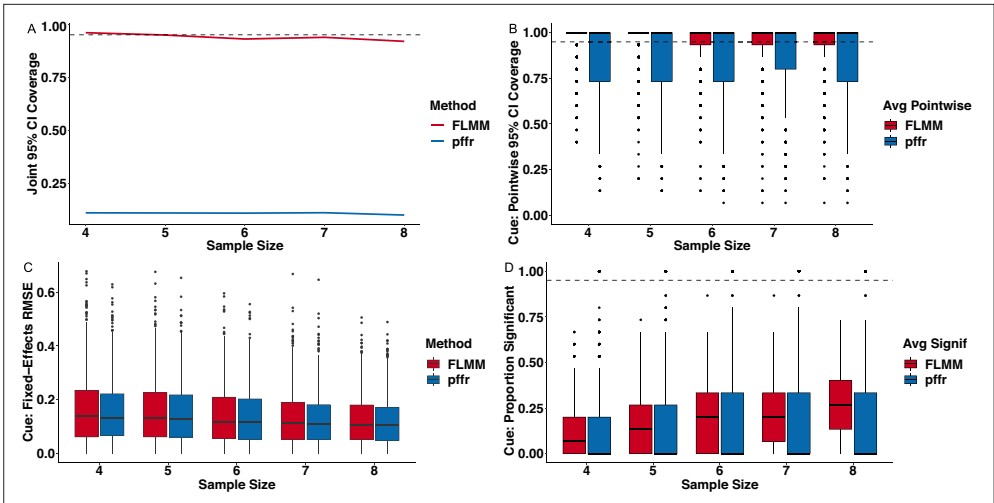

**Appendix 5—figure 4.** Functional Linear Mixed Models (*FLMM*) fit with our software achieves comparable or superior performance to the functional linear mixed model (with the same model specification) fit with the `pffr()` function in the `refund` package. (**A**) *FLMM* achieves *joint* 95% CI coverage at roughly the nominal level. `pffr` does not provide *joint* 95% CIs and thus the *pointwise* 95% CIs that it does provide achieve low *joint* coverage. (**B**) *FLMM* achieves *pointwise* 95% CI coverage at or above the nominal level. The *pointwise* 95% CI coverage of `pffr` is close to but below the nominal level. *pointwise* 95% CI coverage associated with mean difference during cue period (panels) and $n$ on the x-axis. Higher values indicate better CI coverage. (**C**) *FLMM* and `pffr` exhibit comparable fixed-effects estimation performance. Estimation error (RMSE), $\frac{1}{\sqrt{R}} \left\| \bar{\beta}_1(s) - \hat{\bar{\beta}}_1(s) \right\|_2$ where $\bar{\beta}_1(s) = \frac{1}{|\mathcal{S}|} \sum_{s \in \mathcal{S}} \beta_k(s)$, associated with mean difference during the cue periods (panels) and $n$ on the x-axis. Lower numbers indicate more accurate estimates. (**D**) *FLMM* exhibits superior statistical power compared to `pffr` during the cue period. (**B**-**D**) Since estimates are provided at each timepoint for both methods, pointwise performance is averaged across the time-points. We summarize these simulation replicate-specific averages as one point in a boxplot.

## Appendix 6

### Additional reanalyses

We include analyses here conducted on a second recent article *Coddington et al., 2023*; *Dudman, 2023* proposing a new reinforcement learning model for the role of mesolimbic dopamine in learning.

### 6.1 Additional reanalyses results

### 6.1.1 Functional methods allow for testing how signal 'dynamics' early in training predict behavior later in training

We next analyze data from a second paper focused on the role of mesolimbic DA in reward learning (*Coddington et al., 2023*). We first examine how between-animal differences in behavior correlate with average nucleus accumbens dopamine neuron calcium (NAc–DA) changes (DAT-Cre::ai32 transgenic mice were injected with a Cre-dependent jRCaMP1b virus across the ventral midbrain enabling the measurement of calcium dynamics in mesolimbic DA cells specifically). In this experiment, mice were exposed to a 0.5 sec stimulus, followed by reward 1 sec after cue-offset. The authors identified significant correlations between average Reward period NAc–DA (trial-averaged on *early* training sessions) with measures of average behavior (trial-averaged on *late* sessions).

We fit two analogous univariate models with the average behavioral measures as the covariate (cases without repeated observations on the same animal are equivalent to a *functional* linear regression without mixed effects).

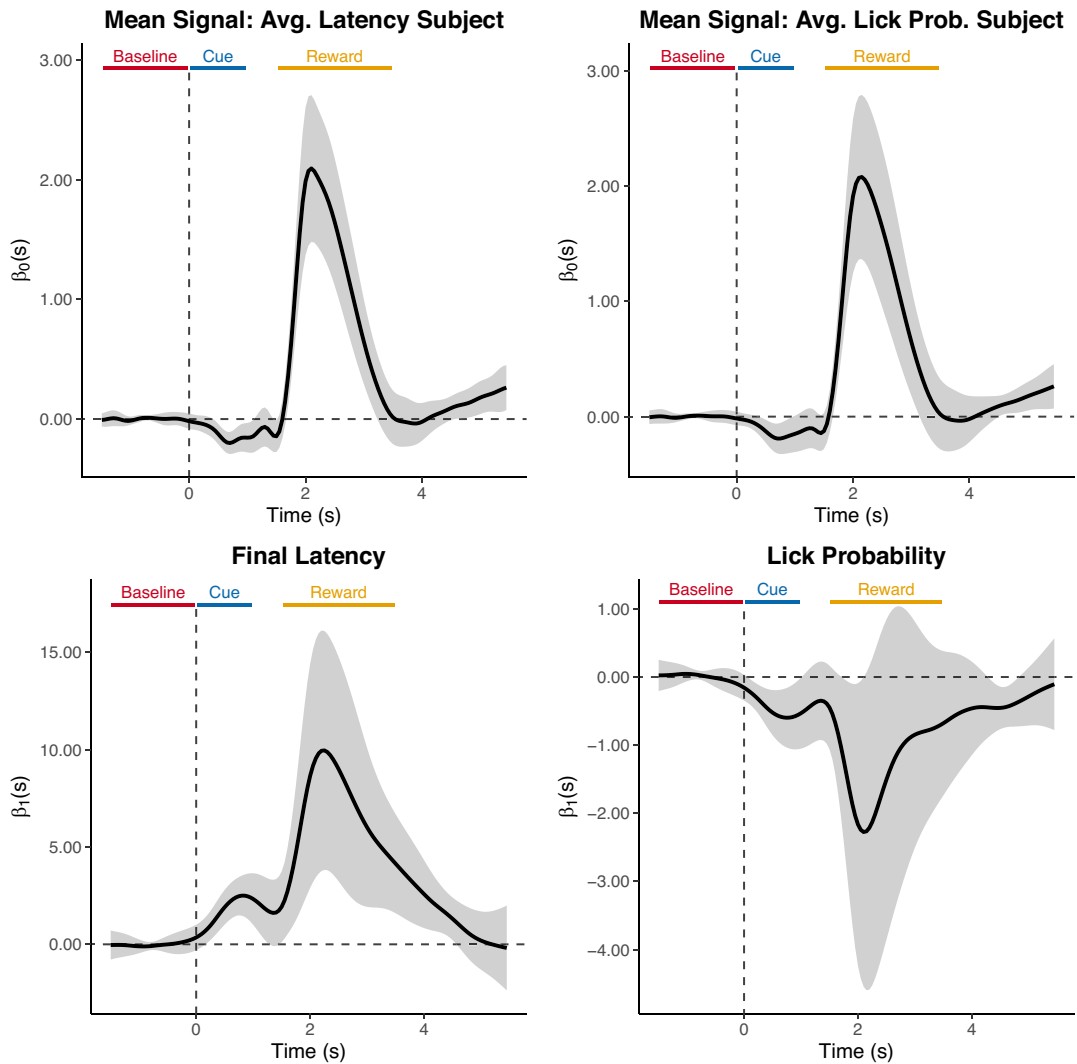

**Appendix 6—figure 1.** Coefficient estimates from Functional Linear Mixed Models (*FLMM*) analyses of `Final Latency` and `Lick Probability` models. The top row contains intercept term plots where the title provides interpretation of the intercept: the average dopamine neuron calcium (NAc-DA) signal on trials when the covariates in the model are at their average value. The bottom row shows the coefficient estimate plot of the covariate in the model. (Left) Association between average NAc-DA (averaged over first 100 trials) and latency to lick `Final Latency` (averaged over trials 700–800). (Right) Association between average NAc-DA (averaged over first 100 trials) lick probability (averaged over trials 700–800).

Our results reveal details missed in the summary measure analyses. Consistent with the author's findings, we found Reward period NAc–DA was associated with `Final Latency`. However, our analyses also revealed significant associations during the Cue period, an effect hard to see on average traces: while the average signal increases substantially during the Reward period, it exhibits comparatively little change during the Cue period. This demonstrates the difficulty of constructing summary measures: the time-windows when the signal is associated with covariates may not align with time-periods when the average signal exhibits noticeable changes. Remarkably, the association sign differs from the direction of average signal change: `Final Latency` is *positively* associated with NAc-DA during both Cue and Reward periods, yet the mean signal *decreases* during the Cue Period and *increases* during the Reward period.

### 6.1.2 *FLMM* provides test of how signal differences *between* trial-types change *across* training

Our final example demonstrates how questions that might otherwise require analysis of summaries of summary measures (e.g. ratios of average AUCs), can be precisely specified in a functional model so as to provide greater detail. Specifically, the authors sought to investigate how the difference in NAc-DA signals between trials with (Lick+) and without (Lick-) preparatory licking changed with learning. They reported a significant correlation between reward collection latency (`Final Latency`) and a "ratio of NAc–DA reward signals [AUCs] on Lick- vs Lick+ trials." We fit an *FLMM* model with an interaction between `Final Latency` and `Lick State` (a Lick-/Lick+ indicator). The interaction term provides a hypothesis test of their question at *each trial time-point*. The longest portion of joint significance occurs *between* Cue and Reward periods. Our analysis confirms the authors' results and adds detail obscured by standard methods: the interaction does not reach joint statistical significance until a couple seconds after reward-delivery, well-beyond the time when the average signal has fallen from its peak (shown by the intercept). This example highlights the challenge with constructing an adequate summary measure for this analysis: the period of clearest effect of the `Lick State` is before the average signal is largest. This may explain why the authors did not construct a summary measure during the interval between the Cue and Reward periods, when the `Lick State` effect was strongest. In sum, this example demonstrates how summary-of-summary measure analyses can be translated into simple *FLMM* regressions that provide greater detail about the time-course and magnitude of the effects throughout the trial.

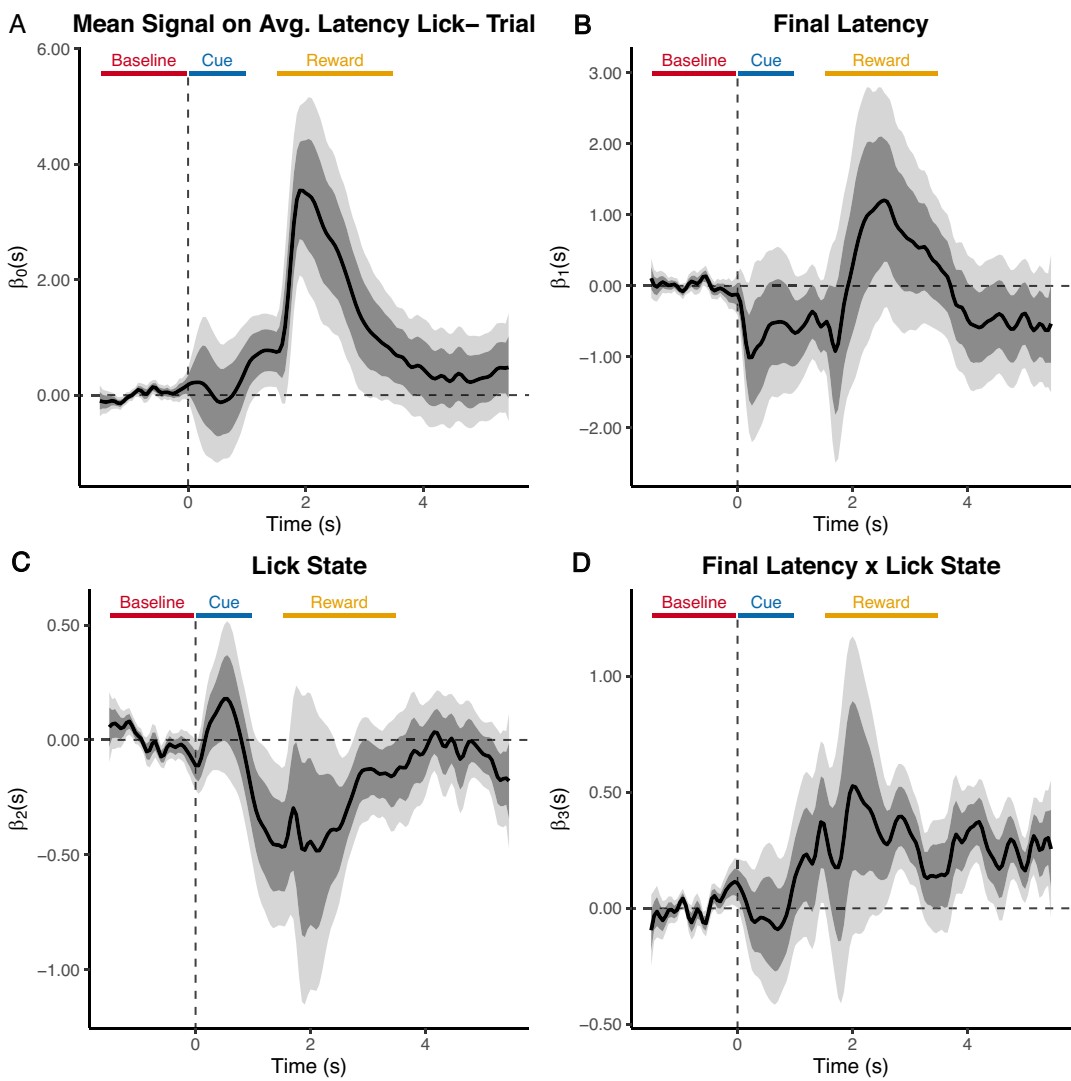

**Appendix 6—figure 2.** Coefficient estimates from a single functional linear mixed model (*FLMM*) analysis of `Final Latency × Lick State` interaction model (including main effects). A simple *FLMM* model enables characterization of how the association between photometry signals and behavioral responding (`Final Latency`) differs between conditions (Lick+/Lick-) at each time-point in the trial. Panel (**A**) contains the intercept term plot where the title provides an interpretation: the average photometry signal on Lick- trials for animals that exhibit average `Final Latency` values. Panels (**B**-**D**) show the three covariates of main effects and interaction of `Lick State` and `Final Latency`. The `Final Latency` functional coefficient is interpreted as the effect of `Final Latency` on Lick- trials. The `Lick State` main effect is interpreted as the difference in average NAc-DA between Lick+ and Lick- trials for an animal with an average `Final Latency` value. The interaction is interpreted as a difference in differences: during the Reward period, a 1 standard deviation increase in average `Final Latency` is associated with pointwise significantly higher NAc–DA signals on Lick+ than on Lick- trials (with a portion of joint significance) during most of the Reward period.

## 6.2 Additional reanalyses methods

### 6.2.1 Methods for Appendix 6.1.1

This paper measured photometry signals in which DAT-Cre::ai32 transgenic mice were injected with a Cre-dependent jRCaMP1b virus across the ventral midbrain enabling the measurement of calcium dynamics in mesolimbic DA cells specifically. We sought to conduct an analysis based upon the following quote:

"Unexpectedly, initial NAc–DA reward signals were negatively correlated with the amount of preparatory behaviour at the end of raining (NAc–DA reward trials 1-100 versus preparatory index

trials 700-800, r=-0.85, P=0.004), as well as the speed of reward collection (NAc-DA reward trials 1-100 versus reward collection latency trials 700-800, r=0.81, P=0.008)."

Preparatory licking and latency-to-lick were used as indicators of learning.

Since there is no way to meaningfully pair the neural activity of one trial with behavior on a separate trial, analyzing trial-level data is not appropriate and we therefore modeled the *average* photometry signal (averaged across trials 1–100) as a function of *average* behavior (averaged across trials 700–800). We removed trials with latencies over 1 second as they constituted behavioral outliers comprising less than 1% of trials.

To conduct a functional regression most analogous to the Pearson correlations the authors conducted, we fit the univariate linear regression models for final latency, $\mathrm{FL}_i$,

$$\mathbb{E}\left[\overline{Y}_i(s) \mid \mathbf{X}_i\right] = \beta_0(s) + \mathrm{FL}_i\beta_1(s).$$

Similarly, we fit the lick probability, $\mathrm{LP}_i$, functional regression model,

$$\mathbb{E}\left[\overline{Y}_i(s) \mid \mathbf{X}_i\right] = \beta_0(s) + \mathrm{LP}_i\beta_1(s).$$

These models were fit with the `R` package `refund` with the `fosr()` function (where fosr abbreviates function-on-scalar regression used for a functional outcome and scalar covariate). Note that this package function only provides *pointwise* 95% CIs and thus the plots in the main text only have one shade of gray for the 95% CIs. Thus the interpretation for this analysis is confined to a *pointwise* interpretation.

### 6.2.2 Methods for Appendix 6.1.2

We sought to conduct an analysis based upon the following quote: "[the proposed] scheme also predicts that [mesolimbic DA] reward signals should reflect the evolution of reward collection policy across learning…Indeed,…mouse data exhibited differential reward responses on trials with ('Lick+') or without ('Lick-') preparatory licking as learning progressed" (*Coddington et al., 2023*). The authors reported (see the Figure 8d caption, *Coddington et al., 2023*) a significant Pearson correlation between the final reward collection latency (`Final Latency`) and a "ratio of NAc–DA reward signals on Lick- vs Lick+ trials." They constructed this measure by first summarizing the reward-period NAc-DA (with AUC), averaged across Lick+ and Lick- trials separately, and then calculating a ratio for each animal. Similar to the analysis we presented in the previous section, the authors compared behavior in one set of trials to dopamine activity in a separate set of trials. Because there is no way to meaningfully pair the neural activity of one trial with behavior on a separate trial, we analyzed the *average* photometry signal (Lick+/Lick- separately averaged over the trials specified in the paper) as a function of *average* behavior.

Specifically, we modeled the trial-averaged signal denoted as $\overline{Y}_{i,v}(s)$ for animal $i$ and lick state $v$. The signal was trial-averaged separately across Lick+ and Lick- trials. We both indicate lick state with the subscript $v$ on $\overline{Y}_{i,v}(s)$ and also as a covariate in the model where $\mathrm{LS}_{i,v} = 1$ for Lick+ trials and $\mathrm{LS}_{i,v} = 0$ for Lick- trials. We denote the final latency below as $\mathrm{FL}_i$. We fit the random slope *FLMM* model,

$$\mathbb{E}\left[\overline{Y}_{i,v}(s) \mid \mathbf{X}_{i,v}, \mathbf{Z}_i, \gamma_i(s)\right] = \beta_0(s) + \gamma_{0,i}(s) + \mathrm{FL}_i\left[\beta_1(s) + \gamma_{1,i}(s)\right] + \mathrm{LS}_{i,v}\beta_2(s) + \mathrm{FL}_i \times \mathrm{LS}_{i,v}\beta_3(s).$$

Given that the outcome was trial-averaged, the number of observations of the outcome was only twice the number of animals (i.e. one observation of the functional outcome for Lick+ and one for Lick-). We were thus limited in the number of random-effects we could include and still retain an identifiable model. For that reason, our model did not include a random intercept. As for all other analyses in our manuscript, we compared multiple candidate models that we specified based upon the author's analyses. We then selected the model with the best AIC/BIC model fit criteria.

We normalized the (average) `Final Latency` variable (across trials) to have mean 0 and unit variance. The intercept, therefore, shows that mean NAc–DA activity for a mouse with average `Final Latency` on Lick- trials is *jointly* significantly elevated during portions of the Reward period. The `Final Latency` functional coefficient is interpreted as the effect of `Final Latency` on Lick-trials and is only briefly pointwise significant in the middle of the Reward period. The `Lick State` main effect is interpreted as the difference in average NAc–DA between Lick+ and Lick- trials for an animal with an average `Final Latency` value. The interaction is interpreted as a difference in

differences: during the Reward period, a 1 standard deviation increase in average `Final Latency` is associated with pointwise significantly higher NAc–DA signals on Lick+ than on Lick- trials (with a portion of joint significance) during most of the Reward period.

