## [Editor Report · eLife Assessment]

This **important** study presents a statistical framework for the analysis of photometry signals and provides an open-source implementation. The evidence supporting the benefits of the presented functional mixed-effect modeling analysis as opposed to (1) summary statistics and (2) other pointwise regression models is **convincing** with a thorough comparison with other methods and datasets. This work will be of great interest to researchers using not only fiber photometry, but other time-series data such as calcium imaging or electrophysiology data, and wanting to implement trial-by-trial temporal analysis, taking also into account variability within the dataset.

---

## [Referee Report · Reviewer #1 (Public review)]

Summary:

Fiber photometry has become a very popular tool in recording neuronal activity in freely behaving animals. Despite the number of papers published with the method, as the authors rightly note, there are currently no standardized ways to analyze the data produced. Moreover, most of the data analyses confine to simple measurements of averaged activity and by doing so, erase valuable information encoded in the data. The authors offer an approach based on functional linear mixed modeling, where beyond changes in overall activity various functions of the data can also be analyzed. More in depth analysis, more variables taken into account, better statistical power all lead to higher quality science.

Strengths:

The framework the authors present is solid and well explained. By reanalyzing formerly published data, the authors also further increase the significance of the proposed tool opening new avenues for reinterpreting already collected data. They also made a convincing case showing that the proposed algorithm works on data with different preprocessing backgrounds.

---

## [Referee Report · Reviewer #2 (Public review)]

Summary:

This work describes a statistical framework that combines functional linear mixed modeling with joint 95% confidence intervals, which improves statistical power and provides less conservative and more robust statistical inferences than in previous studies. Pointwise linear regression analysis has been used extensively to analyze time series signals from a wide range of neuroscience recording techniques, with recent studies applying them to photometry data. The novelty of this study lies in (1) the introduction of joint 95% confidence intervals for statistical testing of functional mixed models with nested random-effects, and (2) providing an open-source R package implementing this framework. This study also highlights how summary statistics as opposed to trial-by-trial analysis can obscure or even change the direction of statistical results by reanalyzing two other studies.

Strengths:

The open-source package in R using a similar syntax as lme4 package for the implementation of this framework, the high fitting speed and the low memory footprint, even in complex models, enhance the accessibility and usage by other researchers.

The reanalysis of two studies using summary statistics on photometry data (Jeong et al., 2022; Coddington et al., 2023) highlights how trial-by-trial analysis at each time-point on the trial can reveal information obscured by averaging across trials. Furthermore, this work also exemplifies how session and subject variability can lead to different conclusions when not considered.

This study also showcases the statistical robustness of FLMM by comparing this method to fitting pointwise linear mixed models and performing t-test and Benjamini-Hochberg correction as performed by Lee et al. (2019).

---

## [Referee Report · Reviewer #3 (Public review)]

Summary:

Loewinger et al. extend a previously described framework (Cui et al., 2021) to provide new methods for statistical analysis of fiber photometry data. The methodology combines functional regression with linear mixed models, allowing inference on complex study designs that are common in photometry studies. To demonstrate its utility, they reanalyze datasets from two recent fiber photometry studies into mesolimbic dopamine. Then, through simulation, they demonstrate the superiority of their approach compared to other common methods.

Strengths:

The statistical framework described provides a powerful way to analyze photometry data and potentially other similar signals. The provided package makes this methodology easy to implement and the extensively worked examples of reanalysis provide a useful guide to others on how to correctly specify models.

Modeling the entire trial (function regression) removes the need to choose appropriate summary statistics, removing the opportunity to introduce bias, for example in searching for optimal windows in which to calculate the AUC. This is demonstrated in the re-analysis of Jeong et al., 2022, in which the AUC measures presented masked important details about how the photometry signal was changing. There is an appropriate level of discussion of the interpretation of the reanalyzed data that highlights the pitfalls of other methods and the usefulness of their methods.

The authors' use of linear mixed methods, allows for the estimation of random effects, which are an important consideration given the repeated-measures design of most photometry studies.

The authors provide a useful guide for how to practically use and implement their methods in an easy-to-use package. These methods should have wide applicability to those who use photometry or similar methods. The development of this excellent open-source software is a great service to the wider neuroscience community.

---

## [Author Response]

The following is the authors’ response to the original reviews.

**eLife Assessment**
This important work presents a new methodology for the statistical analysis of fiber photometry data, improving statistical power while avoiding the bias inherent in the choices that are necessarily made when summarizing photometry data. The reanalysis of two recent photometry data sets, the simulations, and the mathematical detail provide convincing evidence for the utility of the method and the main conclusions, however, the discussion of the re-analyzed data is incomplete and would be improved by a deeper consideration of the limitations of the original data. In addition, consideration of other data sets and photometry methodologies including non-linear analysis tools, as well as a discussion of the importance of the data normalization are needed.

Thank you for reviewing our manuscript and giving us the opportunity to respond and improve our paper. In our revision, we have strived to address the points raised in the comments, and implement suggested changes where feasible. We have also improved our package and created an analysis guide (available on our Github - https://github.com/gloewing/fastFMM and https://github.com/gloewing/photometry_fGLMM), showing users how to apply our methods and interpret their results. Below, we provide a detailed point-by-point response to the reviewers.

**Reviewer #1:**
Summary:Fiber photometry has become a very popular tool in recording neuronal activity in freely behaving animals. Despite the number of papers published with the method, as the authors rightly note, there are currently no standardized ways to analyze the data produced. Moreover, most of the data analyses confine to simple measurements of averaged activity and by doing so, erase valuable information encoded in the data. The authors offer an approach based on functional linear mixed modeling, where beyond changes in overall activity various functions of the data can also be analyzed. More in-depth analysis, more variables taken into account, and better statistical power all lead to higher quality science.Strengths:The framework the authors present is solid and well-explained. By reanalyzing formerly published data, the authors also further increase the significance of the proposed tool opening new avenues for reinterpreting already collected data.

Thank you for your favorable and detailed description of our work!

Weaknesses:However, this also leads to several questions. The normalization method employed for raw fiber photometry data is different from lab to lab. This imposes a significant challenge to applying a single tool of analysis.

Thank you for these important suggestions. We agree that many data pre-processing steps will influence the statistical inference from our method. Note, though, that this would also be the case with standard analysis approaches (e.g., t-tests, correlations) applied to summary measures like AUCs. For that reason, we do not believe that variability in pre-processing is an impediment to widespread adoption of a standard analysis procedure. Rather, we would argue that the sensitivity of analysis results to pre-processing choices should motivate the development of statistical techniques that reduce the need for pre-processing, and properly account for structure in the data arising from experimental designs. For example, even without many standard pre-processing steps, *FLMM* provides smooth estimation results across trial timepoints (i.e., the 'functional domain'), has the ability to adjust for betweentrial and -animal heterogeneity, and provides a valid statistical inference framework that quantifies the resulting uncertainty. We appreciate the reviewer’s suggestion to emphasize and further elaborate on our method from this perspective. We have now included the following in the Discussion section:

*“FLMM* can help model signal components unrelated to the scientific question of interest, and provides a systematic framework to quantify the additional uncertainty from those modeling choices. For example, analysts sometimes normalize data with trial-specific baselines because longitudinal experiments can induce correlation patterns across trials that standard techniques (e.g., repeated measures ANOVA) may not adequately account for. Even without many standard data pre-processing steps, *FLMM* provides smooth estimation results across trial time-points (the 'functional domain'), has the ability to adjust for between-trial and -animal heterogeneity, and provides a valid statistical inference approach that quantifies the resulting uncertainty. For instance, session-to-session variability in signal magnitudes or dynamics (e.g., a decreasing baseline within-session from bleaching or satiation) could be accounted for, at least in part, through the inclusion of trial-level fixed or random effects. Similarly, signal heterogeneity due to subject characteristics (e.g., sex, CS+ cue identity) could be incorporated into a model through inclusion of animal-specific random effects. Inclusion of these effects would then influence the width of the confidence intervals. By expressing one’s 'beliefs' in an *FLMM* model specification, one can compare models (e.g., with AIC). Even the level of smoothing in *FLMM* is largely selected as a function of the data, and is accounted for directly in the equations used to construct confidence intervals. This stands in contrast to 'trying to clean up the data' with a pre-processing step that may have an unknown impact on the final statistical inferences.”

Does the method that the authors propose work similarly efficiently whether the data are normalized in a running average dF/F as it is described in the cited papers? For example, trace smoothing using running averages (Jeong et al. 2022) in itself may lead to pattern dilution.

By modeling trial signals as 'functions', the method accounts for and exploits correlation across trial timepoints and, as such, any pre-smoothing of the signals should not negatively affect the validity of the 95% CI coverage. It will, however, change inferential results and the interpretation of the data, but this is not unique to *FLMM*, or many other statistical procedures.

The same question applies if the z-score is calculated based on various responses or even baselines. How reliable the method is if the data are non-stationery and the baselines undergo major changes between separate trials?

Adjustment for trial-to-trial variability in signal magnitudes or dynamics could be accounted for, at least in part, through the inclusion of trial-level random effects. This heterogeneity would then influence the width of the confidence intervals, directly conveying the effect of the variability on the conclusions being drawn from the data. This stands in contrast to 'trying to clean up the data' with a pre-processing step that may have an unknown impact on the final statistical inferences. Indeed, non-stationarity (e.g., a decreasing baseline within-session) due to, for example, measurement artifacts (e.g., bleaching) or behavioral causes (e.g., satiation, learning) should, if possible, be accounted for in the model. As mentioned above, one can often achieve the same goals that motivate pre-processing steps by instead applying specific *FLMM* models (e.g., that include trial-specific intercepts to reflect changes in baseline) to the unprocessed data. One can then compare model criteria in an objective fashion (e.g., with AIC) and quantify the uncertainty associated with those modeling choices. Even the level of smoothing in *FLMM* is largely selected as a function of the data, and is accounted for directly in the equations used to construct confidence intervals. In sum, our method provides both a tool to account for challenges in the data, and a systematic framework to quantify the additional uncertainty that accompanies accounting for those data characteristics.

Finally, what is the rationale for not using non-linear analysis methods? Following the paper’s logic, non-linear analysis can capture more information that is diluted by linear methods.

This is a good question that we imagine many readers will be curious about as well. We have added in notes to the Discussion and Methods Section 4.3 to address this (copied below). We thank the reviewer for raising this point, as your feedback also motivated us to discuss this point in Part 5 of our Analysis Guide.

Methods

*“FLMM* models each trial’s signal as a function that varies smoothly across trial time-points (i.e., along the 'functional domain'). It is thus a type of non-linear modeling technique over the functional domain, since we do not assume a linear model (straight line). *FLMM* and other functional data analysis methods model data as functions, when there is a natural ordering (e.g., time-series data are ordered by time, imaging data are ordered by x-y coordinates), and are assumed to vary smoothly along the functional domain (e.g., one assumes values of a photometry signal at close time-points in a trial have similar values). Functional data analysis approaches exploit this smoothness and natural ordering to capture more information during estimation and inference.”

Discussion

“In this paper, we specified *FLMM* models with linear covariate–signal relationships *at a fixed trial time-point* across trials/sessions, to compare the *FLMM* analogue of the analyses conducted in (Jeong et al., 2022). However, our package allows modeling of covariate–signal relationships with non-linear functions of covariates, using splines or other basis functions. One must consider, however, the tradeoff between flexibility and interpretability when specifying potentially complex models, especially since *FLMM* is designed for statistical inference.”

**Reviewer #2:**
Summary:This work describes a statistical framework that combines functional linear mixed modeling with joint 95% confidence intervals, which improves statistical power and provides less conservative statistical inferences than in previous studies. As recently reviewed by Simpson et al. (2023), linear regression analysis has been used extensively to analyze time series signals from a wide range of neuroscience recording techniques, with recent studies applying them to photometry data. The novelty of this study lies in (1) the introduction of joint 95% confidence intervals for statistical testing of functional mixed models with nested random-effects, and (2) providing an open-source R package implementing this framework. This study also highlights how summary statistics as opposed to trial-by-trial analysis can obscure or even change the direction of statistical results by reanalyzing two other studies.Strengths:The open-source package in R using a similar syntax as the lme4 package for the implementation of this framework on photometry data enhances the accessibility, and usage by other researchers. Moreover, the decreased fitting time of the model in comparison with a similar package on simulated data, has the potential to be more easily adopted.The reanalysis of two studies using summary statistics on photometry data (Jeong et al., 2022; Coddington et al., 2023) highlights how trial-by-trial analysis at each time-point on the trial can reveal information obscured by averaging across trials. Furthermore, this work also exemplifies how session and subject variability can lead to opposite conclusions when not considered.

We appreciate the in-depth description of our work and, in particular, the R package. This is an area where we put a lot of effort, since our group is very concerned with the practical experience of users.

Weaknesses:Although this work has reanalyzed previous work that used summary statistics, it does not compare with other studies that use trial-by-trial photometry data across time-points in a trial. As described by the authors, fitting pointwise linear mixed models and performing t-test and BenjaminiHochberg correction as performed in Lee et al. (2019) has some caveats. Using joint confidence intervals has the potential to improve statistical robustness, however, this is not directly shown with temporal data in this work. Furthermore, it is unclear how FLMM differs from the pointwise linear mixed modeling used in this work.

Thank you for making this important point. We agree that this offers an opportunity to showcase the advantages of FLMM over non-functional data analysis methods, such as the approach applied in Lee et al. (2019). As mentioned in the text, fitting entirely separate models at each trial timepoint (without smoothing regression coefficient point and variance estimates across timepoints), and applying multiple comparisons corrections as a function of the number of time points has substantial conceptual drawbacks. To see why, consider that applying this strategy with two different sub-sampling rates requires adjustment for different numbers of comparisons, and could thus lead to very different proportions of timepoints achieving statistical significance. In light of your comments, we decided that it would be useful to provide a demonstration of this. To that effect, we have added Appendix Section 2 comparing FLMM with the method in Lee et al. (2019) on a real dataset, and show that FLMM yields far less conservative and more stable inference across different sub-sampling rates. We conducted this comparison on the delay-length experiment (shown in Figure 6) data, sub-sampled at evenly spaced intervals at a range of sampling rates. We fit either a collection of separate linear mixed models (LMM) followed by a Benjamini–Hochberg (BH) correction, or *FLMM* with statistical significance determined with both Pointwise and Joint 95% CIs. As shown in Appendix Tables 1-2, the proportion of timepoints at which effects are statistically significant with FLMM Joint CIs is fairly stable across sampling rates. In contrast, the percentage is highly inconsistent with the BH approach and is often highly conservative. This illustrates a core advantage of functional data analysis methods: borrowing strength across trial timepoints (i.e., the functional domain), can improve estimation efficiency and lower sensitivity to how the data is sub-sampled. A multiple comparisons correction may, however, yield stable results if one first smooths both regression coefficient point and variance estimates. Because this includes smoothing the coefficient point and variance estimates, this approach would essentially constitute a functional mixed model estimation strategy that uses multiple comparisons correction instead of a joint CI. We have now added in a description of this experiment in Section 2.4 (copied below).

“We further analyze this dataset in Appendix Section 2, to compare *FLMM* with the approach applied in Lee et al. (2019) of fitting *pointwise* LMMs (without any smoothing) and applying a Benjamini–Hochberg (BH) correction. Our hypothesis was that the Lee et al. (2019) approach would yield substantially different analysis results, depending on the sampling rate of the signal data (since the number of tests being corrected for is determined by the sampling rate). The proportion of timepoints at which effects are deemed statistically significant by *FLMM* joint 95% CIs is fairly stable across sampling rates. In contrast, that proportion is both inconsistent and often low (i.e., highly conservative) across sampling rates with the Lee et al. (2019) approach. These results illustrate the advantages of modeling a trial signal as a function, and conducting estimation and inference in a manner that uses information across the entire trial.”

In this work, FLMM usages included only one or two covariates. However, in complex behavioral experiments, where variables are correlated, more than two may be needed (see Simpson et al. (2023), Engelhard et al. (2019); Blanco-Pozo et al. (2024)). It is not clear from this work, how feasible computationally would be to fit such complex models, which would also include more complex random effects.

Thank you for bringing this up, as we endeavored to create code that is able to scale to complex models and large datasets. We agree that highlighting this capability in the paper will strengthen the work. We now state in the Discussion section that “[T]he package is fast and maintains a low memory footprint even for complex models (see Section 4.6 for an example) and relatively large datasets.” Methods Section 4.6 now includes the following:

Our fastFMM package scales to the dataset sizes and model specifications common in photometry. The majority of the analyses presented in the Results Section (Section 2) included fairly simple functional fixed and random effect model specifications because we were implementing the *FLMM* versions of the summary measure analyses presented in Jeong et al. (2022). However, we fit the following *FLMM* to demonstrate the scalability of our method with more complex model specifications:

E[Yi,j,l(s)|Xi,j,l,Zi,j,l,γi(s)]=β0(s)+γ0,i(s)+SNi,l[β1(s)+γ1,i(s)]+TNi,j,l[β2(s)+γ2,i(s)]+IRIi,j,l[β3(s)+γ3,i(s)]+Licki,j,l[β4(s)+γ4,i(s)]+TLi,j,l[β5(s)+γ5,i(s)].

We use the same notation as the Reward Number model in Section 4.5.2, with the additional variable TL_i,j,l_ denoting the Total Licks on trial *j* of session *l* for animal *i*. In a dataset with over 3,200 total trials (pooled across animals), this model took ∼1.2 min to fit on a MacBook Pro with an Apple M1 Max chip with 64GB of RAM. Model fitting had a low memory footprint. This can be fit with the code:

model_fit = fui(photometry ~ session + trial + iri + lick_time + licks + (session + trial + iri + lick_time + licks | id), parallel = TRUE, data = photometry_data).

This provides a simple illustration of the scalability of our method. The code (including timing) for this demonstration is now included on our Github repository.

**Reviewer #3:**
Summary:Loewinger et al., extend a previously described framework (Cui et al., 2021) to provide new methods for statistical analysis of fiber photometry data. The methodology combines functional regression with linear mixed models, allowing inference on complex study designs that are common in photometry studies. To demonstrate its utility, they reanalyze datasets from two recent fiber photometry studies into mesolimbic dopamine. Then, through simulation, they demonstrate the superiority of their approach compared to other common methods.Strengths:The statistical framework described provides a powerful way to analyze photometry data and potentially other similar signals. The provided package makes this methodology easy to implement and the extensively worked examples of reanalysis provide a useful guide to others on how to correctly specify models.Modeling the entire trial (function regression) removes the need to choose appropriate summary statistics, removing the opportunity to introduce bias, for example in searching for optimal windows in which to calculate the AUC. This is demonstrated in the re-analysis of Jeong et al., 2022, in which the AUC measures presented masked important details about how the photometry signal was changing.Meanwhile, using linear mixed methods allows for the estimation of random effects, which are an important consideration given the repeated-measures design of most photometry studies.

We would like to thank the reviewer for the deep reading and understanding of our paper and method, and the thoughtful feedback provided. We agree with this summary, and will respond in detail to all the concerns raised.

Weaknesses:While the availability of the software package (fastFMM), the provided code, and worked examples used in the paper are undoubtedly helpful to those wanting to use these methods, some concepts could be explained more thoroughly for a general neuroscience audience.

Thank you for this point. While we went to great effort to explain things clearly, our efforts to be concise likely resulted in some lack of clarity. To address this, we have created a series of analysis guides for a more general neuroscience audience, reflecting our experience working with researchers at the NIH and the broader community. These guides walk users through the code, its deployment in typical scenarios, and the interpretation of results.

While the methodology is sound and the discussion of its benefits is good, the interpretation and discussion of the re-analyzed results are poor:In section 2.3, the authors use FLMM to identify an instance of Simpson’s Paradox in the analysis of Jeong et al. (2022). While this phenomenon is evident in the original authors’ metrics (replotted in Figure 5A), FLMM provides a convenient method to identify these effects while illustrating the deficiencies of the original authors’ approach of concatenating a different number of sessions for each animal and ignoring potential within-session effects.

Our goal was to demonstrate that *FLMM* provides insight into why the opposing within- and between-session effects occur: the between-session and within-session changes appear to occur at different trial timepoints. Thus, while the AUC metrics applied in Jeong et al. (2022) are enough to show the *presence of* Simpson’s paradox, it is difficult to hypothesize *why* the opposing within-/between-session effects occur. An AUC analysis cannot determine at what trial timepoints (relative to licking) those opposing trends occur.

The discussion of this result is muddled. Having identified the paradox, there is some appropriate speculation as to what is causing these opposing effects, particularly the decrease in sessions. In the discussion and appendices, the authors identify (1) changes in satiation/habitation/motivation, (2) the predictability of the rewards (presumably by the click of a solenoid valve) and (3) photobleaching as potential explanations of the decrease within days. Having identified these effects, but without strong evidence to rule all three out, the discussion of whether RPE or ANCCR matches these results is probably moot. In particular, the hypotheses developed by Jeong et al., were for a random (unpredictable) rewards experiment, whereas the evidence points to the rewards being sometimes predictable. The learning of that predictability (e.g. over sessions) and variation in predictability (e.g. by attention level to sounds of each mouse) significantly complicate the analysis. The FLMM analysis reveals the complexity of analyzing what is apparently a straightforward task design.

While we are disappointed to hear the reviewer felt our initial interpretations and discussion were poor, the reviewer brings up an excellent point re: potential reward predictability that we had not considered. They have convinced us that acknowledging this alternative perspective will strengthen the paper, and we have added it into the Discussion. We agree that the ANCCR/RPE model predictions were made for unpredictable rewards and, as the reviewer rightly points out, there is evidence that the animals may sense the reward delivery. After discussing extensively with the authors of Jeong et al. (2022), it is clear that they went to enormous trouble to prevent the inadvertent generation of a CS+, and it is likely changes in pressure from the solenoid (rather than a sound) that may have served as a cue. Regardless of the learning theory one adopts (RPE, ANCCR or others), we agree that this potential learned predictability could, at least partially, account for the increase in signal magnitude across sessions. As this paper is focused on analysis methods, we feel that we can contribute most thoughtfully to the dopamine–learning theory conversation by presenting this explanation in detail, for consideration in future experiments. We have substantially edited this discussion and, as per the reviewer’s suggestion, have qualified our interpretations to reflect the uncertainty in explaining the observed trends.

If this paper is not trying to arbitrate between RPE and ANCCR, as stated in the text, the post hoc reasoning of the authors of Jeong et al 2022 provided in the discussion is not germane. Arbitrating between the models likely requires new experimental designs (removing the sound of the solenoid, satiety controls) or more complex models (e.g. with session effects, measures of predictability) that address the identified issues.

Thank you for this point. We agree with you that, given the scope of the paper, we should avoid any extensive comparison between the models. To address your comment, we have now removed portions of the Discussion that compared RPE and ANCCR. Overall, we agree with the reviewer, and think that future experiments will be needed for conclusively testing the accuracy of the models’ predictions for random (unpredicted) rewards. While we understand that our description of several conversations with the Jeong et al., 2022 authors could have gone deeper, we hope the reviewer can appreciate that inclusion of these conversations was done with the best of intentions. We wish to emphasize that we also consulted with several other researchers in the field when crafting our discussion. We do commend the authors of Jeong et al., 2022 for their willingness to discuss all these details. They could easily have avoided acknowledging any potential incompleteness of their theory by claiming that our results do not invalidate their predictions for a random reward, because the reward could potentially have been predicted (due to an inadvertent CS+ generated from the solenoid pressure). Instead, they emphasized that they thought their experiment did test a random reward, to the extent they could determine, and that our results suggest components of their theory that should be updated. We think that engagement with re-analyses of one’s data, even when findings are at odds with an initial theoretical framing, is a good demonstration of open science practice. For that reason as well, we feel providing readers with a perspective on the entire discussion will contribute to the scientific discourse in this area.

Finally, we would like to reiterate that this conversation is happening at least in part because of our method: by analyzing the signal at every trial timepoint, it provides a formal way to test for the presence of a neural signal indicative of reward delivery perception. Ultimately, this was what we set out to do: help researchers ask questions of their data that may have been harder to ask before. We believe that having a demonstration that we can indeed do this for a 'live' scientific issue is the most appropriate way of demonstrating the usefulness of the method.

Of the three potential causes of within-session decreases, the photobleaching arguments advanced in the discussion and expanded greatly in the appendices are not convincing. The data being modeled is a processed signal (∆*F/F*) with smoothing and baseline correction and this does not seem to have been considered in the argument. Furthermore, the photometry readout is also a convolution of the actual concentration changes over time, influenced by the on-off kinetics of the sensor, which makes the interpretation of timing effects of photobleaching less obvious than presented here and more complex than the dyes considered in the cited reference used as a foundation for this line of reasoning.

We appreciate the nuance of this point, and we have made considerable efforts in the Results and Discussion sections to caution that alternative hypotheses (e.g., photobleaching) cannot be definitively ruled out. In response to your criticism, we have consulted with more experts in the field regarding the potential for bleaching in this data, and it is not clear to us why photobleaching would be visible in one time-window of a trial, but not at another (less than a second away), despite high ∆*F/F* magnitudes in both time-windows. We do wish to point out that the Jeong et al. (2022) authors were also concerned about photobleaching as a possible explanation. At their request, we analyzed data from additional experiments, collected from the same animals. In most cases, we did not observe signal patterns that seemed to indicate photobleaching. Given the additional scrutiny, we do not think that photobleaching is more likely to invalidate results in this particular set of experiments than it would be in any other photometry experiment. While the role of photobleaching may be more complicated with this sensor than others in the references, that citation was included primarily as a way of acknowledging that it is possible that non-linearities in photobleaching could occur. Regardless, your point is well taken and we have qualified our description of these analyses to express that photobleaching cannot be ruled out.

Within this discussion of photobleaching, the characterization of the background reward experiments used in part to consider photobleaching (appendix 7.3.2) is incorrect. In this experiment (Jeong et al., 2022), background rewards were only delivered in the inter-trial-interval (i.e. not between the CS+ and predicted reward as stated in the text). Both in the authors’ description and in the data, there is a 6s before cue onset where rewards are not delivered and while not described in the text, the data suggests there is a period after a predicted reward when background rewards are not delivered. This complicates the comparison of this data to the random reward experiment.

Thank you for pointing this out! We removed the parenthetical on page 18 of the appendix that incorrectly stated that rewards can occur between the CS+ and the predicted reward.

The discussion of the lack of evidence for backpropagation, taken as evidence for ANCCR over RPE, is also weak.

Our point was initially included to acknowledge that, although our method yields results that conflict with the conclusions described by Jeong et al., 2022 on data from some experiments, on other experiments our method supports their results. Again, we believe that a critical part of re-analyzing shared datasets is acknowledging both areas where new analyses support the original results, as well as those where they conflict with them. We agree with the reviewer that qualifying our results so as not to emphasize support for/against RPE/ANCCR will strengthen our paper, and we have made those changes. We have qualified the conclusions of our analysis to emphasize they are a demonstration of how *FLMM* can be used to answer a certain style of question with hypothesis testing (how signal dynamics change across sessions), as opposed to providing evidence for/against the backpropagation hypothesis.

A more useful exercise than comparing FLMM to the methods and data of Jeong et al., 2022, would be to compare against the approach of Amo et al., 2022, which identifies backpropagation (data publicly available: DOI: 10.5061/dryad.hhmgqnkjw). The replication of a positive result would be more convincing of the sensitivity of the methodology than the replication of a negative result, which could be a result of many factors in the experimental design. Given that the Amo et al. analysis relies on identifying systematic changes in the timing of a signal over time, this would be particularly useful in understanding if the smoothing steps in FLMM obscure such changes.

Thank you for this suggestion. Your thoughtful review has convinced us that focusing on our statistical contribution will strengthen the paper, and we made changes to further emphasize that we are not seeking to adjudicate between RPE/ANCCR. Given the length of the manuscript as it stands, we could only include a subset of the analyses conducted on Jeong et al., 2022, and had to relegate the results from the Coddington et al., data to an appendix. Realistically, it would be hard for us to justify including analyses from a third dataset, only to have to relegate them to an appendix. We did include numerous examples in our manuscript where we already replicated positive results, in a way that we believe demonstrates the sensitivity of the methodology. We have also been working with many groups at NIH and elsewhere using our approach, in experiments targeting different scientific questions. In fact, one paper that extensively applies our method, and compares the results with those yielded by standard analysis of AUCs, is already published (Beas et al., 2024). Finally, in our analysis guide we describe additional analyses, not included in the manuscript, that replicate positive results. Hence there are numerous demonstrations of *FLMM*’s performance in less controversial settings. We take your point that our description of the data supporting one theory or the other should be qualified, and we have corrected that. Specifically for your suggestion of Amo et al. 2022, we have not had the opportunity to personally reanalyze their data, but we are already in contact with other groups who have conducted preliminary analyses of their data with *FLMM*. We are delighted to see this, in light of your comments and our decision to restrict the scope of our paper. We will help them and other groups working on this question to the extent we can.

**Recommendations for the Authors:**

**Reviewer #2:**
First, I would like to commend the authors for the clarity of the paper, and for creating an open-source package that will help researchers more easily adopt this type of analysis.

Thank you for the positive feedback!

I would suggest the authors consider adding to the manuscript, either some evidence or some intuition on how feasible would be to use FLMM for very complex model specifications, in terms of computational cost and model convergence.

Thank you for this suggestion. As we described above in response to Reviewer #2’s Public Reviews, we have added in a demonstration of the scalability of the method. Since our initial manuscript submission, we have further increased the package’s speed (e.g., through further parallelization). We are releasing the updated version of our package on CRAN.

From my understanding, this package might potentially be useful not just for photometry data but also for two-photon recordings for example. If so, I would also suggest the authors add to the discussion this potential use.

This is a great point. Our updated manuscript Discussion includes the following:

“The *FLMM* framework may also be applicable to techniques like electrophysiology and calcium imaging. For example, our package can fit functional generalized LMMs with a count distribution (e.g., Poisson). Additionally, our method can be extended to model time-varying covariates. This would enable one to estimate how the level of association between signals, simultaneously recorded from different brain regions, fluctuates across trial time-points. This would also enable modeling of trials that differ in length due to, for example, variable behavioral response times (e.g., latency-topress).”

**Reviewer #3:**
The authors should define ’function’ in context, as well as provide greater detail of the alternate tests that FLMM is compared to in Figure 7.

We include a description of the alternate tests in Appendix Section 5.2. We have updated the Methods Section (Section 4) to introduce the reader to how ‘functions’ are conceptualized and modeled in the functional data analysis literature. Specifically, we added the following text:

*“FLMM* models each trial’s signal as a function that varies smoothly across trial time-points (i.e., along the 'functional domain'). It is thus a type of non-linear modeling technique over the functional domain, since we do not assume a linear model (straight line). *FLMM* and other functional data analysis methods model data as functions, when there is a natural ordering (e.g., time-series data are ordered by time, imaging data are ordered by x-y coordinates), and are assumed to vary smoothly along the functional domain (e.g., one assumes values of a photometry signal at close time-points in a trial have similar values). Functional data analysis approaches exploit this smoothness and natural ordering to capture more information during estimation and inference.”

Given the novelty of estimating joint CIs, the authors should be clearer about how this should be reported and how this differs from pointwise CIs (and how this has been done in the past).

We appreciate your pointing this out, as the distinction is nuanced. Our manuscript includes a description of how *joint* CIs enable one to interpret effects as statistically significant for time-intervals as opposed to individual timepoints. Unlike *joint* CIs, assessing significance with pointwise CIs suffers from multiple-comparisons problems. As a result of your suggestion, we have included a short discussion of this to our analysis guide (Part 1), entitled 'Pointwise or Joint 95% Confidence Intervals.' The Methods section of our manuscript also includes the following:

“The construction of *joint* CIs in the context of functional data analysis is an important research question; see Cui et al. (2021) and references therein. Each *point* at which the *pointwise* 95% CI does not contain 0 indicates that the coefficient is *statistically significantly* different from 0 at that point. Compared with *pointwise* CIs, *joint* CIs takes into account the autocorrelation of signal values across trial time-points (the functional domain). Therefore, instead of interpreting results at a specific timepoint, *joint* CIs enable *joint* interpretations at multiple locations along the functional domain. This aligns with interpreting covariate effects on the photometry signals across time-intervals (e.g., a cue period) as opposed to at a single trial time-point. Previous methodological work has provided functional mixed model implementations for either *joint* 95% CIs for simple random-effects models (Cui et al., 2021), or *pointwise* 95% CIs for nested models (Scheipl et al., 2016), but to our knowledge, do not provide explicit formulas or software for computing *joint* 95% CIs in the presence of general random-effects specifications.”

The authors identify that many photometry studies are complex nested longitudinal designs, using the cohort of 8 animals used in five task designs of Jeong et al. 2022 as an example. The authors miss the opportunity to illustrate how FLMM might be useful in identifying the effects of subject characteristics (e.g. sex, CS+ cue identity).

This is a fantastic point and we have added the following into the Discussion:

“...[S]ignal heterogeneity due to subject characteristics (e.g., sex, CS+ cue identity) could be incorporated into a model through inclusion of animal-specific random effects.”

In discussing the delay-length change experiment, it would be more accurate to say that proposed versions of RPE and ANCCR do not predict the specific change.

Good point. We have made this change.

Minor corrections:Panels are mislabeled in Figure 5.

Thank you. We have corrected this.

The Crowder (2009) reference is incorrect, being a review of the book with the book presumably being the correct citation.

Good catch, thank you! Corrected.

In Section 5 (first appendix), the authors could include the alternate spelling ’fibre photometry’ to capture any citations that use British English spelling.

This is a great suggestion, but we did not have time to recreate these figures before re-submission.

Section 7.4 is almost all quotation, though unevenly using the block quotation formatting. It is unclear why such a large quotation is included.

Thank you for pointing this out. We have removed this Appendix section (formerly Section 7.4) as the relevant text was already included in the Methods section.

References

Sofia Beas, Isbah Khan, Claire Gao, Gabriel Loewinger, Emma Macdonald, Alison Bashford, Shakira Rodriguez-Gonzalez, Francisco Pereira, and Mario A Penzo. Dissociable encoding of motivated behavior by parallel thalamo-striatal projections. *Current Biology*, 34(7):1549–1560, 2024.

Erjia Cui, Andrew Leroux, Ekaterina Smirnova, and Ciprian Crainiceanu. Fast univariate inference for longitudinal functional models. *Journal of Computational and Graphical Statistics*, 31:1–27, 07 2021. doi: 10.1080/10618600.2021.1950006.

Huijeong Jeong, Annie Taylor, Joseph R Floeder, Martin Lohmann, Stefan Mihalas, Brenda Wu, Mingkang Zhou, Dennis A Burke, and Vijay Mohan K Namboodiri. Mesolimbic dopamine release conveys causal associations. *Science*, 378(6626):eabq6740, 2022. doi: 10.1126/science.abq6740. URL https://www. science.org/doi/abs/10.1126/science.abq6740.

Rachel S Lee, Marcelo G Mattar, Nathan F Parker, Ilana B Witten, and Nathaniel D Daw. Reward prediction error does not explain movement selectivity in dms-projecting dopamine neurons. *eLife*, 8:e42992, apr 2019. ISSN 2050-084X. doi: 10.7554/eLife.42992. URL https://doi.org/10.7554/eLife.42992.

Fabian Scheipl, Jan Gertheiss, and Sonja Greven. Generalized functional additive mixed models. *Electronic Journal of Statistics*, 10(1):1455 – 1492, 2016. doi: 10.1214/16-EJS1145. URL https://doi.org/10.1214/16-EJS1145.